# Don't Go Breaking My LLM: The Impact of Pruning Attention Layers on Explanation Faithfulness and Confidence Calibration

**Pietro Tropeano**                                                   *pitr@di.ku.dk*
*University of Copenhagen*
*Copenhagen, Denmark*

**Maria Maistro**                                                      *mm@di.ku.dk*
*University of Copenhagen*
*Copenhagen, Denmark*

**Tuukka Ruotsalo**                                                    *tr@di.ku.dk*
*University of Copenhagen*
*Copenhagen, Denmark*
*LUT University*
*Lahti, Finland*

**Christina Lioma**                                                    *c.lioma@di.ku.dk*
*University of Copenhagen*
*Copenhagen, Denmark*

**Reviewed on OpenReview:** *https://openreview.net/forum?id=VxZd6HfMOo*

## Abstract

Pruning Large Language Models (LLMs) reduces memory and inference costs by removing parts of the network, producing smaller models that retain most of their accuracy. As attention layers are the most resource-intensive parts of LLMs, pruning them is a promising compression strategy. Prior work shows that up to 33% of attention layers can be pruned with minimal accuracy loss. Nevertheless, the impact of attention pruning on model interpretability, specifically faithfulness and confidence calibration, remains unstudied. To address this gap, we study how pruning attention layers affects explanation faithfulness and confidence calibration across five LLMs and eight datasets. While the pruned models often maintain high accuracy, we find that their faithfulness and calibration often degrade. Notably, faithfulness and calibration can fluctuate significantly, even when accuracy remains stable, highlighting a misalignment between model confidence, interpretability, and accuracy. Our findings suggest that layer pruning can affect LLMs' interpretability and reliability in ways not captured by accuracy and efficiency measures alone. We recommend including explainability and calibration metrics when evaluating pruned models. The code is available at: `https://github.com/pietrotrope/Dont_Go_Breaking_My_LLM`

## 1 Introduction

Large Language Models (LLMs) have achieved exceptional results across tasks, yet their high computational demands make them costly to deploy. Pruning is a family of model compression techniques that reduces inference time and memory requirements by removing parts of the model, while aiming to preserve performance. A specific type of pruning is layer removal, also known as layer drop (Gromov et al., 2025; He et al., 2026), which cuts entire layers from the model. This method is simple and model agnostic, making

it relevant in current research. Layer removal in LLMs can reduce computational costs with minimal accuracy loss, especially when pruning attention layers rather than MLPs; up to 33% of attention layers can be removed without harming performance (Siddiqui et al., 2024). However, the impact of layer pruning on key aspects of interpretability, such as faithfulness (do model rationales accurately reflect the reasoning process of the model?) and confidence calibration (do predicted probabilities reflect true likelihoods?), remains underexplored (Jacovi & Goldberg, 2020; Naeini et al., 2015).

To fill this gap, we present the first systematic study of how pruning attention layers affects explanation faithfulness and confidence calibration. Specifically, we make three contributions. First, we provide the first evaluation of the impact of attention layer pruning on the faithfulness of LIME (Ribeiro et al., 2016) and Kernel SHAP (Lundberg & Lee, 2017) feature attributions, measured using comprehensiveness and sufficiency (DeYoung et al., 2020). Second, we present the first analysis of how attention layer pruning influences the alignment between model confidence and accuracy by computing the Estimated Calibration Error (ECE) (Naeini et al., 2015). Third, we relate changes in explanation faithfulness and confidence alignment to changes in model accuracy, investigating whether accuracy degradation due to attention layer pruning is a good proxy for its effects on explanation faithfulness and confidence calibration.

We find that pruning attention layers negatively affects LLMs' explanation faithfulness. Notably, changes in faithfulness are often unstable and can occur independently of changes in accuracy. For both pruned and unpruned models, we analyse the calibration between model confidence and accuracy and find that removing more layers generally causes them to follow different trends. As with faithfulness, the effect of pruning on calibration is not aligned with the effect on accuracy, highlighting the limitations of using only accuracy to evaluate pruned models. Overall, we provide novel evidence that attention layer pruning can unpredictably degrade model interpretability and confidence calibration in ways that are not captured by accuracy.

## 2 Related work

Model pruning is a family of compression approaches used to improve the efficiency of LLMs by removing parts of the model, such as parameters or structures (e.g., neurons, features, layers). Pruning methods aim to reduce model size, lower computational costs, and accelerate inference, while preserving the model's performance. Pruning methods can be either unstructured or structured.

**Unstructured Pruning.** Unstructured pruning methods remove individual parameters (weights) from a neural network without enforcing any specific sparsity pattern. There exist several unstructured pruning methods for LLMs, such as SparseGPT (Frantar & Alistarh, 2023), Wanda (Sun et al., 2024), and RIA (Zhang et al., 2024d). These methods can preserve most of the models' performance while increasing their sparsity. However, even though unstructured pruning reduces the number of parameters, it often does not lead to faster inference on standard hardware (e.g., CPUs or GPUs), since the resulting sparse patterns are irregular and hard to leverage (Li et al., 2017). Recent pruners, therefore, include a semi-structured variant, which imposes some structure on the sparsity patterns and can, in principle, enable hardware acceleration (Li et al., 2017; Frantar & Alistarh, 2023; Sun et al., 2024; Zhang et al., 2024d;b).

**Structured Pruning and Layer Removal.** Structured pruning methods remove entire structures, such as attention heads, neurons, or entire layers, from a neural network. Several structured pruning methods have been developed for LLMs (Ma et al., 2023; Xia et al., 2024; Michel et al., 2019; Ashkboos et al., 2024; Men et al., 2024). These methods can lead to tangible efficiency improvements; however, they tend to yield worse performance to sparsity trade-offs compared to unstructured pruning (Hu et al., 2025).

A subset of the structured pruning literature specifically targets the removal of entire layers from the network. Siddiqui et al. (2024) investigate the removal of both MLP and attention layers in decoder-based architectures, finding that pruning attention layers leads to smaller accuracy drops compared to pruning MLPs. Remarkably, up to 33% of attention layers can be removed without compromising performance on specific downstream tasks. Additionally, pruning attention layers reduces the computational cost associated with the KV-cache. In another example, Gromov et al. (2025) remove layers using the similarity between their input and output representations as a proxy for layer importance. Yet, studies on how layer pruning

Table 1: Number of samples, number of classes, and task for each dataset.

| Dataset | # samples | # Classes | Task |
|---|---|---|---|
| ARC Easy | 2376 | 4 | Multiple Choice QA |
| ARC Challenge | 1172 | 4 | Multiple Choice QA |
| OpenBookQA | 500 | 4 | Multiple Choice QA |
| BoolQ | 3270 | 2 | Question Answering (Yes/No) |
| TweetEval | 2000 | 3 | Sentiment Analysis |
| Rotten Tomatoes | 1066 | 2 | Sentiment Analysis |
| SST-2 | 872 | 2 | Sentiment Analysis |
| RTE | 277 | 2 | Natural Language Inference |

Table 2: Number of parameters, number of layers, and vocabulary size for each LLM.

| Model | # parameters | # layers | vocab size |
|---|---|---|---|
| Mistral 7B | 7B | 32 | 32,000 |
| Llama-2 7B | 7B | 32 | 32,000 |
| Llama-3 8B | 8B | 32 | 128,256 |
| Qwen-3 4B | 4B | 36 | 151,936 |
| Qwen-3 8B | 8B | 36 | 151,936 |

Table 3: Summary of the used feature attribution methods and evaluation measures.

| Effectiveness | Features attribution | Faithfulness | Confidence Calibration |
|---|---|---|---|
| Accuracy | LIME | Comprehensiveness | Expected Calibration Error (ECE) |
| $\Delta$ Accuracy | Kernel SHAP | Sufficiency | |

affects model performance have primarily focused on measures like accuracy and inference time (Men et al., 2024; He et al., 2026; Zhang et al., 2024c; Siddiqui et al., 2024; Kim et al., 2024; Song et al., 2024).

**Pruned models evaluation.** Most pruning studies evaluate models using accuracy and inference time (Frantar & Alistarh, 2023; Sun et al., 2024; Zhang et al., 2024d; Men et al., 2024; He et al., 2026; Zhang et al., 2024c; Siddiqui et al., 2024; Kim et al., 2024; Song et al., 2024). Other work has examined internal properties of pruned models, such as the similarity of representations across layers (Gromov et al., 2025; Song et al., 2024; Zhang et al., 2024a). Prior work also focuses on pruned models' fact recall and retrieval abilities (Jin et al., 2023), hallucinations in abstractive summarisation (Chrysostomou et al., 2024), and on inspecting the effect of pruning on models' biases and stereotypes Kirsten et al. (2025); Hooker et al. (2020), fairness (Ramesh et al., 2023), toxicity (Xu et al., 2024), lie detection (Fu et al., 2025), and safety (Hasan et al., 2024; GNVV et al., 2025). However, current research still overlooks other critical models' properties. Two such properties are faithfulness, which measures how accurately attributions capture the model's actual reasoning process (Jacovi & Goldberg, 2020; Lyu et al., 2024; Serrano & Smith, 2019; DeYoung et al., 2020; Ju et al., 2022), and confidence calibration, which measures how well the model's predicted probabilities reflect actual accuracy (Naeini et al., 2015). Both are important for developing interpretable and trustworthy AI systems.

Faithfulness and confidence calibration have been studied in the context of LLM evaluation (Jacovi & Goldberg, 2020; DeYoung et al., 2020; Guo et al., 2017; Naeini et al., 2015), but remain unexplored in the area of layer pruning. It is precisely this research gap that our work addresses.

## 3 Methodology

We study the effects of layer pruning, where entire layers are removed from the models. All analyses and results in this work are defined within this setting; different pruning settings (e.g., unstructured or weight-based pruning) may exhibit different behaviours. Precisely, we study if pruning entire layers in LLMs affects key model properties beyond accuracy, specifically explanation faithfulness and confidence calibration, and how these change in relation to accuracy. To this end, we evaluate accuracy, explanations' faithfulness, plausibility, and confidence calibration across five LLMs and eight datasets, progressively removing up to 16 attention layers. See Tabs. 1, 2, and 3 for a summary of the datasets, models, feature attributions, and measures of effectiveness, faithfulness of explanations, and confidence calibration in the paper.

### 3.1 Datasets and Language Models

**Datasets.** We use eight established datasets (in their default versions): ARC Easy and ARC Challenge (Clark et al., 2018), OpenBookQA (Mihaylov et al., 2018), BoolQ (Clark et al., 2019), RTE (Wang et al., 2018), the sentiment partition of TweetEval (Rosenthal et al., 2017), Rotten Tomatoes (Pang & Lee, 2005), and SST-2 (Socher et al., 2013). We use zero shot prompts available on LM Evaluation Harness (Gao et al., 2024), except for TweetEval and Rotten Tomatoes, for which we select prompt templates from promptsource

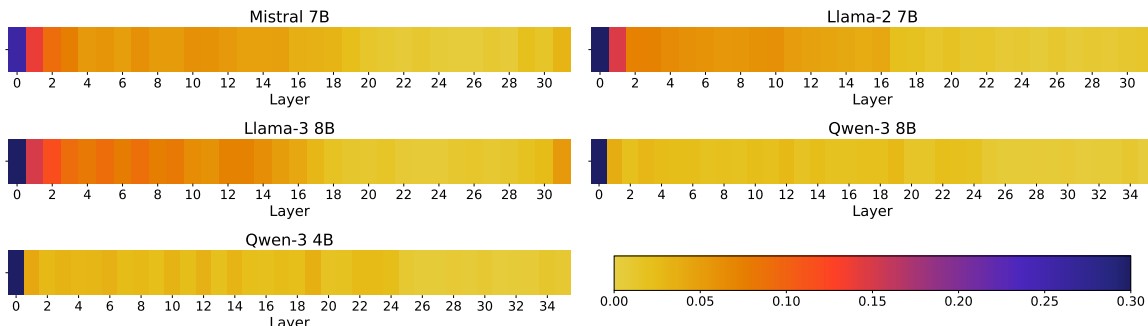

Figure 1: Importance scores of attention layers for all models.The importance scores are computed as 1 - the cosine similarity between the layer's input and output. A score of 0 means the lowest importance.

(Bach et al., 2022). See Sec. A.2 of the Appendix for the exact prompt templates. An overview of the datasets is in Tab. 1, and more details are available in Sec. A.1 of the Appendix.

**Language Models.** We use the 7B version of two widely used open-source models: Mistral (Jiang et al., 2023) and Llama-2 (Touvron et al., 2023), Qwen-3 in its 4B and 8B variants, and Llama-3 in its 8B variant (Yang et al., 2025; Team, 2024). An overview of the models is in Tab. 2, and more details are available in Sec. A.3 of the Appendix.

### 3.2 Pruning Attention Layers

In this study, we adopt a simple approach to layer pruning introduced by He et al. (2026), which removes layers from an LLM based on their relative importance. The importance score of each layer, or *block influence*, is estimated by computing the cosine similarity between the layer's input and output representations (Men et al., 2024). This calculation is based on a small amount of calibration data, i.e., data samples that are used to guide pruning. He et al. (2026) show that their pruning method is robust to the amount of calibration data, therefore, we use eight sequences of 2048 tokens sampled from the C4 dataset (Raffel et al., 2020). Layers are pruned by descending input-output similarity. Fig. 1 shows the importance score for each attention layer in all LLMs. In line with prior work (He et al., 2026; Men et al., 2024; Gromov et al., 2025), we find that early layers tend to have higher importance overall, especially for Mistral and Llama. For Qwen, this trend is less consistent, with only the first layer showing notably higher importance. Later layers remain more likely to be pruned across models. We evaluate the model at each pruning step, progressively removing one attention layer at a time, up to 16 layers. We do not consider MLP pruning in our experiments, as, unlike attention layer pruning, which largely preserves performance, removing even a few (e.g., four) MLP layers leads to substantial accuracy drops. For example, in Siddiqui et al. (2024), Mistral's MMLU (Hendrycks et al., 2021b;a) accuracy drops from over 0.6 to less than 0.3 after removing four MLP layers and to less than 0.25 (near random) after removing 12 layers, while remaining above 0.6 when removing up to 12 attention layers. Similarly, in He et al. (2026), Mistral's average accuracy across multiple tasks decreases from 0.703 to 0.534 when removing 8 MLP layers, whereas removing 8 attention layers leads to 0.697 accuracy (negligible drop). For LLaMA, removing 8 MLP layers reduces accuracy from 0.682 to 0.619, compared to 0.681 for attention removal. In addition, prior work shows that MLP layers are critical for LLM reasoning (Shao & Wu, 2025). These results indicate that removing MLP layers eliminates a critical component of the network and rapidly degrades model performance, often to near-random levels. In such settings, model outputs become unreliable, making it not meaningful to evaluate explanation quality on severely degraded predictions.

### 3.3 Explanation Faithfulness Evaluation

Faithfulness captures how accurately an explanation reflects the reasoning process of the model (Jacovi & Goldberg, 2020; Lyu et al., 2024). Common approaches to evaluate faithfulness require studying how removing features leads to changes in model predictions (Serrano & Smith, 2019), or evaluating different input perturbations (DeYoung et al., 2020; Ju et al., 2022). For perturbation-based measures, it is common

to compute their Area Over the Perturbation Curve (AOPC), which quantifies the average change in the scores as increasing amounts of features are removed. In this work, we adopt two AOPC-based measures introduced by DeYoung et al. (2020): *comprehensiveness* and *sufficiency*. These measures quantify how model's confidence changes when the most important input features are removed (comprehensiveness) or retained in isolation (sufficiency). We choose these measures because, in a study against four other measures, they were shown to favour more faithful explanations while being efficient to compute (Chan et al., 2022). **Comprehensiveness** is defined as:

$$\frac{1}{|\beta|+1} \sum_{k=0}^{|\beta|} [m_j(x_i) - m_j(x_i \setminus r_{ik})] \tag{1}$$

where $m_j(x_i)$ is the model's confidence in the predicted class $j$, given input $x_i$. The set $\beta$ contains bins representing different proportions of top-ranked features (e.g., 10%, 20%, 30%). Lastly, $r_{ik}$ is the most important features in bin $k$ identified by an attribution algorithm. The measure ranges in $[-1, 1]$. A high score indicates that the most important features are crucial for the prediction, suggesting that the attributions are faithful to the model. In contrast, a negative score implies that the model becomes more confident when the important features are removed, meaning the attributions may not be faithful to the model.

**Sufficiency** is defined analogously, but considers the difference in the model's confidence on the full input and a perturbed input where only the important features are retained:

$$\frac{1}{|\beta|+1} \sum_{k=0}^{|\beta|} [m_j(x_i) - m_j(r_{ik})] \tag{2}$$

The measure ranges in the interval $[-1, 1]$. A high score indicates that the most important features are not sufficient for the prediction, suggesting that the explanations may not be faithful. A negative score, on the other hand, means the model becomes more confident when presented with only the most important features, supporting the claim that they are sufficient for the prediction.

We compute comprehensiveness and sufficiency (Eq. equation 1 & equation 2), using the top 10%, 20%, 30%, 40%, and 50% of tokens ranked by their attribution scores found with two attribution methods: LIME (Ribeiro et al., 2016), and Kernel SHAP (Lundberg & Lee, 2017). We use LIME and Kernel SHAP as they are based on local model-agnostic explanations, which tend to perform well on faithfulness tasks and yield more comprehensive explanations than attention-based methods (DeYoung et al., 2020). As LIME and Kernel SHAP have a stochastic component, we average all experiments over three random seeds. For computational reasons, we compute these measures on a subset of 200 randomly sampled instances per dataset.[5] For each sample, we use both the task prompt and the sample text to compute token-level importance scores.

### 3.4 Confidence Calibration Evaluation

A trustworthy model should exhibit confidence that closely reflects its actual accuracy (Guo et al., 2017). Such well-calibrated models are not only more interpretable, but can also enhance downstream performance (Guo et al., 2017). To quantify calibration, Naeini et al. (2015) introduced the *Expected Calibration Error (ECE)*, a measure that approximates the difference between a model's predicted confidence and its accuracy. ECE is computed by dividing predictions into $M$ equally spaced confidence bins and calculating the weighted average of the absolute differences between accuracy and confidence within each bin:

$$\text{ECE} = \sum_{l=1}^{M} \frac{|B^l|}{n} \left| \text{acc}(B^l) - \sum_{i=1}^{B^l} \frac{m_j(B_i^l)}{|B^l|} \right| \tag{3}$$

where $B^l$ is the $l$-th bin, $n$ is the total number of samples, $\text{acc}(B^l)$ is the model's accuracy on the samples in bin $l$, and $m_j(B_i^l)$ is the confidence in the predicted class $j$, given the $i$-th input in bin $B^l$. The measure

---

[5]We provide the exact indices of the sampled texts together with the code

Table 4: Inter-annotator agreement per dataset measured using Jaccard distance and Krippendorff's alpha.

| Dataset | Jaccard distance | Krippendorff's alpha |
|---|---|---|
| ARC Easy | 0.666 | 0.664 |
| ARC Challenge | 0.365 | 0.378 |
| OpenBookQA | 0.443 | 0.385 |
| BoolQ | 0.294 | 0.290 |
| TweetEval | 0.384 | 0.352 |
| Rotten Tomatoes | 0.399 | 0.326 |
| SST-2 | 0.426 | 0.391 |
| RTE | 0.402 | 0.409 |

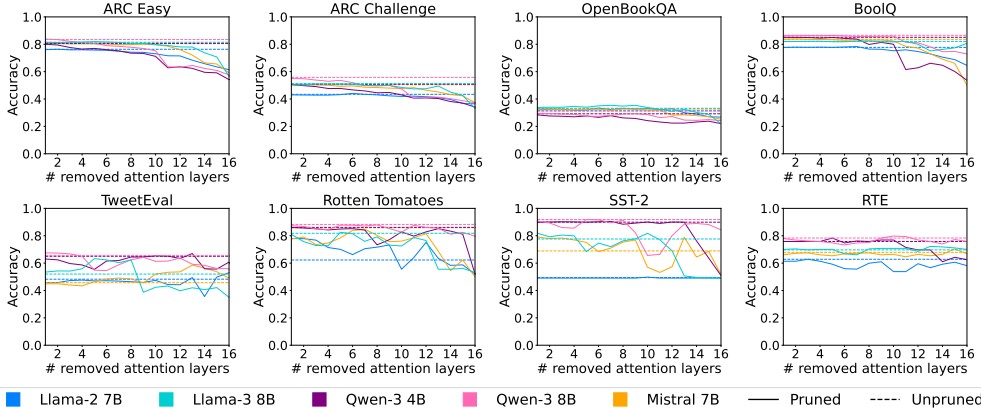

Figure 2: Accuracy of pruned and unpruned models (y axis) on eight different datasets, with varying numbers of removed attention layers (x axis). Accuracy of unpruned models is shown with dashed lines.

ranges in the interval $[0, 1]$. An ECE score near zero reflects well calibrated confidence (predicted probabilities align well with actual accuracy). Higher ECE scores reflect greater misalignment between confidence and accuracy, or less trustworthy model predictions. We measure ECE using 10 bins as per Guo et al. (2017).

### 3.5 Explanation Plausibility Evaluation

We study how attention layer pruning affects the plausibility of model explanations. To this end, four annotators manually annotated eight randomly sampled instances per dataset by selecting text spans they deemed relevant to the task (Hayati et al., 2021). Of these, three samples were shared across annotators to assess agreement, while the remaining five were unique to each annotator. The instructions given to the annotators are available in App. A.5. Annotations were converted to word-level relevance labels. Tab. 4 shows the Inter-annotator agreement on shared samples. Agreement is high for ARC Easy and moderate for OpenBookQA and RTE. Consistent with prior work on similar annotation protocols, for the remaining datasets, agreement is generally low (Hayati et al., 2021).

We use these annotations to evaluate the plausibility of feature attributions across both pruned and unpruned models, and across all datasets. Following prior work (Edin et al., 2024), we measure Area Under the Precision-Recall Curve (AUPRC), Precision@$K$, Recall@$K$, F1@$K$, and Intersection-over-Union (IoU) DeYoung et al. (2020). Except for AUPRC, which evaluates performance across multiple decision boundaries, we avoid additional classification-based metrics as they require selecting an optimal decision boundary on a validation set, which is unavailable in our setting. We consider $K \in 10, 20, 30, 40, 50\%$ of features. For IoU, we match the number of selected features to the number of features identified by annotators.

## 4 Results

Next, we present our experimental results. We begin by evaluating the impact of pruning on model accuracy in Sec. 4.1. We then turn to faithfulness in Sec. 4.2, examining the quality of feature attributions and

Table 5: Summary of Accuracy (Acc) and Δ Accuracy (Δ) across all datasets and models when removing 1, 4, 8, 12 and 16 attention layers. The highest drop in accuracy is bolded for each model configuration. Accuracy increases are underlined. * indicates statistical significance ($p < 0.05$) under the McNemar test. Results for the remaining configurations are available in Appendix, Tabs. 8-47.

| Model | # removed layers | ARC Easy Acc | Δ | ARC Challenge Acc | Δ | OpenBookQA Acc | Δ | BoolQ Acc | Δ | TweetEval Acc | Δ | Rotten Tomatoes Acc | Δ | SST-2 Acc | Δ | RTE Acc | Δ |
|---|---|---|---|---|---|---|---|---|---|---|---|---|---|---|---|---|---|
| Llama-2 | (orig) | 0.763 | +0.000 | 0.434 | +0.000 | 0.314 | +0.000 | 0.777 | +0.000 | 0.482 | +0.000 | 0.623 | +0.000 | 0.494 | +0.000 | 0.628 | +0.000 |
| Llama-2 | 1 | 0.761 | -0.003 | 0.430 | -0.004 | 0.320 | +0.006 | 0.777 | -0.001 | 0.458* | **-0.025** | 0.780* | +0.157 | 0.491 | -0.003 | 0.614 | -0.014 |
| Llama-2 | 4 | 0.760 | -0.004 | 0.428 | -0.006 | 0.318 | +0.004 | 0.779 | +0.002 | 0.476 | -0.007 | 0.714* | +0.091 | 0.491 | -0.003 | 0.614 | **-0.014** |
| Llama-2 | 8 | 0.742* | -0.021 | 0.430 | -0.004 | 0.332 | +0.018 | 0.765 | -0.013 | 0.464* | -0.018 | 0.717* | +0.094 | 0.491 | -0.003 | 0.599 | **-0.029** |
| Llama-2 | 12 | 0.715* | **-0.048** | 0.417 | -0.017 | 0.328 | +0.014 | 0.747* | -0.031 | 0.442* | -0.040 | 0.751* | +0.129 | 0.491 | -0.003 | 0.596 | -0.032 |
| Llama-2 | 16 | 0.614* | **-0.149** | 0.365* | -0.069 | 0.268* | -0.046 | 0.646* | -0.132 | 0.530* | +0.048 | 0.520* | -0.103 | 0.491 | -0.003 | 0.581 | -0.047 |
| Llama-3 | (orig) | 0.815 | +0.000 | 0.513 | +0.000 | 0.332 | +0.000 | 0.821 | +0.000 | 0.520 | +0.000 | 0.818 | +0.000 | 0.776 | +0.000 | 0.697 | +0.000 |
| Llama-3 | 1 | 0.814 | -0.000 | 0.504 | **-0.009** | 0.338 | +0.006 | 0.820 | -0.000 | 0.534* | +0.014 | 0.811 | -0.008 | 0.819* | +0.042 | 0.700 | +0.004 |
| Llama-3 | 4 | 0.816 | +0.002 | 0.509 | -0.004 | 0.346 | +0.014 | 0.822 | +0.001 | 0.558* | +0.038 | 0.727* | **-0.091** | 0.799 | +0.023 | 0.700 | +0.004 |
| Llama-3 | 8 | 0.806* | -0.009 | 0.497 | -0.016 | 0.348 | +0.016 | 0.823 | +0.002 | 0.624* | +0.104 | 0.792* | **-0.026** | 0.763 | -0.014 | 0.704 | +0.007 |
| Llama-3 | 12 | 0.781* | -0.034 | 0.475* | -0.038 | 0.310 | -0.022 | 0.800* | -0.021 | 0.398* | **-0.122** | 0.765* | -0.053 | 0.690* | -0.086 | 0.686 | -0.011 |
| Llama-3 | 16 | 0.564* | -0.250 | 0.331* | -0.182 | 0.220* | -0.112 | 0.807* | -0.014 | 0.350* | -0.170 | 0.531* | **-0.287** | 0.491* | -0.286 | 0.700 | +0.004 |
| Mistral | (orig) | 0.809 | +0.000 | 0.504 | +0.000 | 0.326 | +0.000 | 0.836 | +0.000 | 0.457 | +0.000 | 0.858 | +0.000 | 0.689 | +0.000 | 0.679 | +0.000 |
| Mistral | 1 | 0.808 | -0.001 | 0.500 | -0.004 | 0.328 | +0.002 | 0.836 | +0.001 | 0.449* | -0.009 | 0.777* | **-0.082** | 0.790* | +0.101 | 0.661 | -0.018 |
| Mistral | 4 | 0.804 | -0.005 | 0.501 | -0.003 | 0.320 | -0.006 | 0.835 | -0.000 | 0.434* | -0.023 | 0.747* | **-0.112** | 0.769* | +0.080 | 0.661 | -0.018 |
| Mistral | 8 | 0.798* | -0.011 | 0.484* | -0.020 | 0.328 | +0.002 | 0.830 | -0.006 | 0.484* | +0.027 | 0.803* | **-0.055** | 0.753* | +0.064 | 0.675 | -0.004 |
| Mistral | 12 | 0.761* | -0.048 | 0.464* | -0.040 | 0.282* | -0.044 | 0.772* | -0.064 | 0.536* | +0.079 | 0.815* | -0.043 | 0.573* | **-0.116** | 0.664 | -0.014 |
| Mistral | 16 | 0.592* | -0.216 | 0.373* | -0.131 | 0.252* | -0.074 | 0.501* | -0.335 | 0.490 | +0.033 | 0.500* | **-0.358** | 0.517* | -0.172 | 0.668 | -0.011 |
| Qwen-3 8B | (orig) | 0.835 | +0.000 | 0.558 | +0.000 | 0.312 | +0.000 | 0.866 | +0.000 | 0.656 | +0.000 | 0.883 | +0.000 | 0.919 | +0.000 | 0.783 | +0.000 |
| Qwen-3 8B | 1 | 0.838 | +0.003 | 0.549 | -0.009 | 0.296 | **-0.016** | 0.863 | -0.002 | 0.675* | +0.019 | 0.873* | -0.009 | 0.908* | -0.010 | 0.780 | -0.004 |
| Qwen-3 8B | 4 | 0.811* | -0.023 | 0.530* | -0.028 | 0.288 | -0.024 | 0.862 | -0.003 | 0.650 | -0.006 | 0.845* | -0.038 | 0.856* | **-0.063** | 0.758* | -0.025 |
| Qwen-3 8B | 8 | 0.778* | -0.056 | 0.497* | **-0.061** | 0.290 | -0.022 | 0.857* | -0.009 | 0.624* | -0.032 | 0.879 | -0.004 | 0.913 | -0.006 | 0.751* | -0.032 |
| Qwen-3 8B | 12 | 0.632* | **-0.203** | 0.416* | -0.142 | 0.272 | -0.040 | 0.811* | -0.055 | 0.636* | -0.020 | 0.861 | -0.022 | 0.800* | -0.118 | 0.769 | -0.014 |
| Qwen-3 8B | 16 | 0.576* | **-0.258** | 0.373* | -0.185 | 0.242* | -0.070 | 0.729* | -0.136 | 0.559* | -0.097 | 0.796* | -0.086 | 0.843* | -0.076 | 0.758 | -0.025 |
| Qwen-3 4B | (orig) | 0.805 | +0.000 | 0.508 | +0.000 | 0.292 | +0.000 | 0.851 | +0.000 | 0.650 | +0.000 | 0.861 | +0.000 | 0.899 | +0.000 | 0.758 | +0.000 |
| Qwen-3 4B | 1 | 0.800 | -0.005 | 0.507 | -0.001 | 0.284 | -0.008 | 0.851 | +0.000 | 0.630* | **-0.020** | 0.860 | -0.001 | 0.899 | +0.000 | 0.758 | +0.000 |
| Qwen-3 4B | 4 | 0.765* | -0.040 | 0.477* | -0.031 | 0.270 | -0.022 | 0.845 | -0.006 | 0.587* | **-0.064** | 0.842* | -0.019 | 0.901 | +0.002 | 0.758 | +0.000 |
| Qwen-3 4B | 8 | 0.735* | -0.070 | 0.445* | -0.062 | 0.262 | -0.030 | 0.799* | -0.052 | 0.644 | -0.006 | 0.734* | **-0.128** | 0.884 | -0.015 | 0.758 | +0.000 |
| Qwen-3 4B | 12 | 0.635* | -0.170 | 0.408* | -0.100 | 0.224* | -0.068 | 0.628* | **-0.223** | 0.634 | -0.016 | 0.828* | -0.033 | 0.900 | +0.001 | 0.697 | -0.061 |
| Qwen-3 4B | 16 | 0.542* | -0.263 | 0.340* | -0.167 | 0.220* | -0.072 | 0.537* | -0.314 | 0.608* | -0.042 | 0.521* | -0.341 | 0.509* | **-0.390** | 0.625* | -0.134 |

how they relate to model predictions through comprehensiveness and sufficiency. Finally, we assess the confidence calibration of pruned and unpruned models and explore its relation to model accuracy in Sec. 4.3. A summary of our findings is available in Tab. 7

## 4.1 How does Pruning Attention Layers Affect Accuracy?

Fig. 2 plots model accuracy versus attention layer pruning, while Tab. 5 reports the accuracy drops across all datasets when removing 1, 4, 8, 12, and 16 layers from each model, with statistically significant changes indicated by an asterisk (*). Across models and datasets, pruning up to eight layers leads to minimal accuracy loss (under 0.06) for most cases. The only exception is Qwen-3 4B on Rotten Tomatoes, which drops by 0.13 at this pruning depth, but interestingly, its accuracy recovers with further pruning, showing less than a 0.04 drop compared to the original model when 16 layers are removed. For this same pruning range, the observed reductions are statistically significant for most cases on TweetEval and Rotten Tomatoes. In contrast, for ARC Easy, ARC Challenge, OpenBookQA, and BoolQ, accuracy degradation typically becomes statistically significant only as more layers are removed, a trend that holds consistently across all models. When 12 layers are removed, accuracy reductions remain modest: below 0.13 for Mistral 7B and Llama, and up to 0.23 for Qwen, while the number of statistically significant drops increases (see Tab. 5). On SST-2, all models except Llama-2 and Qwen-3 4B tend to exhibit statistically significant drops. Whereas, for RTE, most accuracy reductions remain not significant and typically below 3%. Overall, these results indicate

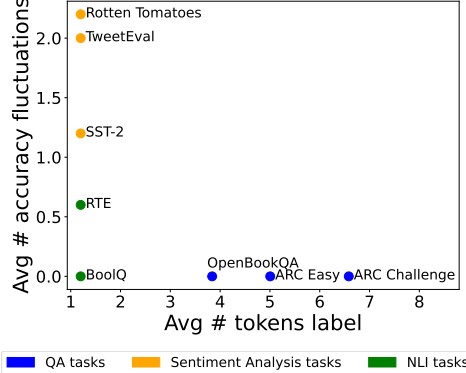

Figure 3: Accuracy fluctuations (y-axis) as pruning increases, versus average number of label tokens (x-axis). Numbers are averaged over the five LLMs.

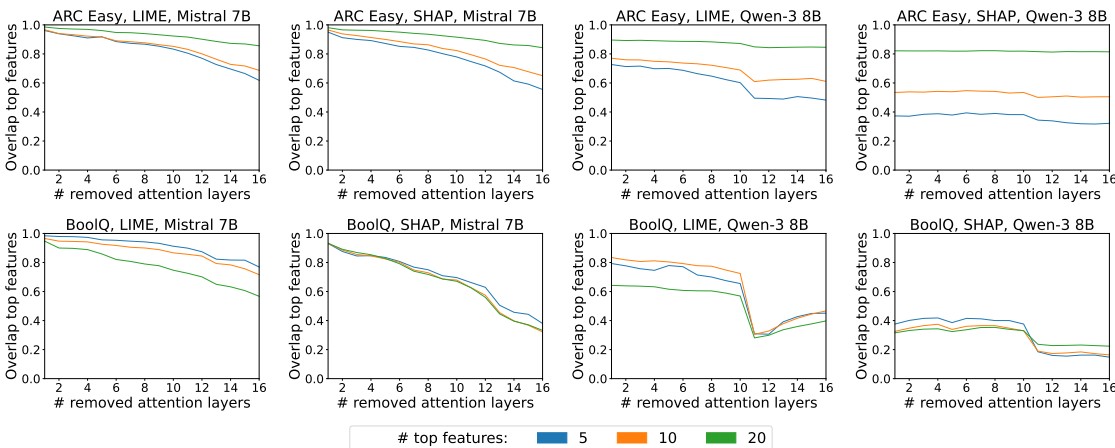

Figure 4: Overlap between the top 5, 10, and 20 LIME and Kernel SHAP features (y-axis) for pruned and unpruned Mistral 7B, as a function of the number of pruned attention layers (x-axis).

that **LLMs can be substantially pruned with only modest reductions in accuracy**. Additional results for the remaining pruning configurations are reported in the Appendix (Tabs. 8-47).

A notable pattern is the consistent accuracy reduction for all models on QA tasks (ARC Easy, ARC Challenge, and OpenBookQA) and BoolQ (the datasets in the first row of Fig. 2), whereas accuracy on sentiment analysis tasks (TweetEval, Rotten Tomatoes, and SST-2) and RTE (datasets in the second row of Fig. 2) is more unstable, with frequent accuracy fluctuations[1] as more attention layers are pruned. We hypothesise that this could be due to label length: Fig. 3 presents the average number of accuracy fluctuations per dataset (averaged over five models) versus the average number of tokens required to represent the labels. Datasets with shorter labels consistently exhibit at least one fluctuation, except for BoolQ, whereas those with longer labels show none. This could be because single-token predictions are more sensitive to pruning-induced noise, while multi-token labels allow such noise to be averaged out across tokens, resulting in more stable accuracy. Fig. 8 of Appendix B shows this in more detail.

## 4.2 How does Pruning Attention Layers Affect Explanation Faithfulness?

Fig. 4 shows the mean overlap between the top-$k$ most important features ($k \in \{5, 10, 20\}$) using LIME and Kernel SHAP for pruned and unpruned Mistral 7B and Qwen-8B on ARC Easy and BoolQ (other models and datasets show overall similar trends; see Figs. 10 and 11 of Appendix B). We compute the overlap as the number of shared features divided by $k$,[2] where 1 means that pruned and unpruned models rely on the same top features, 0 means no shared features. Pruned and unpruned models tend to rely on similar top features. For example, considering Mistral on ARC Easy, the overlap of LIME features remains close to 1 when pruning a single layer, regardless of $k$. Yet, **the similarity between the most important features for pruned and unpruned models decreases as more attention layers are pruned**. This trend generalises across models and tasks, except for Mistral 7B and the NLI tasks (BoolQ and RTE). For Mistral 7B, this trend occurs in 10/16 dataset and attribution combinations; for the NLI datasets, it appears in 14/20 models and attribution combinations (see Figs. 10 and 11 in Appendix B).

We next examine comprehensiveness and sufficiency. Fig. 5 shows the relative reduction in comprehensiveness, sufficiency, and accuracy between pruned and unpruned Mistral 7B and Qwen-3 8B on ARC Challenge and SST-2 as attention layers are progressively removed (see Figs. 20-23 and 12-15 in Appendix B for the remaining relative reduction plots, as well as the absolute drops in comprehensiveness and sufficiency and their standard deviations). Complementing these visual trends, Tab. 6 summarises LIME's Comprehensive-

---

[1]We define a fluctuation as a 0.05 increase/decrease in accuracy that follows a prior decrease/increase. See Appendix A.4 for details on how we count the number of fluctuations.

[2]If the input contains fewer than $k$ features, we divide by the number of features in the input.

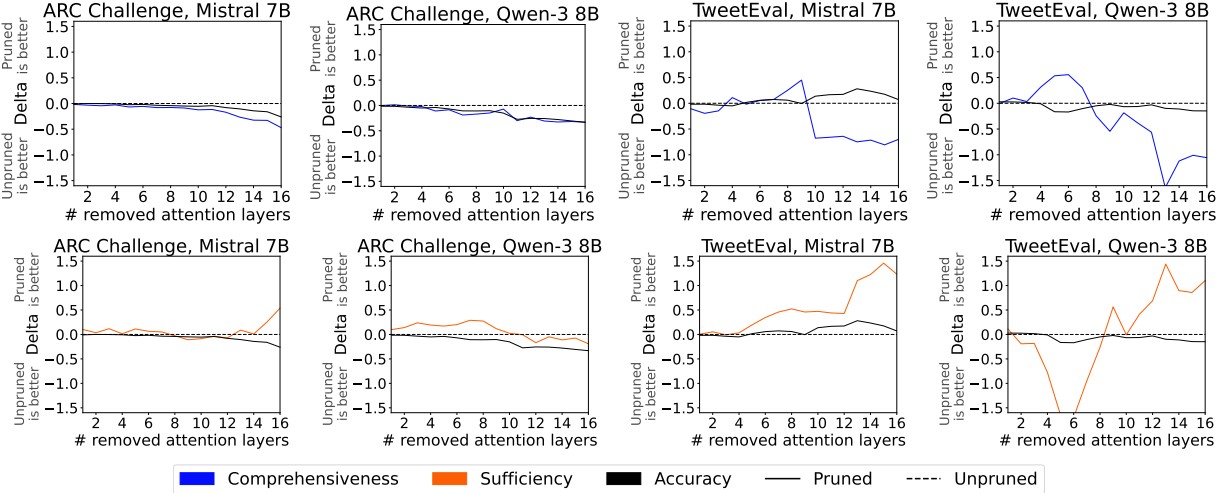

Figure 5: Relative change in LIME's comprehensiveness and sufficiency, and in accuracy (y axis) between pruned and unpruned models across attention layer pruning levels (x axis) for Mistral 7B and Qwen-3 8B. Accuracy and comprehensiveness changes are rescaled so that decreases appear lower in the plots and increases higher.

ness and Sufficiency across all datasets when removing 1, 4, 8, 12, and 16 layers from each model, with statistically significant changes indicated by an asterisk (*). Detailed additional results for other pruning configurations and for Kernel SHAP are provided in the Appendix (Tables 8-47). For both Mistral and Qwen, pruning reduces comprehensiveness from their unpruned counterparts, typically worsening it as more layers are removed. Dataset-wise, comprehensiveness drops are observed across all QA tasks (ARC Easy, ARC Challenge, and OpenBookQA), although changes in both comprehensiveness and sufficiency on these datasets do not exhibit a consistent pattern of statistical significance and vary across models and pruning configurations. In contrast, the majority of settings for BoolQ and RTE, as well as results on TweetEval, Rotten Tomatoes, and SST-2, show statistically significant changes that are largely independent of the model (see Figs. 20 and 22 in Appendix). Sufficiency also tends to decrease with pruning, though less consistently, e.g., for Llama-3, sufficiency increases as more layers are removed. This suggests that pruning attention layers can make the model's reasoning harder to explain, as comprehensiveness indicates the model relies less on the features identified as most important, while sufficiency suggests those same features are still sufficient for the prediction. This means that **pruning attention layers can lead to explanations that are less faithful to the model**. This behaviour could be partly attributed to a decrease in the model's average confidence as more attention layers are pruned. Since both comprehensiveness and sufficiency rely on the model's confidence scores, a drop in overall confidence tends to lower their values as well (see Eq. equation 1 & equation 2).[3] In addition, layers are pruned based on the magnitude of the transformations they apply to their inputs. While individually small, these transformations may still encode information important for explanations, so removing them may contribute to degrading explanation quality.

On datasets with short labels, comprehensiveness and sufficiency scores fluctuate unstably as attention layers are pruned, while they exhibit more stable trends on datasets with longer labels (see Fig. 9). This trend is aligned with accuracy (see Fig. 3), and may be explained by the fact that faithfulness measures compute average attribution scores over label tokens. On short labels (one or two tokens), the measure is sensitive to token-level variation and noise, possibly leading to increased fluctuations. In contrast, longer labels aggregate over more tokens, reducing noise and therefore fluctuation trends. However, as seen for Qwen-3 8B on TweetEval in Fig. 5 (see Figs. 20–23 for the remaining results), these fluctuations in faithfulness can be notably more pronounced than those observed in accuracy, indicating that **pruning attention layers can lead to unstable fluctuations in explanation faithfulness that do not consistently mirror**

---

[3]While we initially considered that this might be due to the different bins used to compute these metrics, we find that each bin has a similar pattern (see Figs. 1619 in Appendix B).

Table 6: Summary of Comprehensiveness and Sufficiency (LIME) across all datasets and models when removing 1, 4, 8, 12 and 16 attention layers. * indicates statistical significance ($p < 0.05$) under the Wilcoxon test. All LIME results and results for Kernel SHAP are available in Appendix, Tabs. 8-47.

| Model | # rem. layers | ARC Easy Comp. | Suff. | ARC Challenge Comp. | Suff. | OpenBookQA Comp. | Suff. | BoolQ Comp. | Suff. | RTE Comp. | Suff. | TweetEval Comp. | Suff. | Rotten Tomatoes Comp. | Suff. | SST-2 Comp. | Suff. |
|---|---|---|---|---|---|---|---|---|---|---|---|---|---|---|---|---|---|
| Llama-2 | (orig) | 0.233 | 0.190 | 0.255 | 0.140 | 0.245 | 0.160 | 0.282 | 0.092 | 0.092 | 0.040 | 0.072 | 0.133 | 0.092 | 0.128 | 0.108 | 0.052 |
| Llama-2 | 1 | 0.234 | 0.189 | 0.253 | 0.140 | 0.247 | 0.162 | 0.267* | 0.105* | 0.098* | 0.023* | 0.038* | 0.048* | 0.046* | 0.038* | 0.200* | 0.097* |
| Llama-2 | 4 | 0.235 | 0.180 | 0.258 | 0.151 | 0.250 | 0.171 | 0.294* | 0.104* | 0.153* | 0.014* | 0.017* | 0.123 | 0.044* | 0.098 | 0.233* | 0.074* |
| Llama-2 | 8 | 0.248* | 0.204* | 0.244 | 0.162* | 0.243 | 0.184* | 0.112* | 0.072 | 0.179* | 0.056* | 0.110* | 0.095* | 0.052* | 0.041* | 0.123* | -0.081* |
| Llama-2 | 12 | 0.215* | 0.229* | 0.216* | 0.167* | 0.221* | 0.185* | 0.076* | 0.048* | 0.217* | 0.187* | 0.052 | -0.035* | -0.067* | -0.110* | -0.012* | -0.110* |
| Llama-2 | 16 | 0.200* | 0.208* | 0.194* | 0.160* | 0.162* | 0.168* | -0.004* | -0.042* | 0.209* | 0.083* | 0.006* | -0.045* | 0.072* | -0.132* | 0.246* | 0.115* |
| Llama-3 | (orig) | 0.200 | 0.126 | 0.222 | 0.092 | 0.237 | 0.139 | 0.384 | 0.127 | 0.141 | 0.143 | 0.016 | 0.166 | 0.106 | 0.093 | 0.105 | 0.097 |
| Llama-3 | 1 | 0.165* | 0.153* | 0.209 | 0.109* | 0.176* | 0.193* | 0.088* | 0.349* | 0.078* | 0.105* | 0.036* | 0.179 | 0.097 | 0.082 | 0.092* | 0.090 |
| Llama-3 | 4 | 0.203 | 0.126 | 0.229 | 0.102 | 0.244 | 0.150* | 0.342* | 0.125 | 0.135 | 0.118* | 0.032* | 0.156 | 0.086 | 0.059* | 0.062* | 0.051* |
| Llama-3 | 8 | 0.197 | 0.121 | 0.239 | 0.113 | 0.235 | 0.163* | 0.336* | 0.116* | 0.167* | 0.117* | 0.217* | 0.191 | 0.111 | 0.104 | 0.123* | 0.107 |
| Llama-3 | 12 | 0.207 | 0.132 | 0.209 | 0.100 | 0.195* | 0.136 | 0.333* | 0.123 | 0.130 | 0.108* | 0.247* | 0.167 | 0.081* | 0.079 | 0.160* | 0.135* |
| Llama-3 | 16 | 0.123* | 0.070* | 0.147* | 0.085 | 0.111* | 0.068* | 0.373 | 0.110* | 0.224* | 0.170* | 0.272* | 0.223* | 0.163* | 0.174* | 0.238* | 0.248* |
| Mistral | (orig) | 0.266 | 0.155 | 0.278 | 0.141 | 0.259 | 0.171 | 0.344 | 0.145 | 0.152 | 0.124 | 0.214 | 0.464 | 0.097 | 0.111 | 0.110 | 0.126 |
| Mistral | 1 | 0.276* | 0.160* | 0.273* | 0.136 | 0.259 | 0.171 | 0.347* | 0.150* | 0.159 | 0.119* | 0.191* | 0.462* | 0.100 | 0.109 | 0.049* | 0.029* |
| Mistral | 4 | 0.278* | 0.160* | 0.271 | 0.144 | 0.260 | 0.176 | 0.334 | 0.154* | 0.173* | 0.117* | 0.238* | 0.448* | 0.079 | 0.033* | 0.042* | 0.006* |
| Mistral | 8 | 0.280* | 0.160 | 0.257* | 0.147 | 0.258 | 0.182* | 0.316* | 0.157* | 0.157 | 0.120* | 0.268* | 0.221* | 0.098 | 0.076* | 0.047* | 0.029* |
| Mistral | 12 | 0.270* | 0.174* | 0.231* | 0.137 | 0.218* | 0.155 | 0.195* | 0.140 | 0.149 | 0.115* | 0.077* | 0.265* | 0.192* | 0.155* | 0.269* | 0.217* |
| Mistral | 16 | 0.213* | 0.155 | 0.148* | 0.091* | 0.139* | 0.115* | -0.003* | 0.053* | 0.134 | 0.116 | 0.064* | -0.108* | 0.026* | 0.069* | 0.095 | 0.005* |
| Qwen-3 8B | (orig) | 0.232 | 0.155 | 0.273 | 0.122 | 0.214 | 0.136 | 0.603 | 0.211 | 0.328 | 0.321 | 0.228 | 0.230 | 0.347 | 0.367 | 0.354 | 0.367 |
| Qwen-3 8B | 1 | 0.238 | 0.146 | 0.273 | 0.110* | 0.213 | 0.119 | 0.596 | 0.151* | 0.306* | 0.316 | 0.227* | 0.204 | 0.350 | 0.376* | 0.355 | 0.367 |
| Qwen-3 8B | 4 | 0.234 | 0.155 | 0.268 | 0.098* | 0.200 | 0.119 | 0.545* | 0.174* | 0.276* | 0.286* | 0.300* | 0.408* | 0.335* | 0.376 | 0.298* | 0.308* |
| Qwen-3 8B | 8 | 0.253 | 0.177 | 0.227* | 0.088* | 0.171* | 0.086* | 0.546* | 0.232* | 0.238* | 0.260* | 0.173* | 0.290* | 0.384* | 0.339* | 0.359 | 0.328* |
| Qwen-3 8B | 12 | 0.211 | 0.181 | 0.210* | 0.142 | 0.134* | 0.093* | 0.410* | 0.387* | 0.291* | 0.318 | 0.100* | 0.072* | 0.323 | 0.290* | 0.310* | 0.274* |
| Qwen-3 8B | 16 | 0.197 | 0.164 | 0.183* | 0.145 | 0.151* | 0.086* | 0.396* | 0.323* | 0.251* | 0.272* | -0.013* | -0.024* | 0.262* | 0.218* | 0.280* | 0.216* |
| Qwen-3 4B | (orig) | 0.236 | 0.146 | 0.273 | 0.126 | 0.206 | 0.126 | 0.476 | 0.248 | 0.317 | 0.308 | 0.180 | 0.155 | 0.437 | 0.332 | 0.391 | 0.350 |
| Qwen-3 4B | 1 | 0.233 | 0.137* | 0.268 | 0.137* | 0.200 | 0.124 | 0.477 | 0.264* | 0.313 | 0.309 | 0.269* | 0.262* | 0.429* | 0.339* | 0.392 | 0.354 |
| Qwen-3 4B | 4 | 0.214 | 0.130 | 0.266 | 0.161* | 0.191* | 0.116 | 0.420* | 0.355* | 0.283* | 0.319* | 0.225* | 0.323* | 0.377* | 0.317 | 0.376* | 0.343 |
| Qwen-3 4B | 8 | 0.208* | 0.136 | 0.239 | 0.131 | 0.163* | 0.120 | 0.420* | 0.251 | 0.302 | 0.384* | 0.083* | 0.101* | 0.198* | 0.289* | 0.329* | 0.298* |
| Qwen-3 4B | 12 | 0.161* | 0.126 | 0.214* | 0.115 | 0.149* | 0.077* | 0.220* | 0.147* | 0.247* | 0.241* | 0.041* | -0.008* | 0.272* | 0.272* | 0.335* | 0.328 |
| Qwen-3 4B | 16 | 0.126* | 0.092* | 0.149* | 0.063* | 0.110* | 0.050* | 0.164* | 0.140* | 0.203* | 0.240* | 0.071* | 0.058* | 0.272* | 0.254* | 0.309* | 0.286* |

**changes in model accuracy**. We hypothesise that layer pruning can reduce reliance on both informative and noisy features; however, this effect is not uniform across layers. Depending on which layers are removed, pruning may differentially affect informative and noisy signals, leading to non-monotonic behaviour and the observed fluctuations that do not necessarily align with accuracy drops.

### 4.3 How does Pruning Attention Layers Affect Confidence Calibration?

Since both comprehensiveness and sufficiency involve confidence, faithfulness may be influenced by model confidence. This raises a question: how well does confidence align with accuracy? To explore this, we next examine confidence calibration, which measures the alignment between predicted confidence and accuracy.

The first row of Fig. 6 shows the ECE of pruned Mistral 7B and Llama-3 8B on ARC Easy and TweetEval (the rest of the plots are in Fig. 34 of the appendix). As more attention layers are removed, misalignment between model confidence and accuracy increases for both Mistral and Llama, indicating that **pruning attention layers can worsen the confidence calibration of the model**. We hypothesise this is because layer pruning perturbs internal representations and logits, therefore models' confidence too. Calibration depends on alignment between confidence and accuracy, so even small logits shifts can degrade calibration by affecting confidence but not enough for a prediction flip. In addition, pruning attention layers may make the model over or under confident on individual samples, which further contributes to deterioration in confidence calibration. Qwen is an exception to this: its calibration error improves on sentiment analysis tasks (TweetEval, Rotten Tomatoes, and SST-2). This may be because pruning reduces both the model's confidence and accuracy. Since ECE measures how well confidence matches accuracy, by grouping predictions into confidence bins and comparing the average confidence and accuracy in each bin, lower confidence can cause predictions to be clustered into fewer bins. At the same time, as accuracy slightly drops (Fig. 2), the gap between confidence and accuracy within each bin may also shrink, potentially lowering ECE, despite worse overall accuracy.

Lastly, we examine the relationship between confidence calibration and accuracy. The second row of Fig. 6 shows the relative changes in ECE and accuracy of pruned Mistral 7B and Llama-3 8B on ARC Easy and TweetEval (the rest of the plots are in Fig. 35 in the Appendix). We see that ECE can drop disproportionally

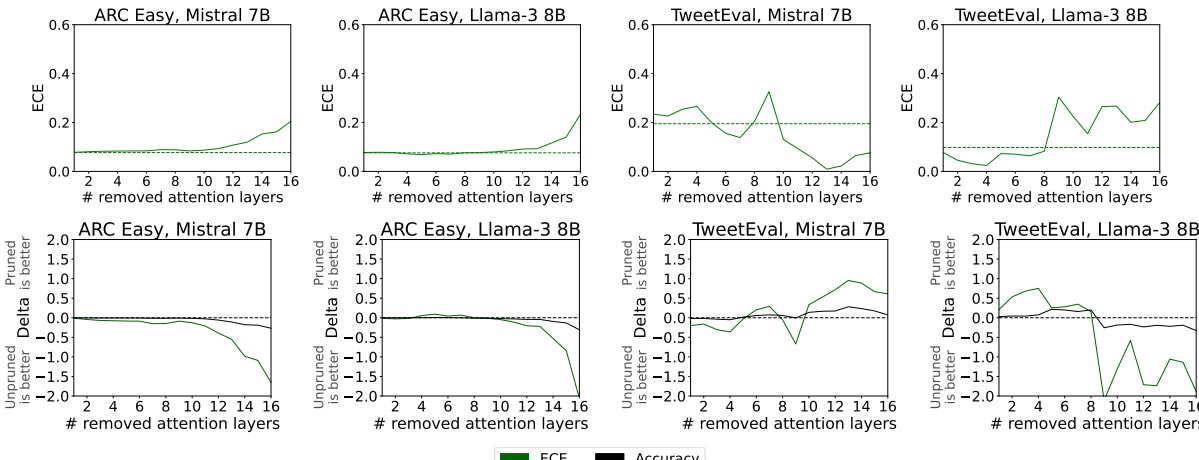

Figure 6: First row: ECE (y axis) for Mistral 7B and Llama-3 8B for different levels of attention layer pruning (x axis). Second row: Relative change in ECE (green line) and accuracy (black line) (y axis) for Mistral 7B and Llama-3 8B for different levels of attention layer pruning (x axis). Accuracy changes are rescaled so that decreases appear lower in the plots and increases higher.

compared to accuracy. In addition, calibration error scores fluctuate unstably on datasets with short labels as attention layers are pruned, while having more stable trends on datasets with longer average label lengths. As this trend is similar to accuracy (see Fig. 3), we show the results in the appendix (see Fig. 8). We believe this trend is partially explained by the fact that for single-token or short-token predictions, model confidence depends heavily on a very small number of tokens, since log-likelihoods are computed over only a limited subset of the sequence. As a result, pruning-induced noise or information loss has a disproportionately large effect, leading to greater fluctuations in predicted confidence for short texts and, consequently, more unstable ECE estimates. As seen for Llama-3 8B on TweetEval in Fig. 6, these fluctuations in calibration error can be notably more pronounced than those observed in accuracy, indicating that **pruning attention layers can lead to unstable fluctuations of model calibration that do not consistently mirror changes in model accuracy**. As previously mentioned, calibration depends on the alignment between confidence and accuracy, and even small logit perturbations can change confidence without affecting predictions, but changing the calibration scores. Additionally, pruning may remove attention heads that are not critical for accuracy but are important for confidence estimation. As a result, we believe that the model can maintain similar accuracy while becoming overconfident or underconfident, leading to unstable and non-monotonic changes in calibration error.

### 4.4 How does Pruning Attention Layers Affect Plausibility?

The first, second, and third column of Fig. 7 report average F1, precision, and recall respectively for the top-$K$ LIME features (with $K \in \{10, 20, 30, 40, 50\}\%$) identified on Mistral, against human relevance annotations (see Figs. 24-29 for the remaining results). Across measures, plausibility shows a broadly stable behaviour under pruning, likely because pruning mostly preserves features ranking, and plausibility measures compare these rankings directly against human annotations. In contrast, explanation faithfulness varies substantially.

F1 remains comparatively stable, with a small decrease as more layers are pruned. This effect is more pronounced on the top 10% features and is attenuated as $K$ increases, suggesting that pruning primarily perturbs the top ranking attributions while leaving the overall set of important features similar when more features are considered. Precision is mostly invariant across $K$, aside from isolated cases (e.g., Llama-3 8B on ARC Easy), indicating that the proportion of selected features aligning with human annotations is robust to both pruning and the choice of $K$. In contrast, recall shows clearly separated curves for different $K$, increasing with larger feature sets. For most combinations of models and datasets, both precision and recall exhibit a small drop as layers are removed.

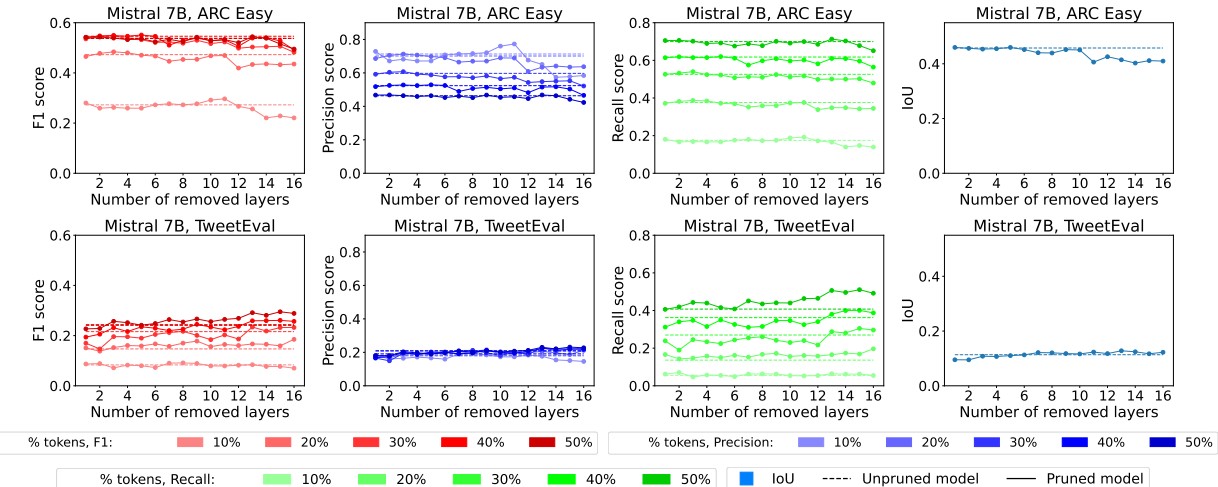

Figure 7: **Columns (left to right):** average F1, Precision, Recall, and IoU between LIME attributions for Mistral 7B and human annotations (y axis), as a function of pruned attention layers (x axis). Dotted lines denote the unpruned model. For F1, Precision, and Recall, colours indicate the proportion of top features considered ($K$). Rows correspond to datasets: ARC Easy (top) and TweetEval (bottom).

The fourth column of Fig. 7 shows average Intersection-over-Union (similar trends hold for AUPRC and were reported in Appendix for brevity; see Figs. 30-33). Consistent with the F1 analysis, both measures remain broadly stable across pruning levels, with modest drops as more layers are removed. Overall, this shows that explanations' plausibility is resilient to pruning, though still gradually degraded. In contrast, prior results show that faithfulness is more strongly affected, suggesting that while pruning preserves the ranking of salient features, and therefore the agreement with human annotations, it can alter the model's actual reliance on them. As a result, explanations may remain plausible to humans while becoming less faithful to the model's decision process.

## 4.5 Recommendations for Faithfulness and Calibration Aware Pruning

We present recommendations for designing layer pruning strategies that account for explanation faithfulness and confidence calibration, motivated by our finding that attention layer pruning can degrade both properties independently of accuracy.

Preserving the faithfulness of explanations and confidence calibration ensures that pruned models remain not only accurate but also reliable in their explanations and predictive confidence, increasing their overall trustworthiness. Our results show that both properties can deteriorate without corresponding drops in accuracy, making accuracy alone an insufficient proxy for model quality. We therefore recommend monitoring faithfulness and calibration alongside accuracy throughout the pruning process and using them as stopping criteria. Pruning strategies can be improved by incorporating these properties directly into layer importance scoring, or by adopting iterative approaches that revert pruning steps when they compromise such properties. More broadly, pruning should be treated as a multi-objective optimisation problem, balancing accuracy, explanation faithfulness, and confidence calibration while targeting a desired sparsity level. Finally, post-hoc correction methods, such as lightweight adapters (e.g., LoRA), can be applied after pruning to recover performance while explicitly optimising faithfulness and confidence calibration.

## 5 Conclusions and Limitations

We studied the impact of attention layer removal on accuracy, explanation faithfulness, plausibility, and confidence calibration for LLMs. We tested five LLMs and eight datasets, and found that LLMs can be substantially pruned, removing up to a third of their attention layers with only modest reductions in accu-

Table 7: Our findings and number of times they hold per model and task.[5] Findings that hold more than 75% of the times are bolded.

| Finding | Models | | | | | datasets | | |
|---|---|---|---|---|---|---|---|---|
| | Mistral 7B | Llama-2 7B | Llama-3 8B | Qwen-3 8B | Qwen-3 4B | QA | Sentiment Analysis | NLI |
| **8/8** | **8/8** | **8/8** | **8/8** | **8/8** | **15/15** | **15/15** | **10/10** | |
| The overlap between the most important features between pruned and unpruned LLMs drops as pruning increases (Sec. 4.2, Figs. 10 and 11) | 10/16 | **16/16** | **12/16** | **15/16** | **15/16** | **24/30** | **24/30** | 14/20 |
| Pruning attention layers leads to explanations that are less faithful to the model, hence harder to interpret (Sec. 4.2, Fig. 20 to 23) | **16/16** | 11/16 | 8/16 | **16/16** | **16/16** | **30/30** | 21/30 | **16/20** |
| Pruning attention layers makes explanation faithfulness unstable in ways not mirrored by changes in accuracy (Sec. 4.2, Fig. 20 to 23) | 9/16 | 10/16 | 9/16 | 2/16 | 5/16 | 0/30 | **24/30** | 9/20 |
| Pruning attention layers can worsen the confidence calibration of the model (Sec. 4.3, Fig. 34) | **7/8** | **8/8** | **7/8** | 4/8 | 5/8 | **14/15** | 10/15 | 7/10 |
| Pruning attention layers can lead to unstable model calibration in ways not mirrored by changes in accuracy. (Sec. 4.3, Fig. 35) | **6/8** | **6/8** | **6/8** | **6/8** | **6/8** | 5/15 | **15/15** | **10/10** |

racy. For all LLMs but Qwen, the accuracy drop remained relatively small up to 12 removed layers. Despite stable accuracy, we observed that explanation faithfulness degrades with layer pruning. Comprehensiveness and sufficiency scores declined as more layers were removed, indicating that pruned models rely less on the most important input features, yet still consider them sufficient for the prediction. This behaviour may be partly explained by a reduction in the model's average confidence, which affects both comprehensiveness and sufficiency. We also found that pruning can affect explanation faithfulness independently of accuracy, often changing disproportionately and unstably. Our plausibility analysis reveals that, in contrast to faithfulness, which can vary substantially under pruning, plausibility remains comparatively stable, with only mild degradation as layers are removed. We further analysed the alignment between model confidence and accuracy through Expected Calibration Error (ECE), finding that pruning can harm calibration, even when accuracy remains stable. Like explanation faithfulness, the model's error calibration after pruning can change disproportionately and unstably relatively to accuracy, revealing a misalignment between confidence and accuracy that is so far undetected in literature.

Alternative pruning strategies, such as removing MLP layers or applying different pruning criteria like SparseGPT or Wanda, may yield different efficiency-interpretability trade-offs. These methods prune individual weights rather than entire layers and could affect interpretability in unique ways. To our knowledge, no work has compared pruning strategies along this axis, making it a valuable direction for future research.

In addition, further research can explore a broader range of feature attribution methods and faithfulness measures, especially given their low agreement on attribution quality (Barr et al., 2023). Evaluating pruned models across diverse downstream tasks and further exploring plausibility of pruned models using annotated datasets for plausibility, could deepen our understanding of trade-offs between efficiency, faithfulness, plausibility, and calibration. Furthermore, prompt design remains an open question. Our experiments use a single zero-shot prompt, leaving it unclear how sensitive the observed effects are to prompt variations. Investigating whether specific prompting techniques can mitigate degradation in explanation faithfulness and calibration could offer valuable insights, though such analysis would be computationally intensive.

This study investigated the effects of attention layer removal pruning. While this approach provides valuable insights, we acknowledge that it represents only one strategy within a broader class of pruning techniques. Future work should explore the effect of additional layer removal strategies on LLMs and a wider range of pruning strategies, for instance, common pruning methods such as SparseGPT (Frantar & Alistarh, 2023) and Wanda (Sun et al., 2024). Additionally, evaluating pruned models using a broader set of feature attribution methods would further strengthen the generalizability of the conclusions to explainers beyond surrogate-based methods. This work focused on pruned LLMs and, therefore, cannot be applied to black-box models, as it requires access to model weights. Our study is limited to decoder-based LLMs and trends may vary on other architectures. Additionally, our setup required up to 40GB of GPU memory, which may limit

---

[5]We treat BoolQ as an NLI task since it uses text pairs and a prompt structure similar to NLI tasks

the reproducibility of similar studies on hardware with lower memory availability. Finally, due to the high computational cost of faithfulness evaluation, we limited our analysis to 200 randomly sampled instances per dataset. Even under this constraint, our experiments exceeded 6000 GPU hours.

## Broader Impact and Ethics Statement

As LLMs grow more complex, layer pruning is an effective and simple strategy to democratise access to state-of-the-art models. While layer pruning can produce efficient models with minimal accuracy loss, our findings show that it may come at the cost of explanation faithfulness and confidence calibration, two critical properties for trustworthy AI. These trade-offs should be considered when deploying pruned models in the real-world, especially in high-stakes settings where explainability may be as important as accuracy. For example, in legal summarisation, feature attributions can help highlight which parts of a document contributed most to the summary, supporting transparency and trust. However, a pruned LLM may still produce a fluent summary while attributions point towards irrelevant or misleading parts of the input. This faithfulness degradation can mislead legal professionals and ultimately undermine the reliability of the summarisation system. In healthcare, poor calibration may cause models to appear overconfident. This could reduce trust if clinicians notice the misalignment, or lead to relying too much on the LLMs, potentially resulting in unsafe clinical decisions once the model produces a wrong diagnosis. These examples highlight the societal risks of evaluating pruned models solely based on accuracy. Trustworthiness must be assessed holistically, considering not just accuracy but also how models reason and how well their confidence aligns with their actual correctness.

While our findings expose certain limitations of pruning LLMs, we emphasise that pruning remains highly relevant, particularly for low-resource environments and for enabling more sustainable deployment of LLMs. Our goal is not to discourage their use, but rather to encourage their responsible and informed application.

To the best of our knowledge, we have cited all relevant prior work and adhered to the JMLR ethics guidelines. We recognise that studies involving large-scale model evaluations can be computationally intensive, potentially limiting reproducibility. To mitigate this and promote transparency, we share our code and relevant implementation details to facilitate replication of our results.

## Acknowledgments

This work was funded by the Algorithms, Data & Democracy project (Villum & Velux foundations). We also thank Theresia Veronika Rampisela for her valuable feedback on an earlier version of the work.

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

# A   Methodology

We provide additional details on the datasets, models, and attribution methods used in our study.

## A.1   Datasets

We evaluate our models on eight NLP datasets:

**ARC Easy and ARC Challenge.** ARC Easy and ARC Challenge (Clark et al., 2018) are multiple-choice science question datasets, each with four answer options per question. Both are available under a CC BY-SA 4.0 license.

**OpenBookQA** OpenBookQA (Mihaylov et al., 2018) is a multiple-choice question answering dataset that requires multi-step reasoning and common knowledge. Each question has four answer choices. The dataset is available under Apache License 2.0.

**BoolQ.** BoolQ (Clark et al., 2019) consists of yes/no questions paired with contextual passages. It covers a wide range of topics and is available under a CC BY-SA 3.0 license.

**RTE.** RTE (Wang et al., 2018) is a natural language inference dataset derived from RTE1-RTE3, and RTE5 (Dagan et al., 2006; Haim et al., 2006; Giampiccolo et al., 2007; Bentivogli et al., 2009). Each instance is given by a pair of sentences and the task is to predict entailment. Each instance consists of a sentence pair, and the task is to predict entailment. While the GLUE benchmark does not specify a license for RTE, the RTE1-RTE3, and RTE5 datasets were made available for non-commercial research use. To the best of our knowledge, the original licenses are not publicly accessible.

**TweetEval.** TweetEval (Barbieri et al., 2020), (sentiment partition) (Rosenthal et al., 2017) is a Twitter-based sentiment classification dataset with tweets labelled as positive, negative, or neutral. It is available under a CC BY-SA 3.0 Unported license.

**Rotten Tomatoes.** Rotten Tomatoes (Pang & Lee, 2005) is a movie review sentiment dataset containing short reviews labelled as positive or negative. The dataset is distributed under an Apache License 2.0 through public dataset wrappers, though, to the best of our knowledge, the original data does not specify a license.

**SST-2.** SST-2 (Socher et al., 2013) is a binary sentiment classification dataset derived from the Stanford Sentiment Treebank. Similar to Rotten Tomatoes, the dataset is distributed under an Apache License 2.0 through public dataset wrappers, though, to the best of our knowledge, the original data does not specify a license.

### A.2 Prompt templates

We report the prompt templates used in our study:

**ARC Easy and ARC Challenge.** We used the default prompt template provided by LM Evaluation Harness (Gao et al., 2024) to prompt the LLMs on this task. Precisely, the template is:

```
Question: [question]
Answer:
```

**OpenBookQA.** We used the default prompt template provided by LM Evaluation Harness (Gao et al., 2024) to prompt the LLMs on this task. Precisely, the template is:

```
[question]
```

**BoolQ.** We used the default prompt template provided by LM Evaluation Harness (Gao et al., 2024) to prompt the LLMs on this task. Precisely, the template is:

```
[passage]
Question: [question]?
Answer:
```

**RTE.** We used the default prompt template provided by LM Evaluation Harness (Gao et al., 2024) to prompt the LLMs on this task. Precisely, the template is:

```
[sentence1]
Question: [sentence2] True or False?
Answer:
```

**TweetEval.** We used the second prompt template provided by promptsource (Bach et al., 2022) for the sentiment set of TweetEval to prompt the LLMs on this task. Precisely, the template is:

```
What is the sentiment of the tweet?

[tweet]

Possible choices: positive, neutral, or negative
Answer:
```

**Rotten tomatoes.** We used the third prompt template provided by promptsource (Bach et al., 2022) to prompt the LLMs on this task. Precisely, the template is:

```
[review]
Is this review positive or negative?
Answer:
```

**SST-2.** We used the default prompt template provided by LM Evaluation Harness (Gao et al., 2024) to prompt the LLMs on this task. Precisely, the template is:

```
[sentence]
Question: Is this sentence positive or negative?
Answer:
```

### A.3 Language Models

We use five decoder-based LLMs in our experiments:

**Mistral 7B.** Mistral 7B (Jiang et al., 2023), released under the Apache License 2.0

**Llama-2 7B.** Llama-2 7B (Touvron et al., 2023), released under the Llama 2 community license.

**Llama-3 8B.** Llama-3 8B (Team, 2024), released under the Llama 3 community license.

**Qwen-3 8B.** Qwen-3 8B (Yang et al., 2025), released under the Apache License 2.0

**Qwen-3 4B.** Qwen-3 4B (Yang et al., 2025), released under the Apache License 2.0

## A.4 Accuracy, Faithfulness, and Confidence Calibration Fluctuations Across Pruning Thresholds

We compute accuracy, ECE, sufficiency, and comprehensiveness fluctuations (see Figs. 8 and 9). For faithfulness measures, scores are averaged over three random seeds used for attributions. We evaluate fluctuations per model by tracking score trends across pruning levels, from the unpruned model up to removing 16 attention layers. We first determine the initial trend (increasing or decreasing) as we prune the first layer; a fluctuation is counted when the score changes by at least 5% in the opposite direction of the current trend, at which point the trend is updated, and tracking continues. We report the average number of fluctuations across models (y-axis). To account for tokenisation differences, the x-axis shows the average number of tokens per label across models on the dataset.

In summary, this analysis shows how stable model performance and faithfulness measures are under progressive pruning, by quantifying how often their trends change and relating this instability to the number of tokens used to represent labels.

## A.5 Plausibility Study

We include here the instructions that were given to the annotators. The instructions are adapted from Hayati et al. (2021).

```
Instructions
Thank you for participating in this study! Please read the instructions carefully.
Warning: this study may contain offensive words which may be upsetting.
You will be given 64 sentence-labels pairs.
For each sentence, select and bold the word(s) that make you associate the
sentence with a specific label.

Example:

Sentence:
Which of these is a place where a human might live?

Labels:
igloo, cloud, mars, the moon

Annotation:
Which of these is a place where a human might live?
```

## A.6 Summary of Hyperparameters and Experimental Settings

- **Prompting.** All models were evaluated in a zero-shot setting using greedy decoding. We used the standard LM-eval-harness (Gao et al., 2024) prompt templates for each dataset, with the exception of TweetEval and Rotten Tomatoes, for which we selected prompt templates from promptsource (Bach et al., 2022).

- **Layer Pruning.** We followed He et al. (2026) and pruned attention layers iteratively based on their block influence scores (Men et al., 2024), computed as the cosine similarity between each layer's input and output representations. Scores were estimated using **8 sequences of 2048 tokens** sampled from the C4 dataset.

- **Feature attribution.** We compute feature attributions using LIME (Ribeiro et al., 2016) and Kernel SHAP (Lundberg & Lee, 2017). Following Edin et al. (2024), we set the number of samples used to train the surrogate models to three times the number of input features.

- **Faithfulness Evaluation.** Due to computational constraints, we evaluated explanation faithfulness on **200 randomly sampled instances per dataset**. Feature attributions were computed using **three random seeds**. We used 10%, 20%, 30%, 40%, and 50% as thresholds for *comprehensiveness* and sufficiency.

- **Confidence calibration.** We computed Expected Calibration Error (ECE) using **10 equidistant bins**, following Naeini et al. (2015).

- **Statistical tests** We use McNemar's test for accuracy and the Wilcoxon test for faithfulness measures, as the scores are not normally distributed.

- **Hardware Requirements.** Our experiments required up to **40 GB of GPU memory**, and the full evaluation, including faithfulness, exceeded **6000 GPU hours** utilising a combination of NVIDIA L40S, A40 and A100 GPUs.

# B   Results

We report here the complete set of individual results for all analyses presented in the paper.

Figs. 8 and 9 show the number of fluctuations observed in accuracy and ECE, and in comprehensiveness and sufficiency, respectively.

Figs. 10 and 11 present the overlap among the top 5, 10, and 20 features identified by LIME and Kernel SHAP, respectively, for pruned and unpruned models, as a function of the number of pruned attention layers.

Figs. 12-15 report comprehensiveness, sufficiency, and accuracy for pruned models across all datasets, using both LIME and Kernel SHAP, at varying levels of attention layer removal.

In Fig. 5, we found that pruning attention layers can affect both their comprehensiveness and sufficiency. To determine whether this trend is due to specific bins, we plot sufficiency and comprehensiveness for each bin, i.e., the bins containing 10%, 20%, 30%, 40%, and 50% of the tokens. Figs. 16, 17, 18, and 19 demonstrate that the individual bins used to compute sufficiency and comprehensiveness exhibit the same behaviour as the aggregated measures. Therefore, no single bin disproportionately affects the aggregated measurements of sufficiency and comprehensiveness.

Figs. 20-23 illustrate the relative reductions in comprehensiveness, sufficiency, and accuracy between pruned and unpruned models across different levels of attention layer removal.

Figs. 24-29 report average F1, precision, and recall for the top-$K$ features (with $K \in 10, 20, 30, 40, 50\%$) against human relevance annotations.

Figs. 30-33 further report average Intersection-over-Union (IoU) and Area Under the Precision-Recall Curve (AUPRC).

Fig. 34 presents the Expected Calibration Error (ECE) for all models at varying degrees of attention layer removal. Fig. 35 shows the corresponding relative reductions in ECE and accuracy.

Finally, Tabs. 8-47 provide the complete individual results for accuracy, ECE, and the comprehensiveness and sufficiency scores computed using LIME and Kernel SHAP across all models and datasets.

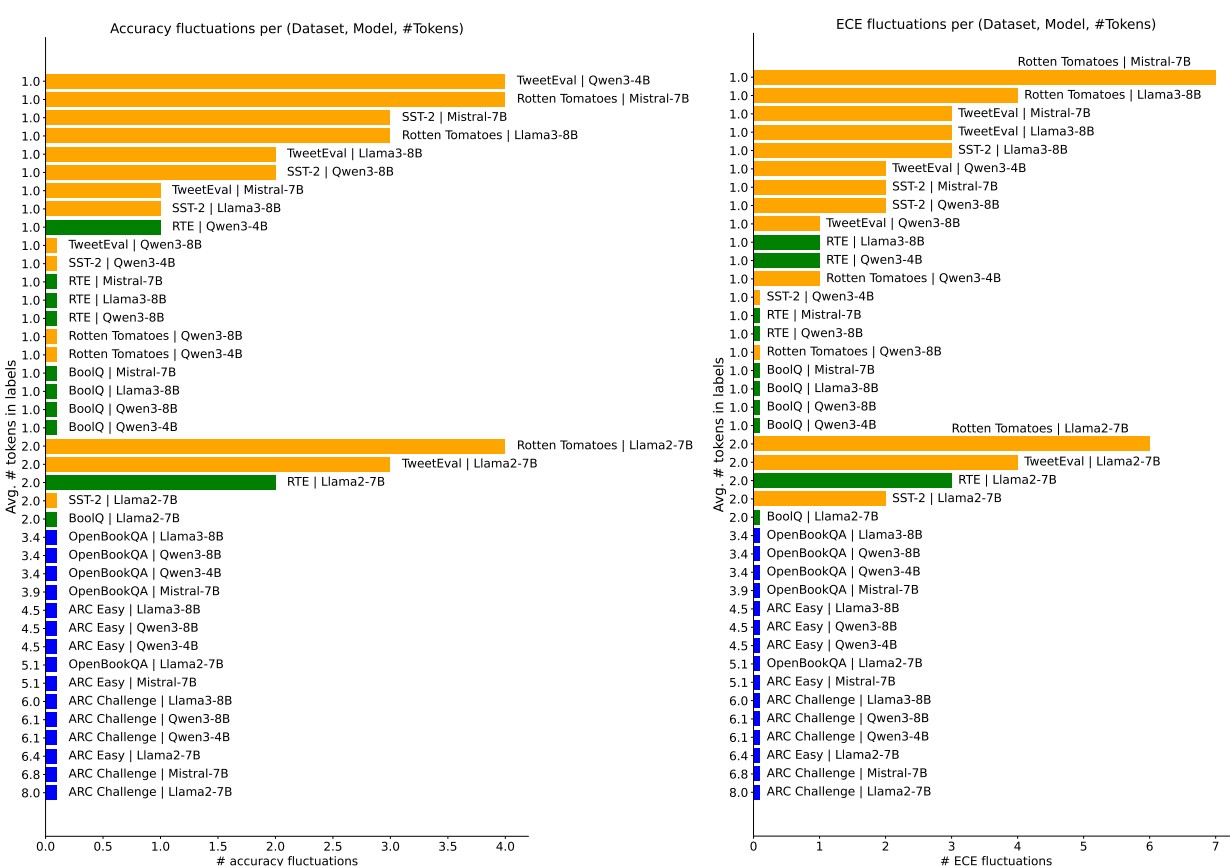

Figure 8: Number of fluctuations (x-axis) in accuracy (first barplot) and ECE (second barplot) for combinations of datasets and models with different average number of label tokens (y-axis).

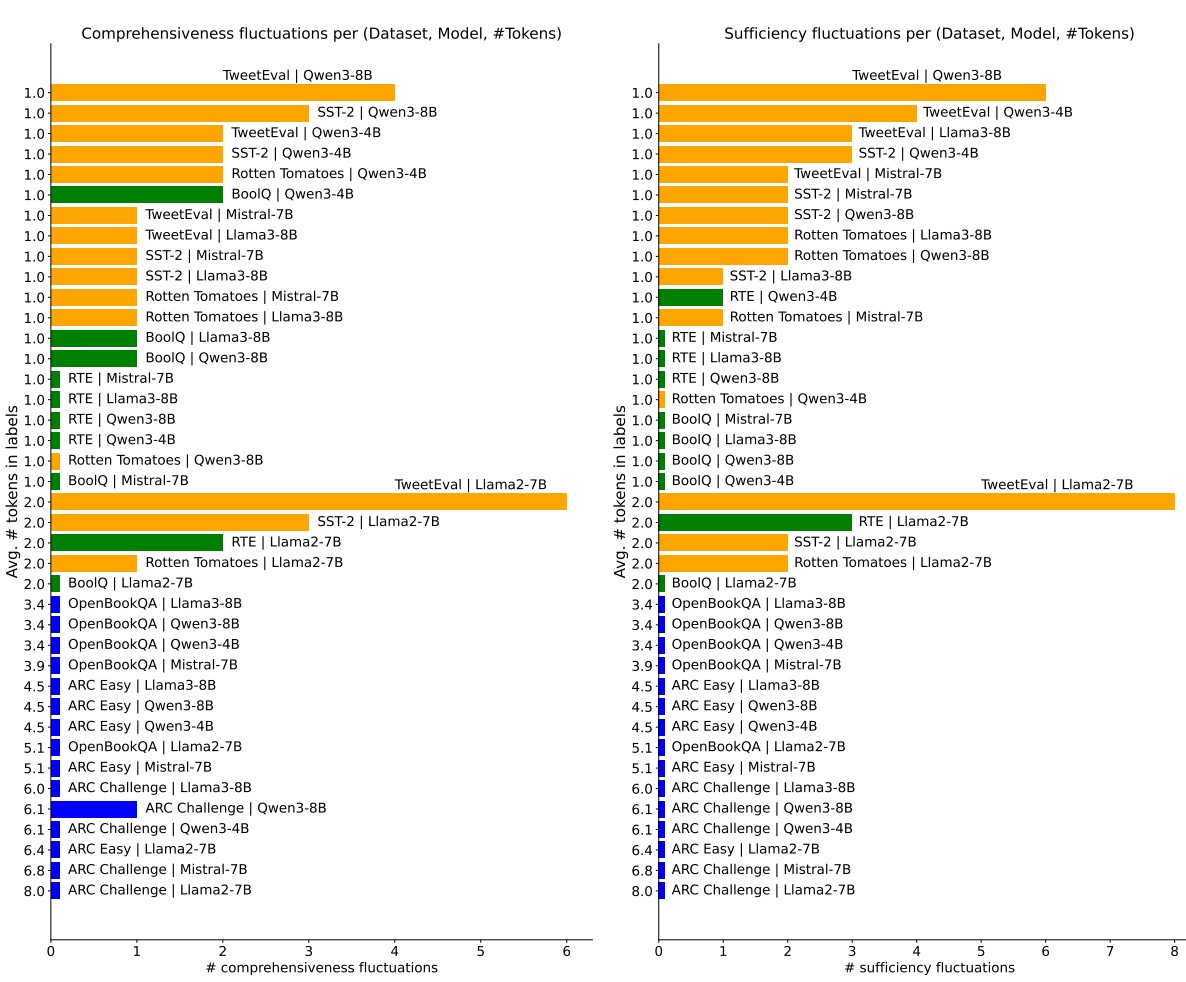

Figure 9: Number of fluctuations (x-axis) in comprehensiveness (first barplot) and sufficiency (second barplot) for combinations of datasets and models with different average number of label tokens (y-axis).

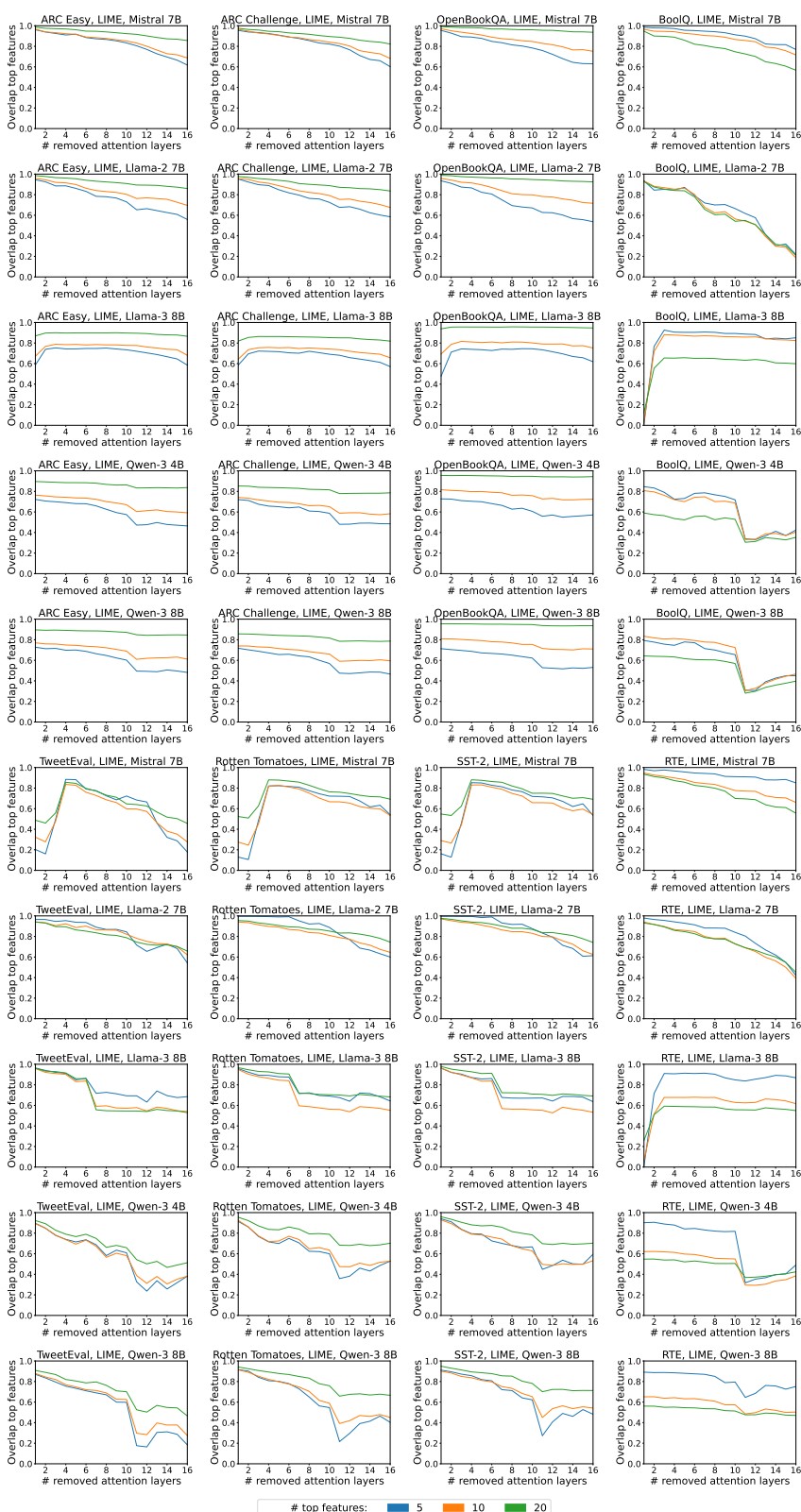

Figure 10: Overlap between the top 5, 10, and 20 features identified by LIME (y-axis) for pruned and unpruned models, as a function of the number of pruned attention layers (x-axis).

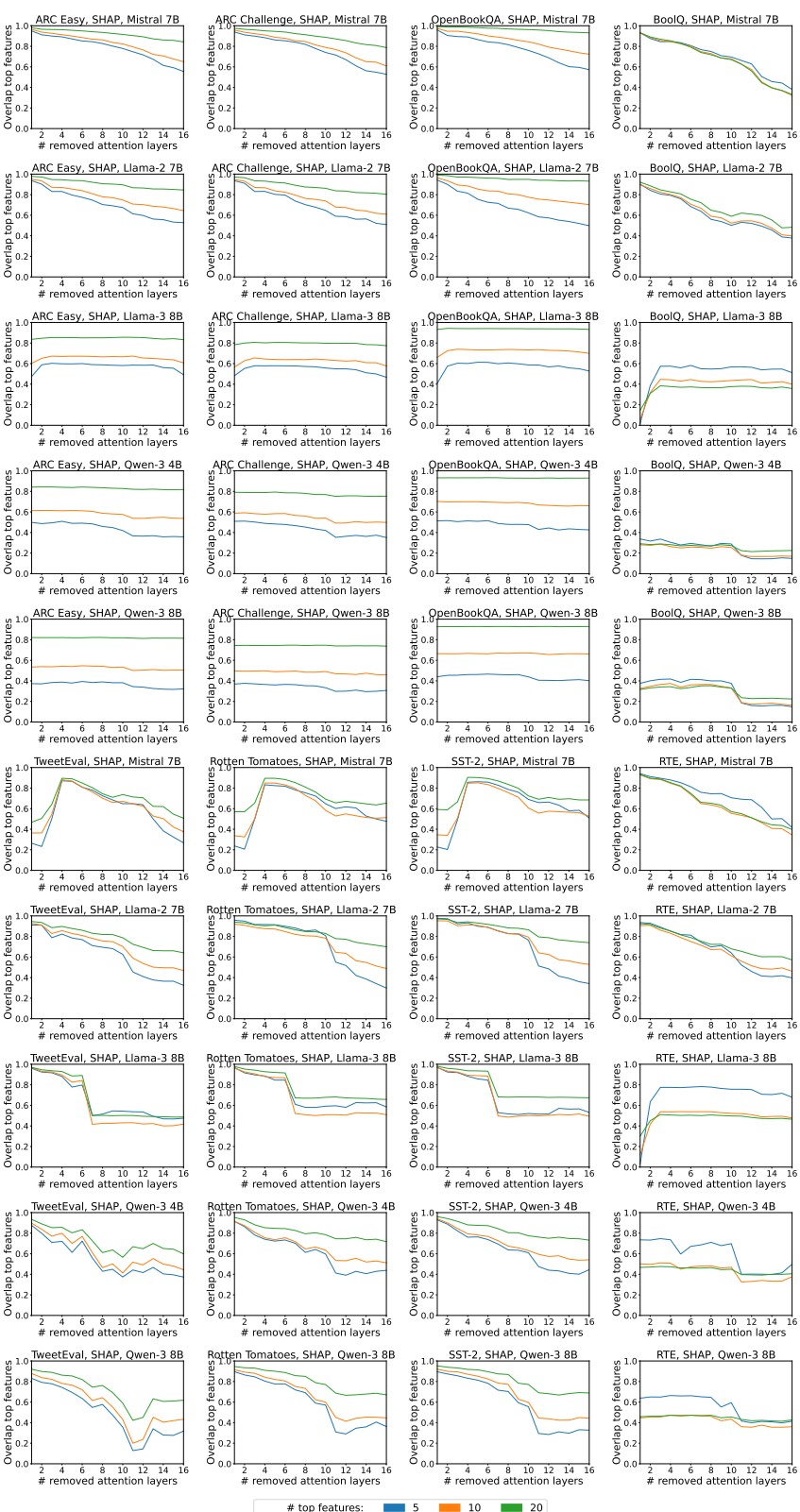

Figure 11: Overlap between the top 5, 10, and 20 features identified by Kernel SHAP (y-axis) for pruned and unpruned models, as a function of the number of pruned attention layers (x-axis).

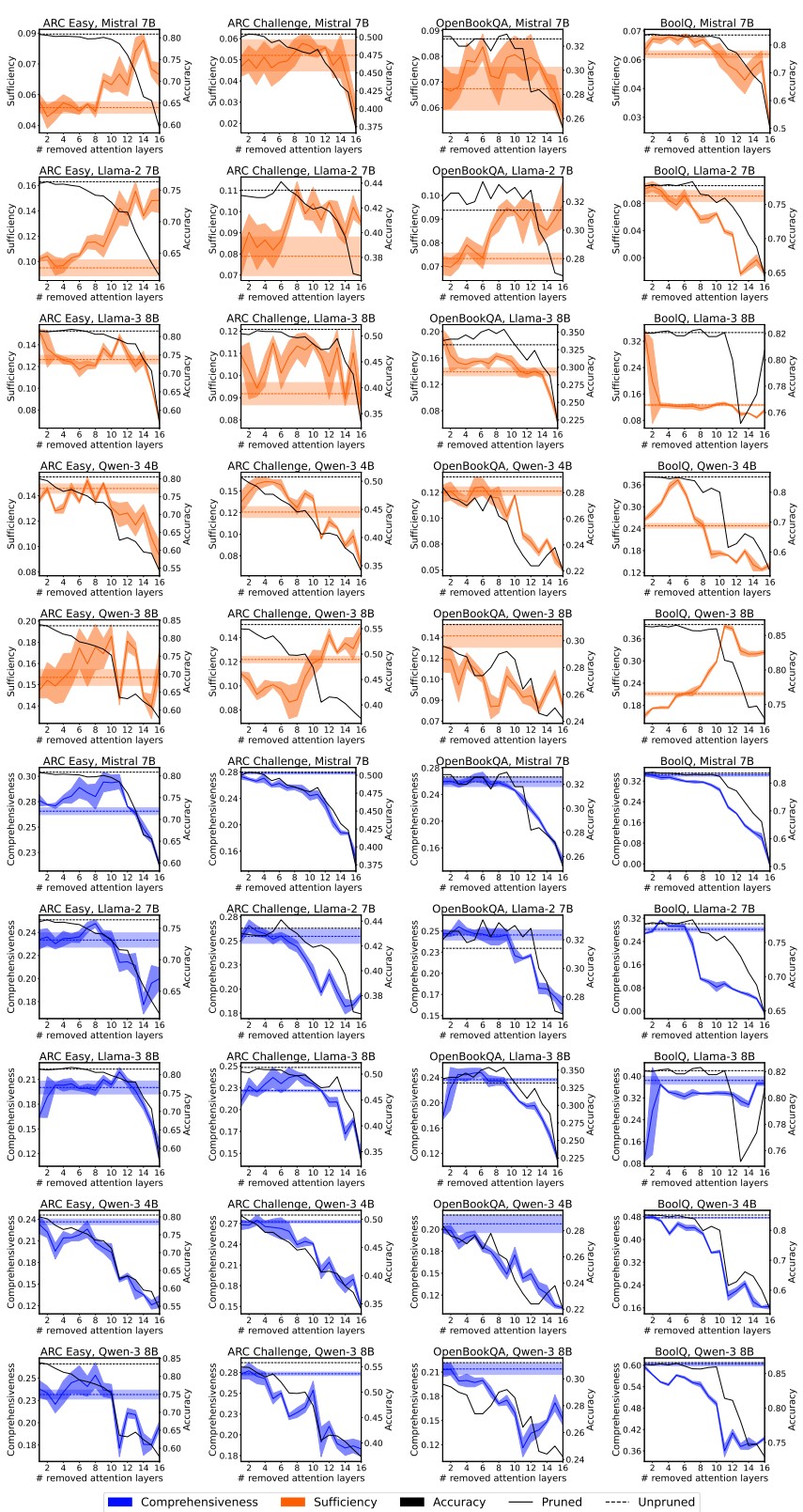

Figure 12: Comprehensiveness (↑), sufficiency (↓), and accuracy (↑) of models (y axis) on QA tasks, using LIME, at varying amounts of removed attention layers (x axis).

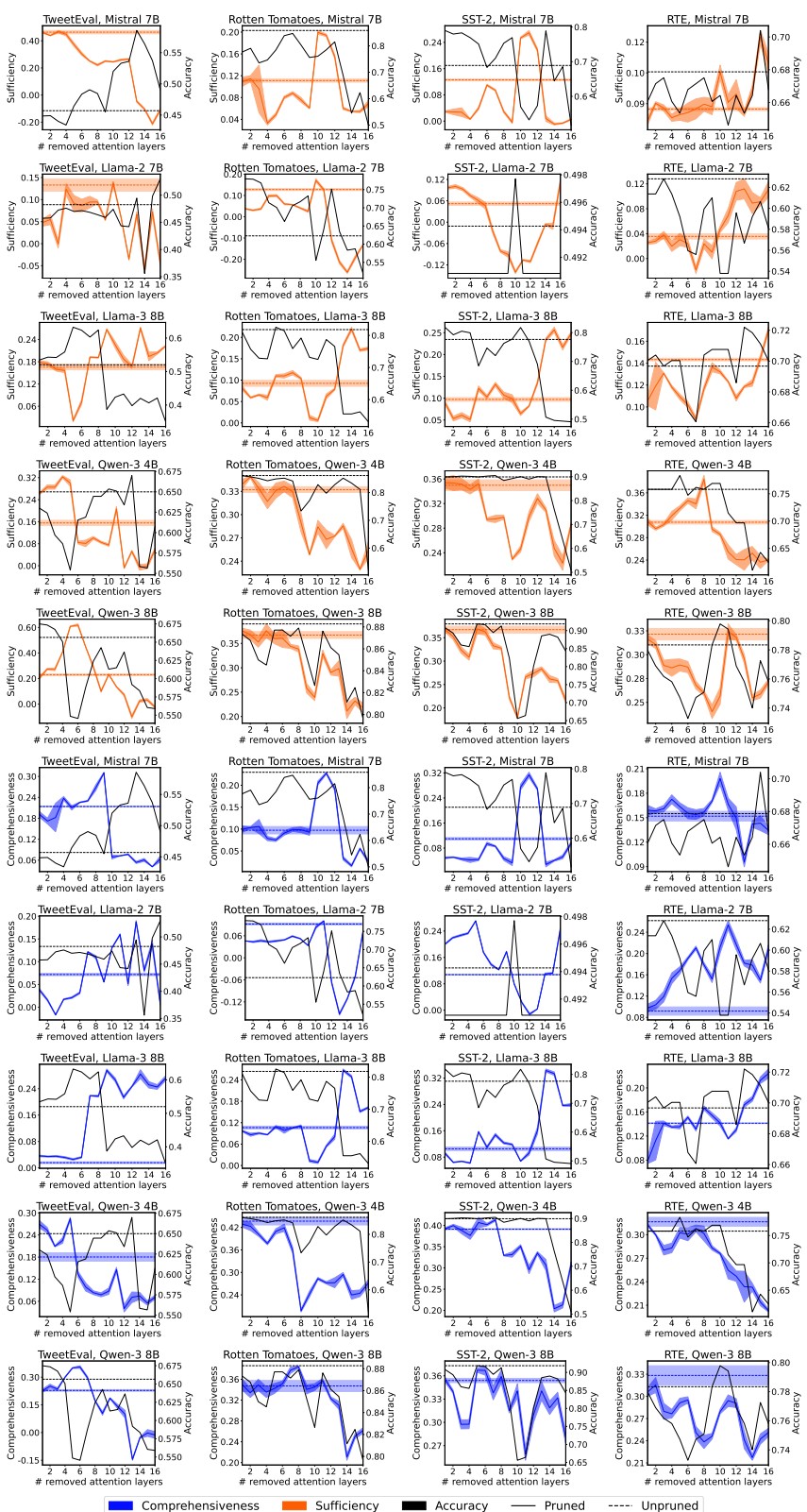

Figure 13: Comprehensiveness (↑), sufficiency (↓), and accuracy (↑) of models (y axis) on sentiment analysis tasks and RTE, using LIME, at varying amounts of removed attention layers (x axis).

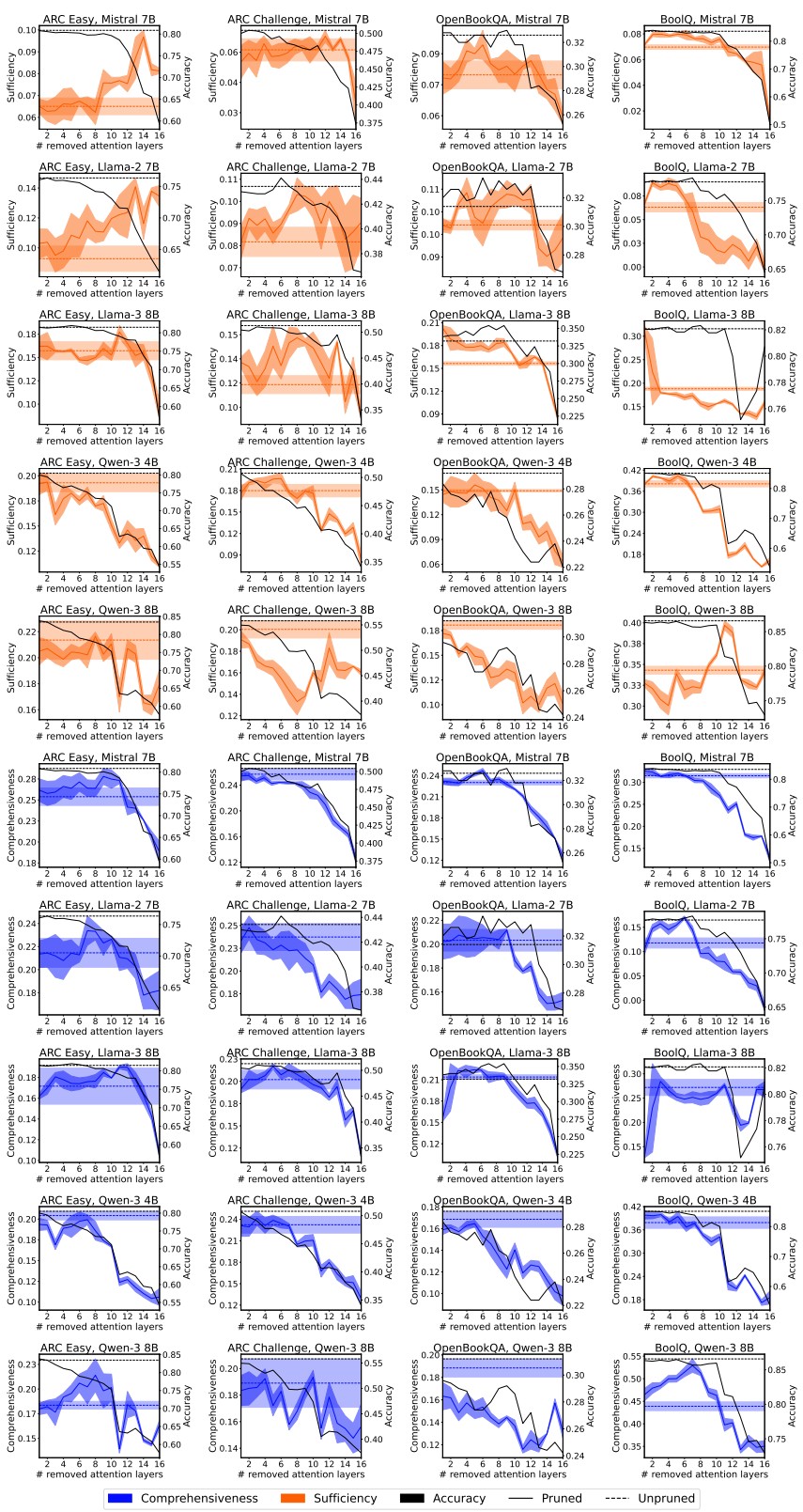

Figure 14: Comprehensiveness (↑), sufficiency (↓), and accuracy (↑) of models (y axis) on QA tasks, using Kernel SHAP, at varying amounts of removed attention layers (x axis).

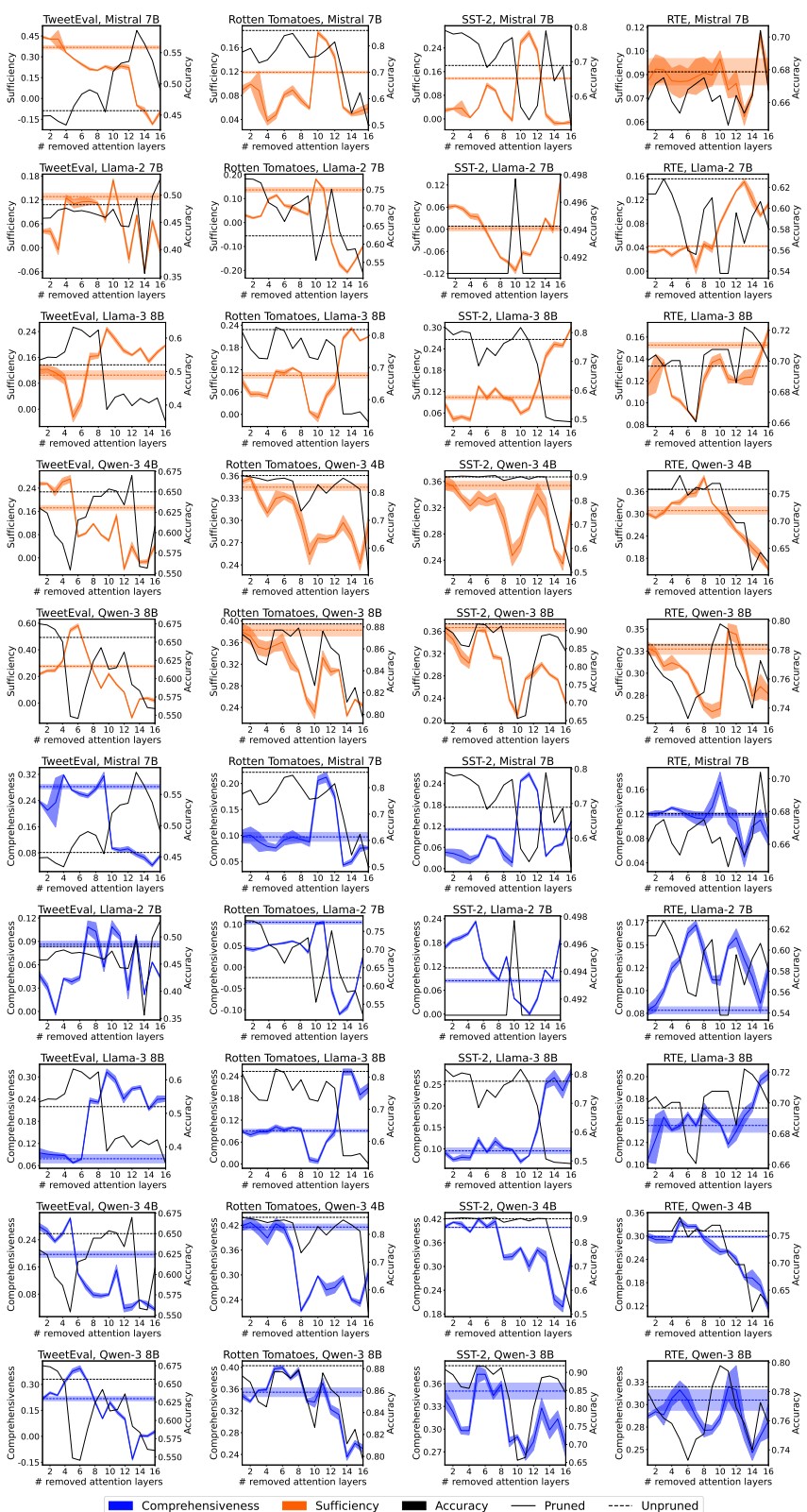

Figure 15: Comprehensiveness (↑), sufficiency (↓), and accuracy (↑) of models (y axis) on sentiment analysis tasks and RTE, using Kernel SHAP, at varying amounts of removed attention layers (x axis).

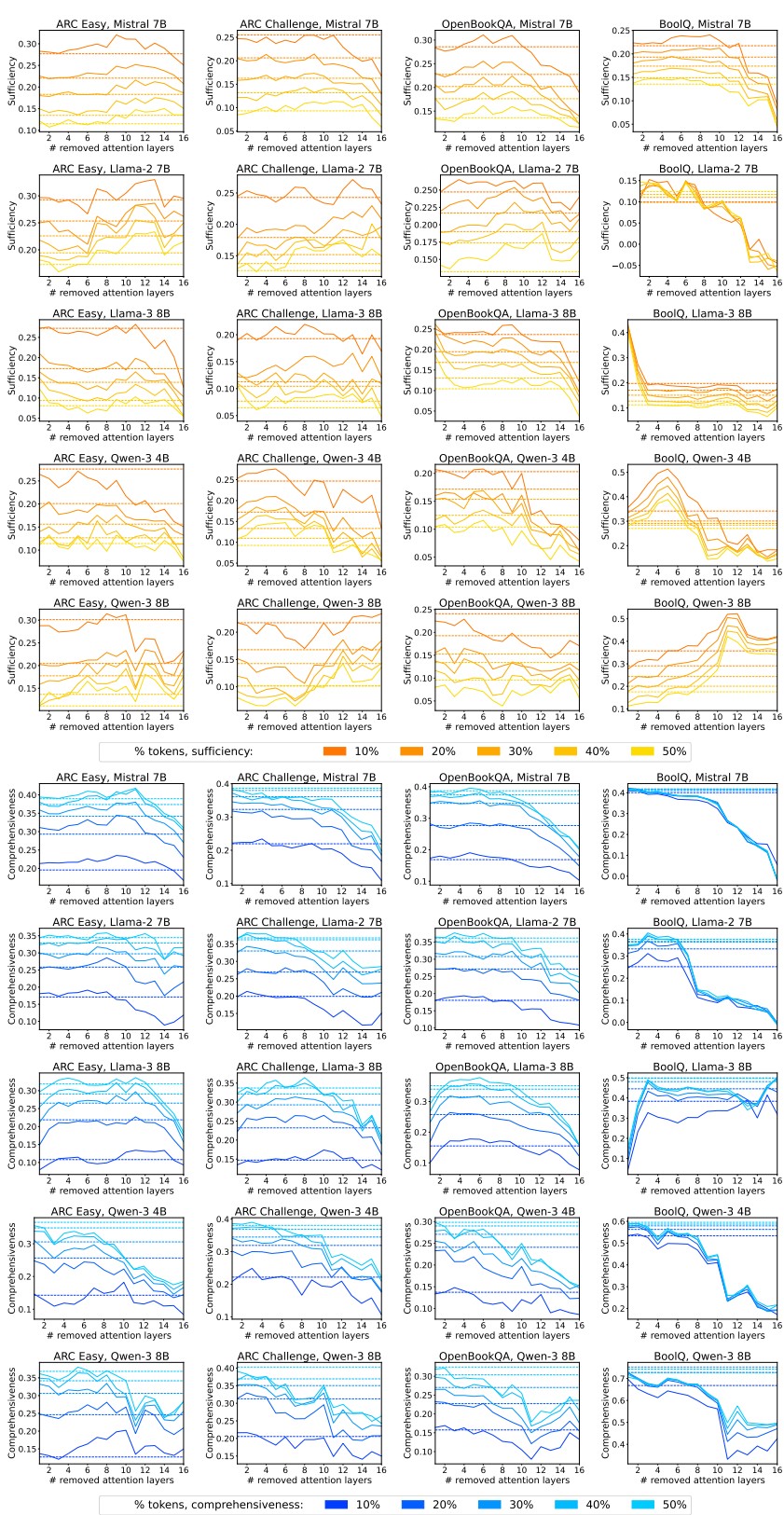

Figure 16: Partial comprehensiveness (↑), sufficiency (↓), and accuracy (↑) of models (y axis) on QA tasks, using LIME, at varying amounts of removed attention layers (x axis).

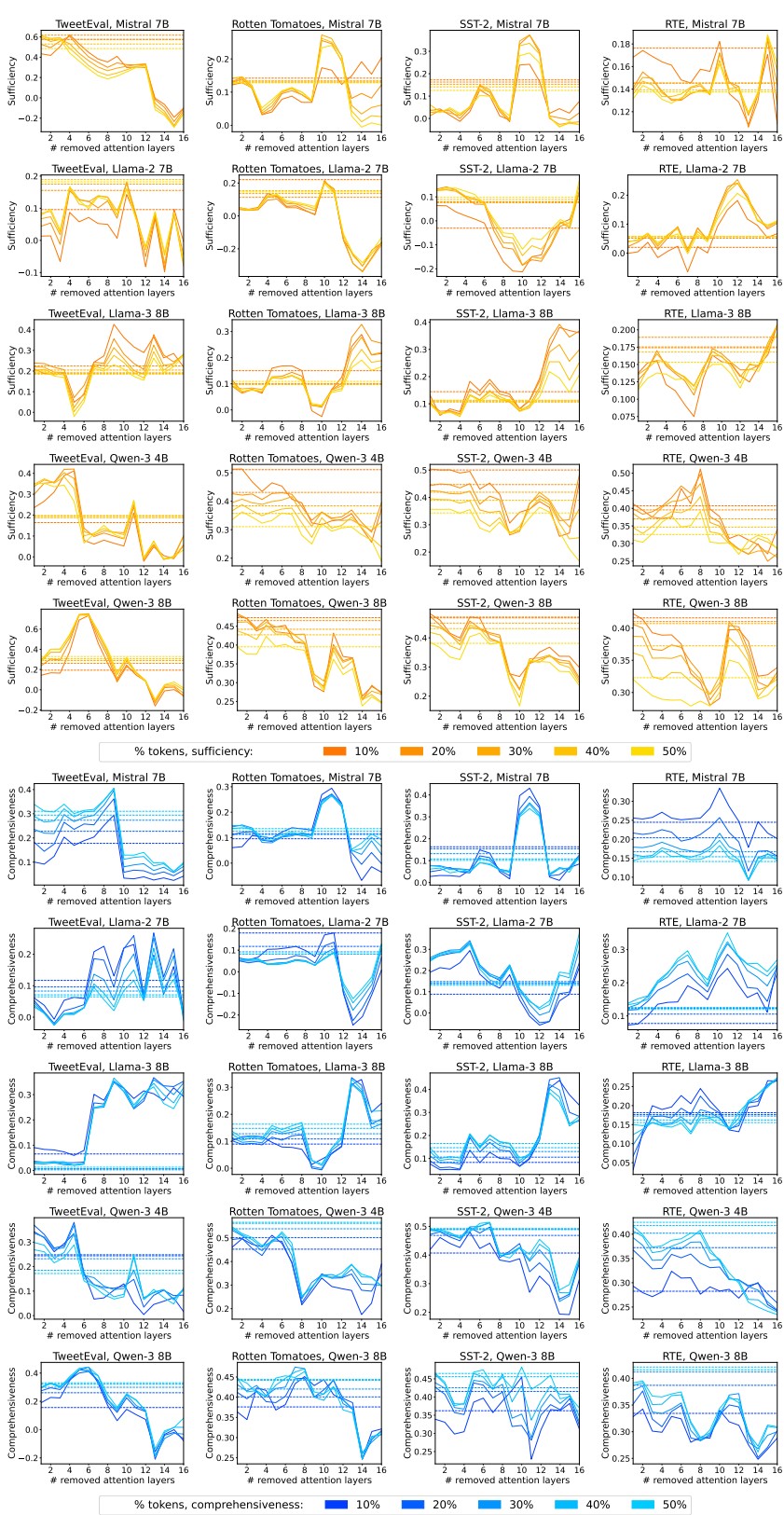

Figure 17: Partial comprehensiveness (↑), sufficiency (↓), and accuracy (↑) of models (y axis) on sentiment analysis and RTE tasks, using LIME, at varying amounts of removed attention layers (x axis).

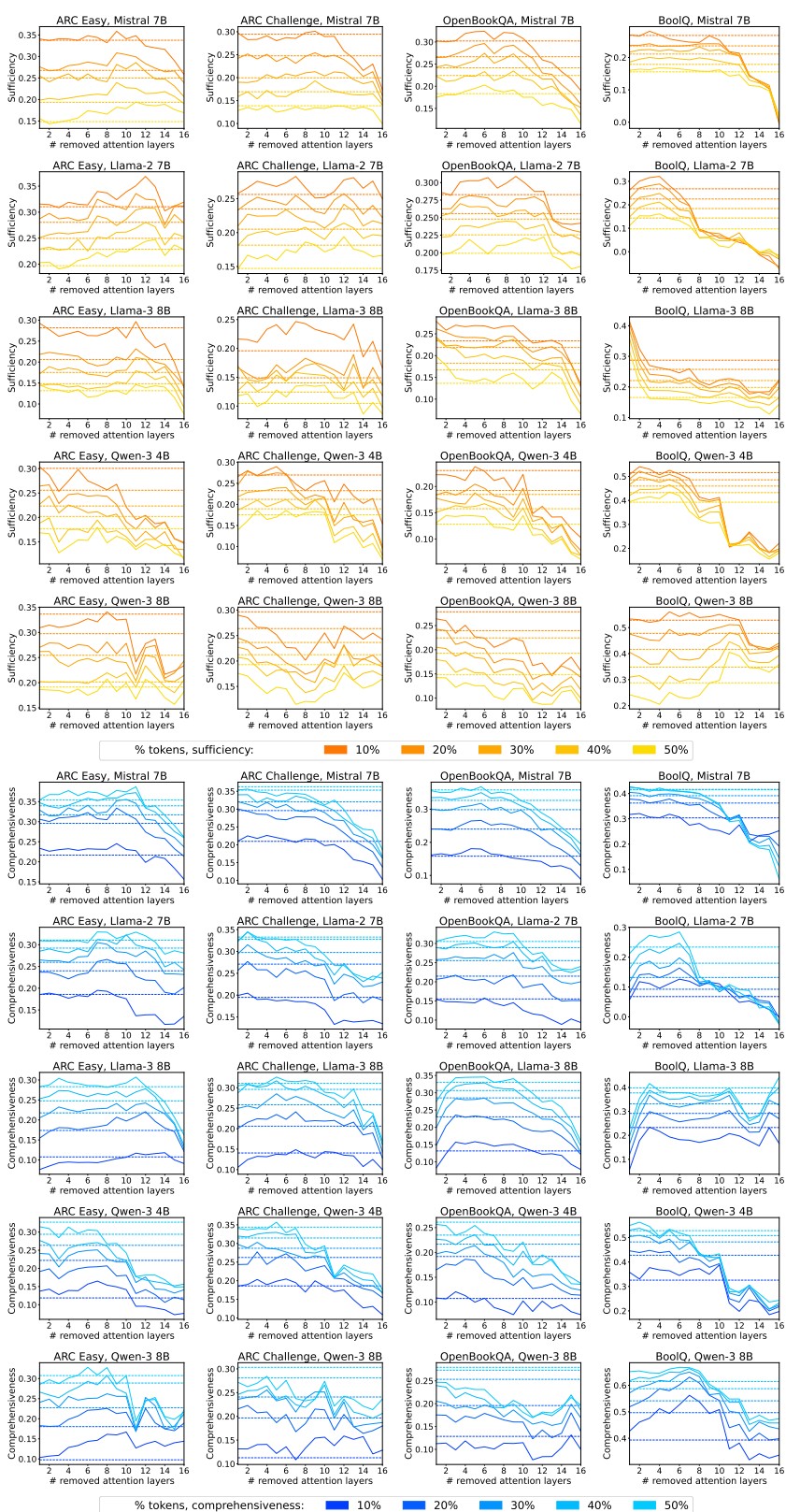

Figure 18: Partial comprehensiveness (↑), sufficiency (↓), and accuracy (↑) of models (y axis) on QA tasks, using Kernel SHAP, at varying amounts of removed attention layers (x axis).

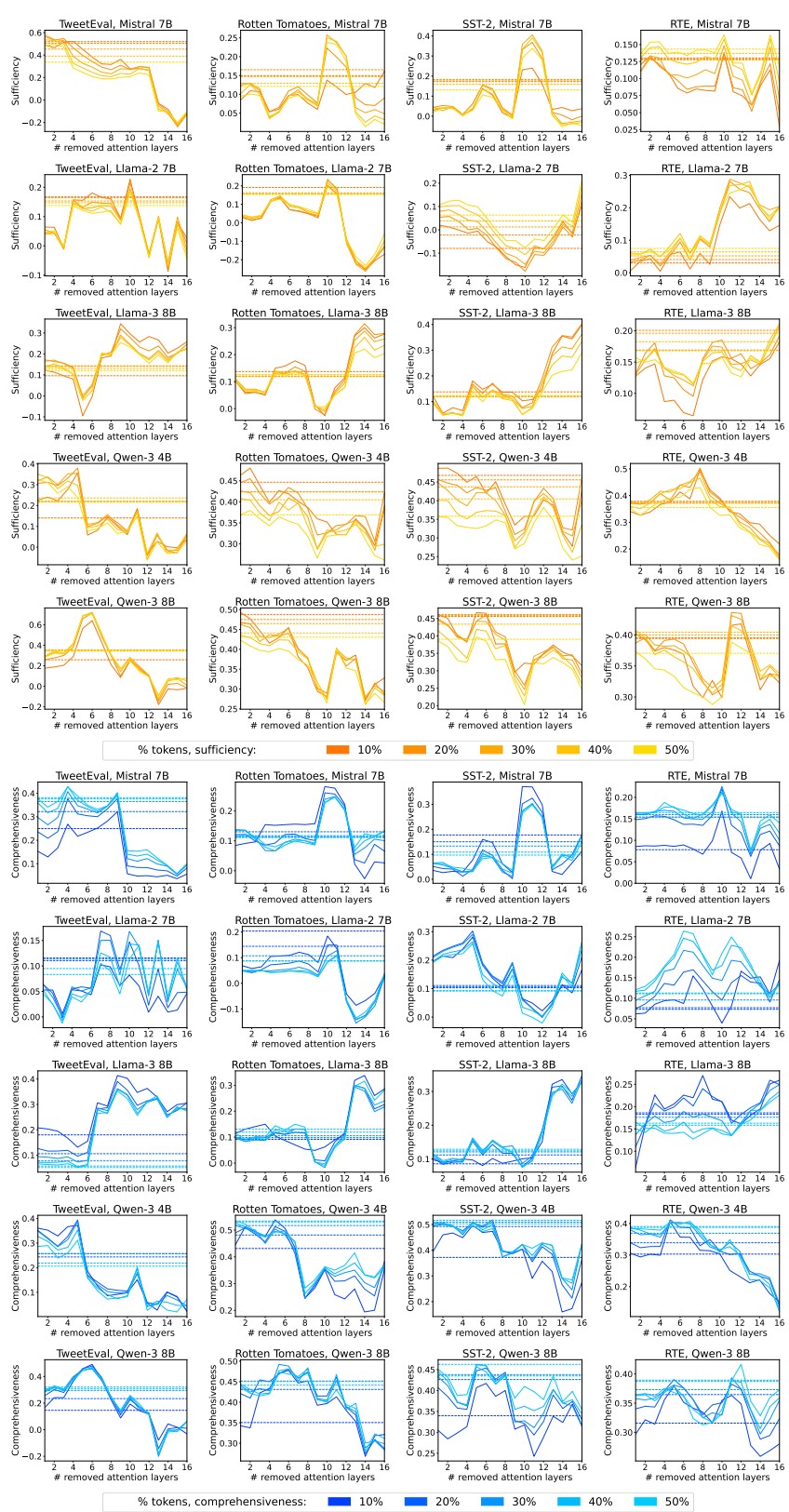

Figure 19: Partial comprehensiveness (↑), sufficiency (↓), and accuracy (↑) of models (y axis) on sentiment analysis and RTE tasks, using Kernel SHAP, at varying amounts of removed attention layers (x axis).

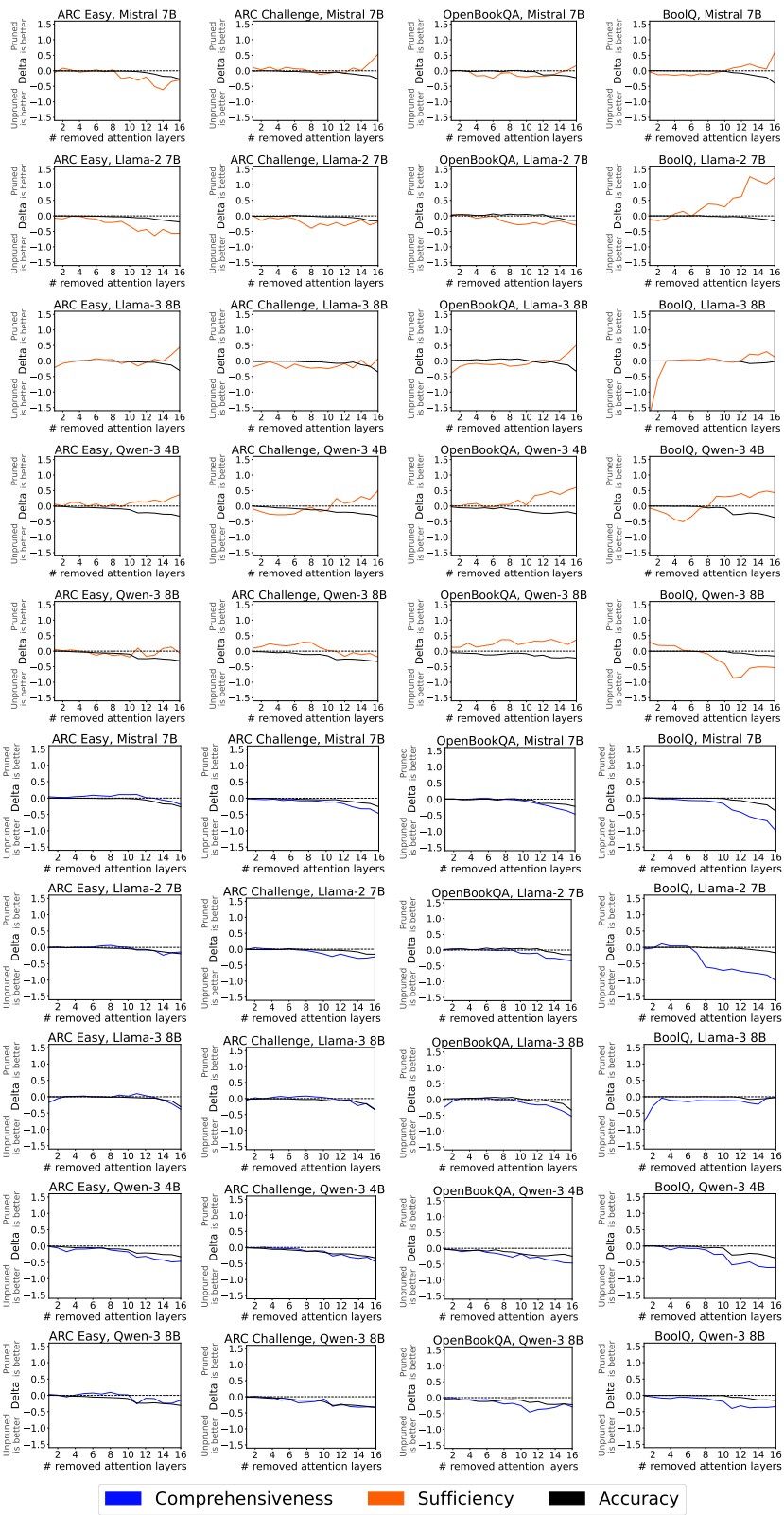

Figure 20: Relative reduction in comprehensiveness and sufficiency using LIME, and accuracy (y axis) between pruned and unpruned models across varying levels of attention layer removal. Accuracy and comprehensiveness reductions are negated, so that negative effects appear on the lower side of the plot.

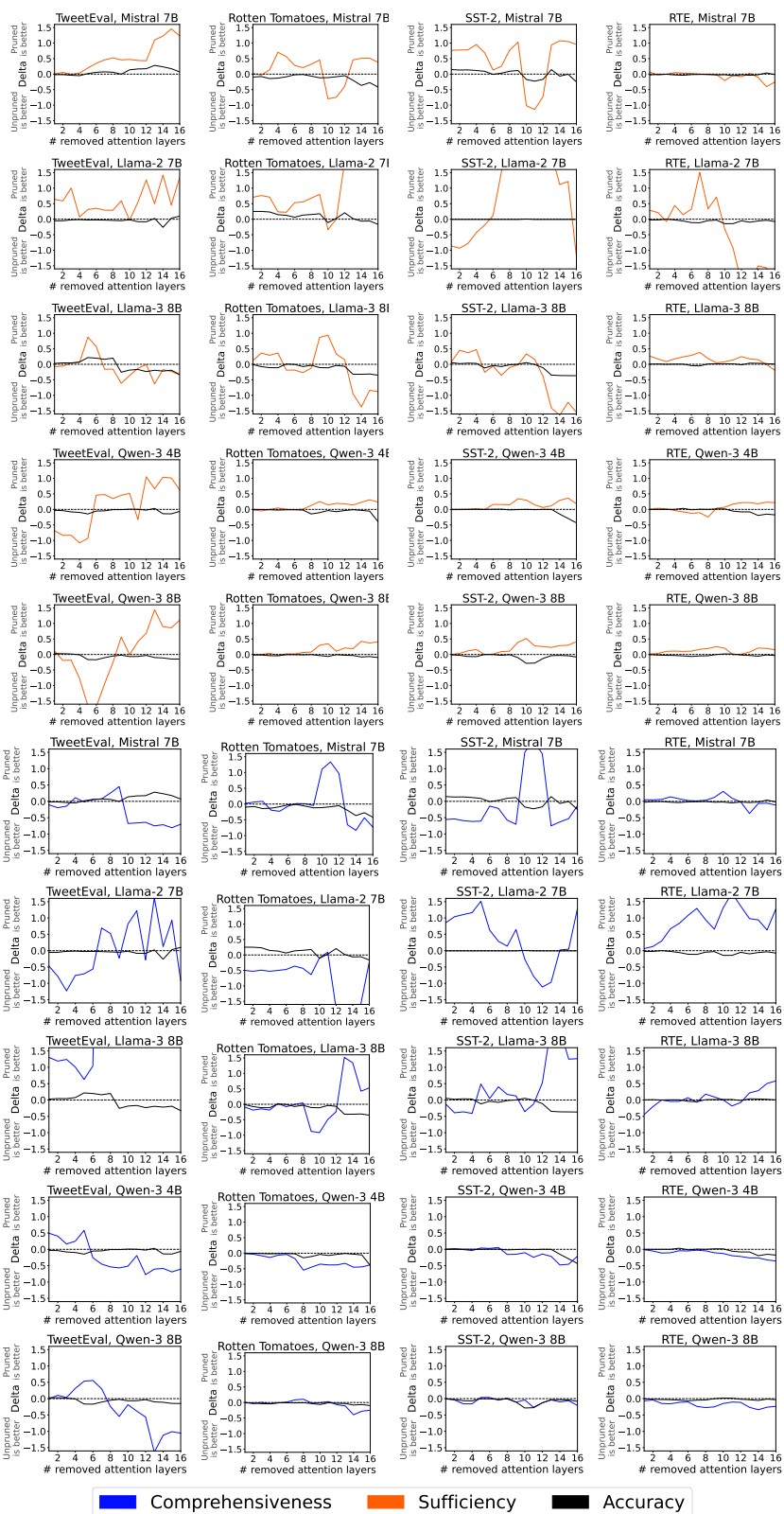

Figure 21: Relative reduction in comprehensiveness and sufficiency using LIME, and accuracy (y axis) between pruned and unpruned models across varying levels of attention layer removal. Accuracy and comprehensiveness reductions are negated, so that negative effects appear on the lower side of the plot.

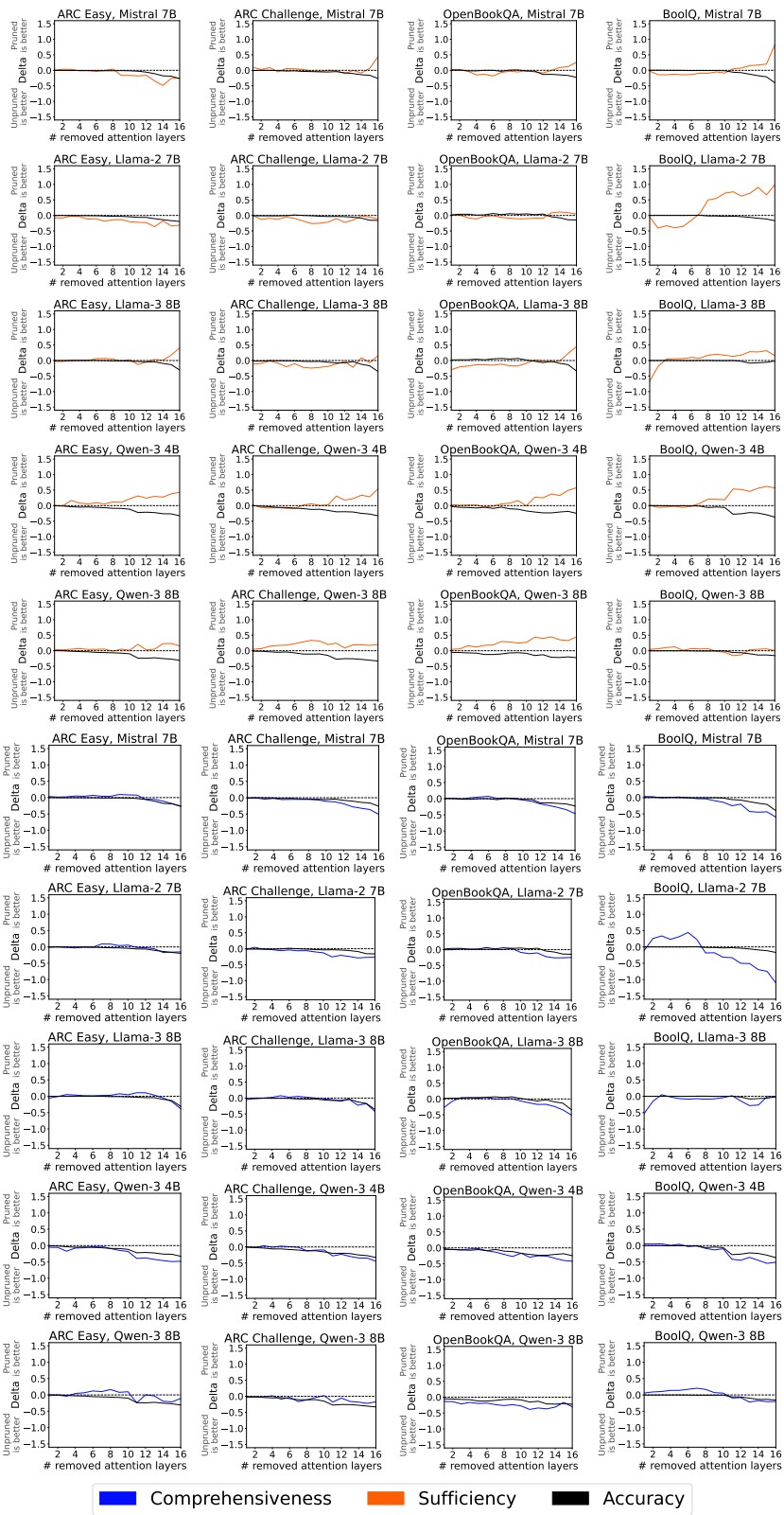

Figure 22: Relative reduction in comprehensiveness and sufficiency using Kernel SHAP, and accuracy (y axis) between pruned and unpruned models across varying levels of attention layer removal. Accuracy and comprehensiveness reductions are negated, so that negative effects appear on the lower side of the plot.

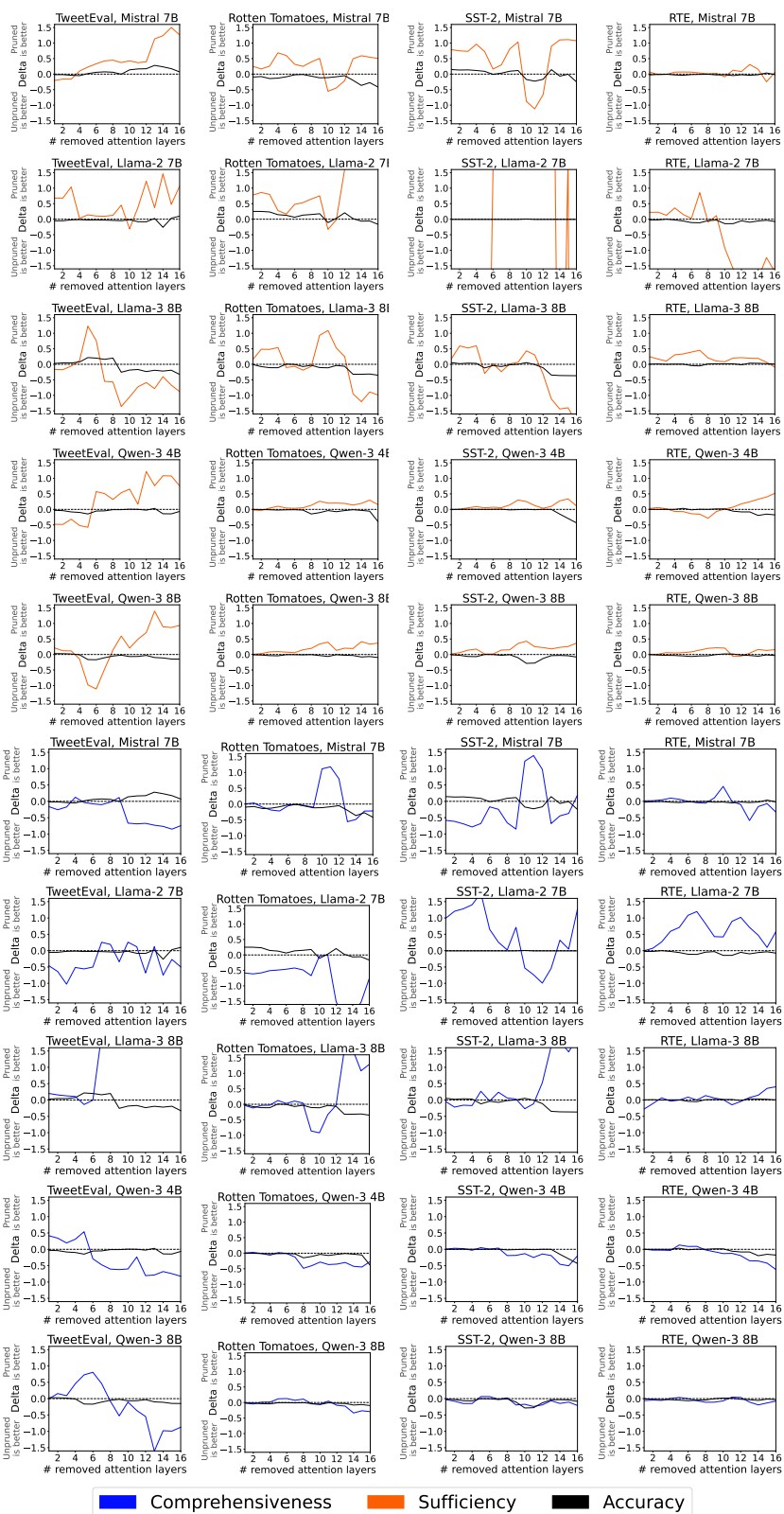

Figure 23: Relative reduction in comprehensiveness and sufficiency using Kernel SHAP, and accuracy (y axis) between pruned and unpruned models across varying levels of attention layer removal. Accuracy and comprehensiveness reductions are negated, so that negative effects appear on the lower side of the plot.

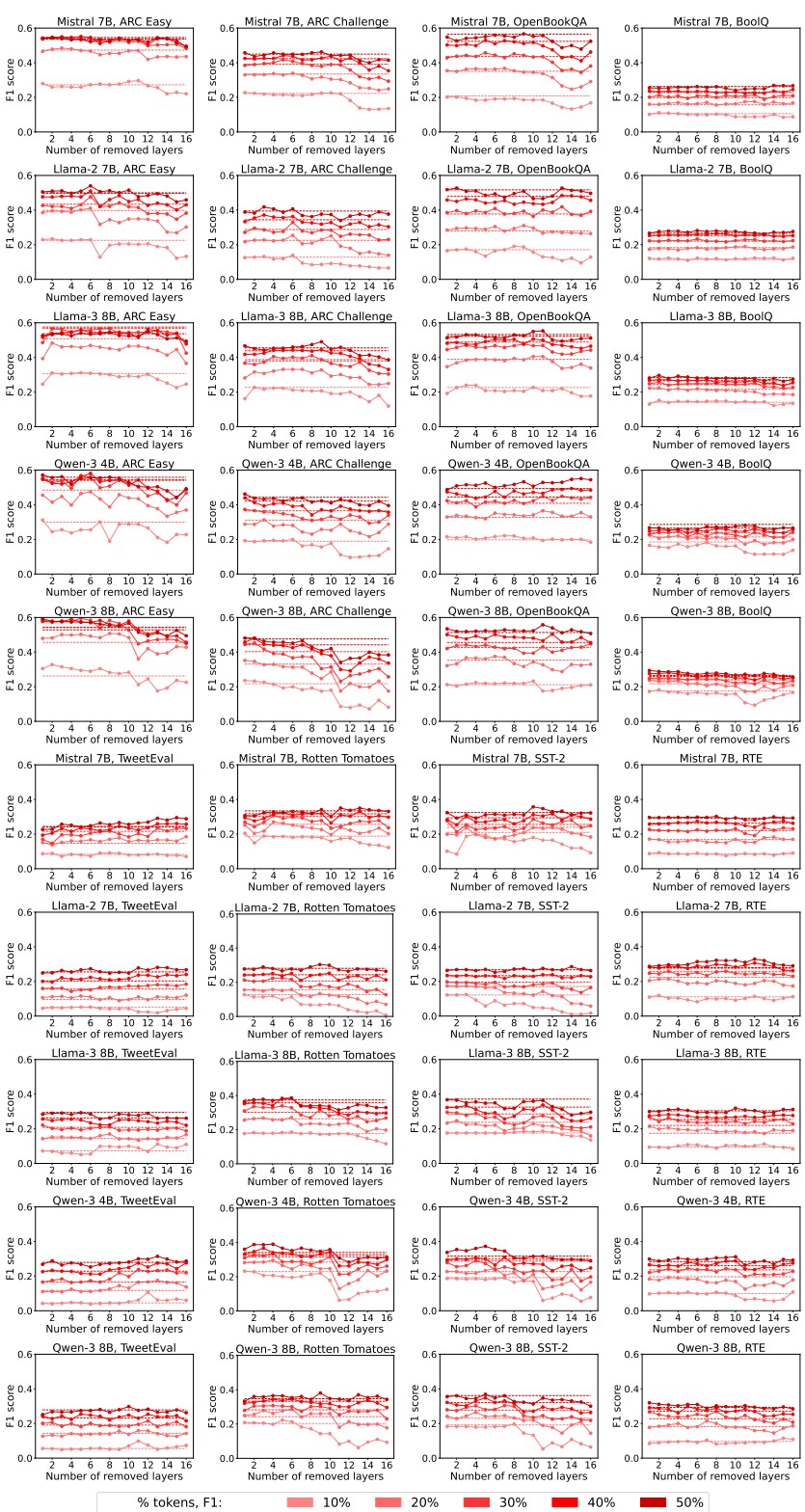

Figure 24: F1 score (y-axis) between LIME attributions and human annotations as a function of pruned attention layers (x-axis). Colours denote the proportion of top features considered ($K$), and dotted lines indicate unpruned models.

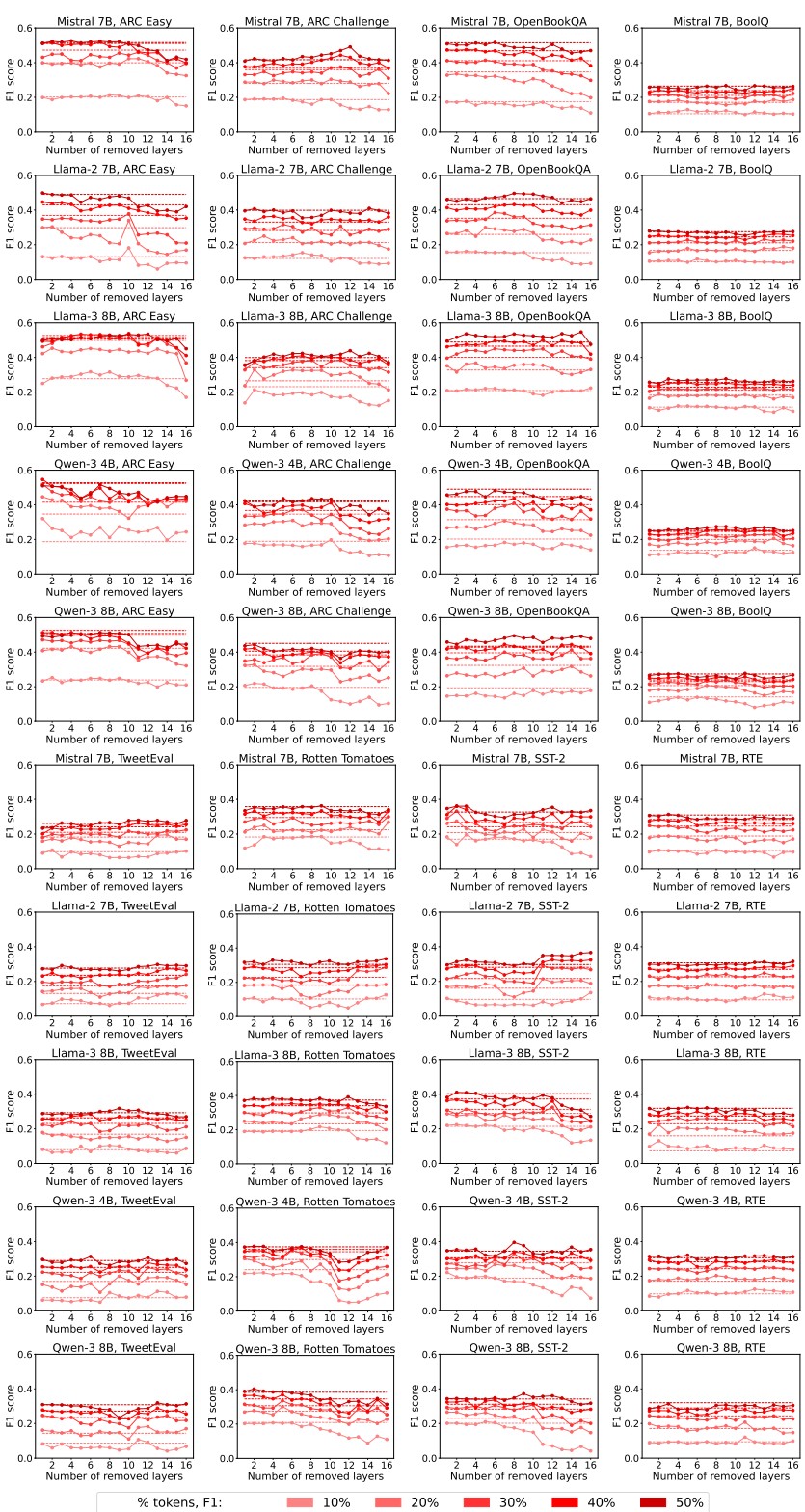

Figure 25: F1 score (y-axis) between Kernel SHAP attributions and human annotations as a function of pruned attention layers (x-axis). Colours denote the proportion of top features considered ($K$), and dotted lines indicate unpruned models.

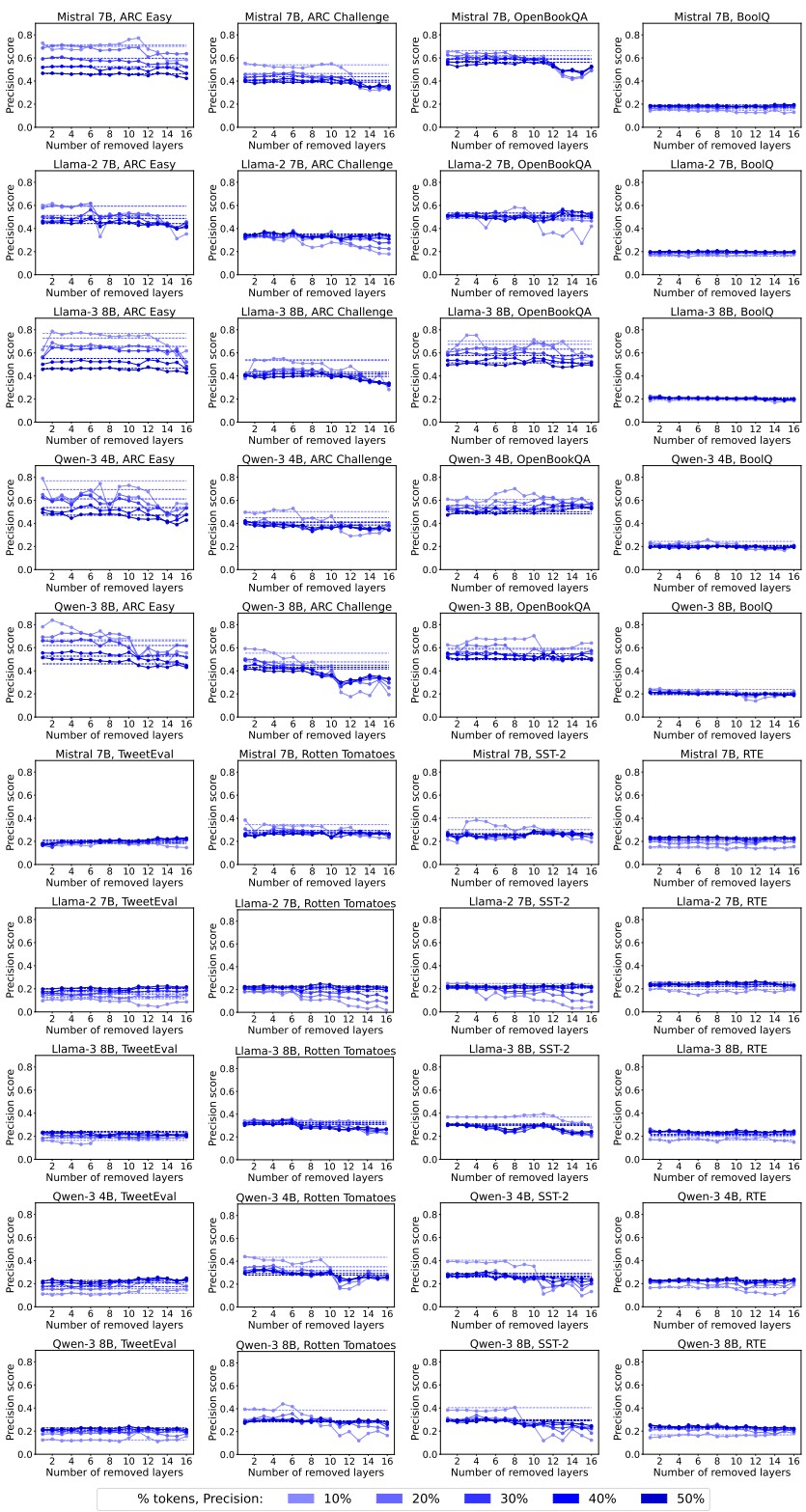

Figure 26: Precision (y-axis) between LIME attributions and human annotations as a function of pruned attention layers (x-axis). Colours denote the proportion of top features considered ($K$), and dotted lines indicate unpruned models.

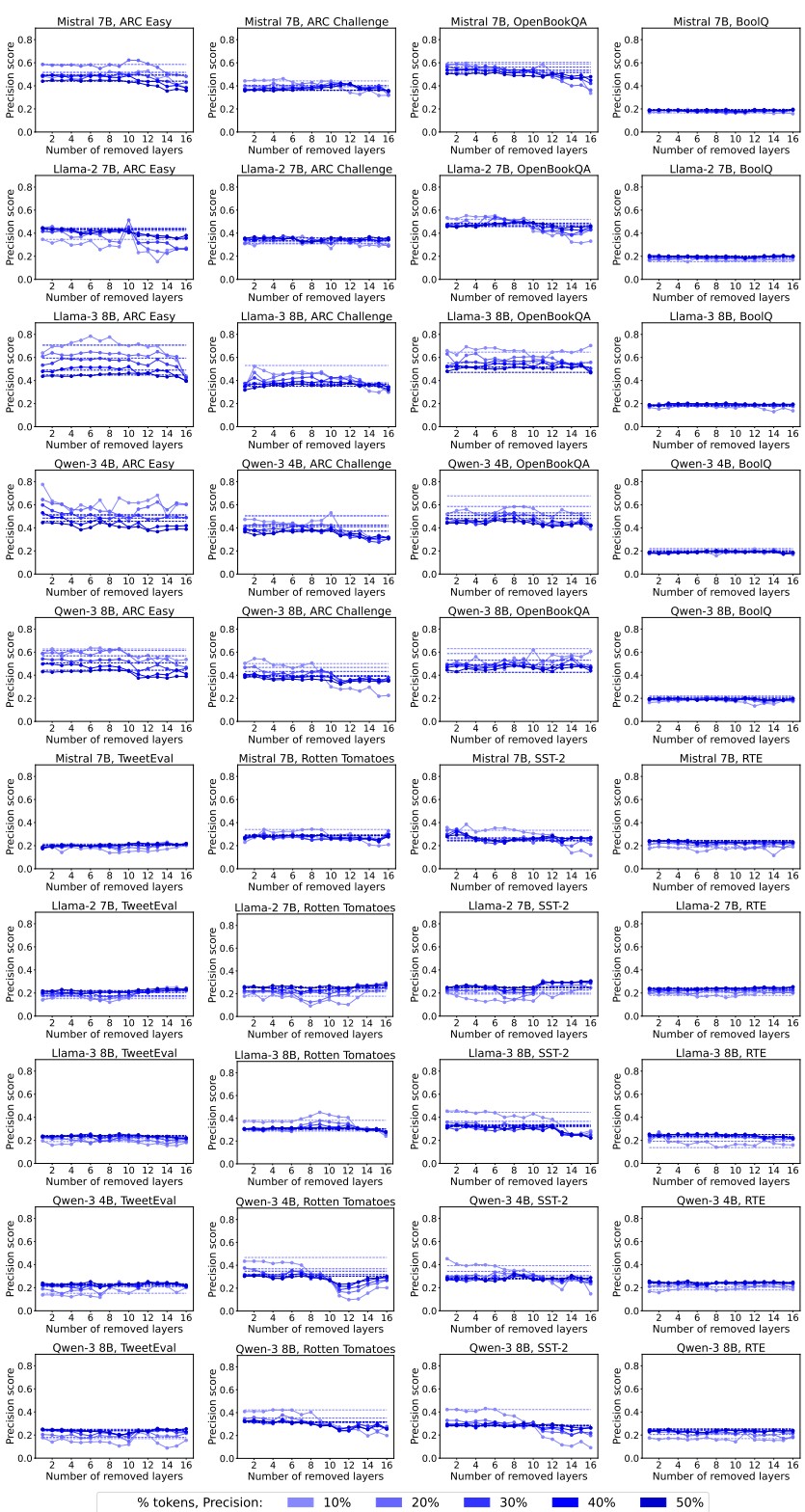

Figure 27: Precision (y-axis) between Kernel SHAP attributions and human annotations as a function of pruned attention layers (x-axis). Colours denote the proportion of top features considered ($K$), and dotted lines indicate unpruned models.

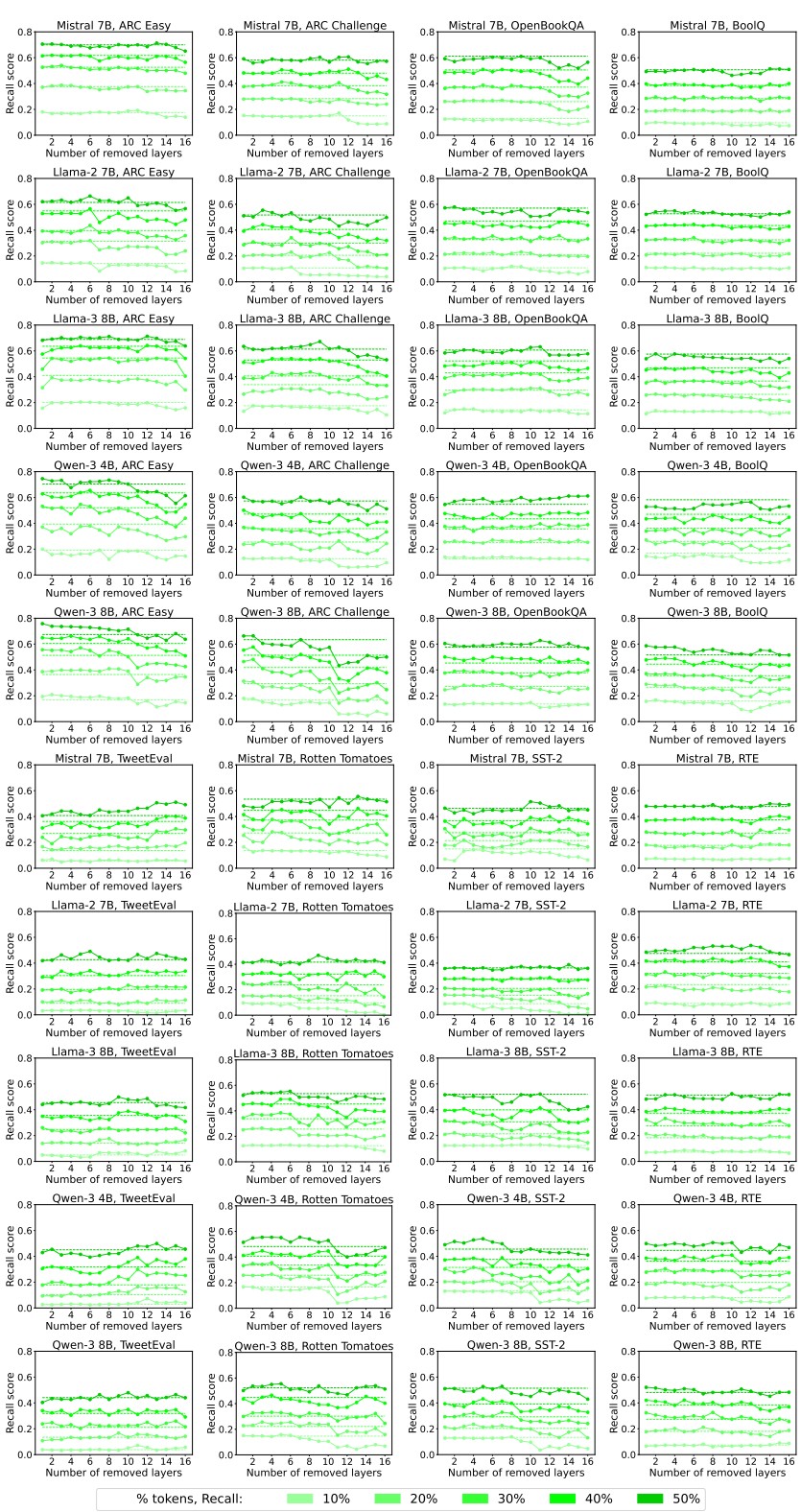

Figure 28: Recall (y-axis) between LIME attributions and human annotations as a function of pruned attention layers (x-axis). Colours denote the proportion of top features considered ($K$), and dotted lines indicate unpruned models.

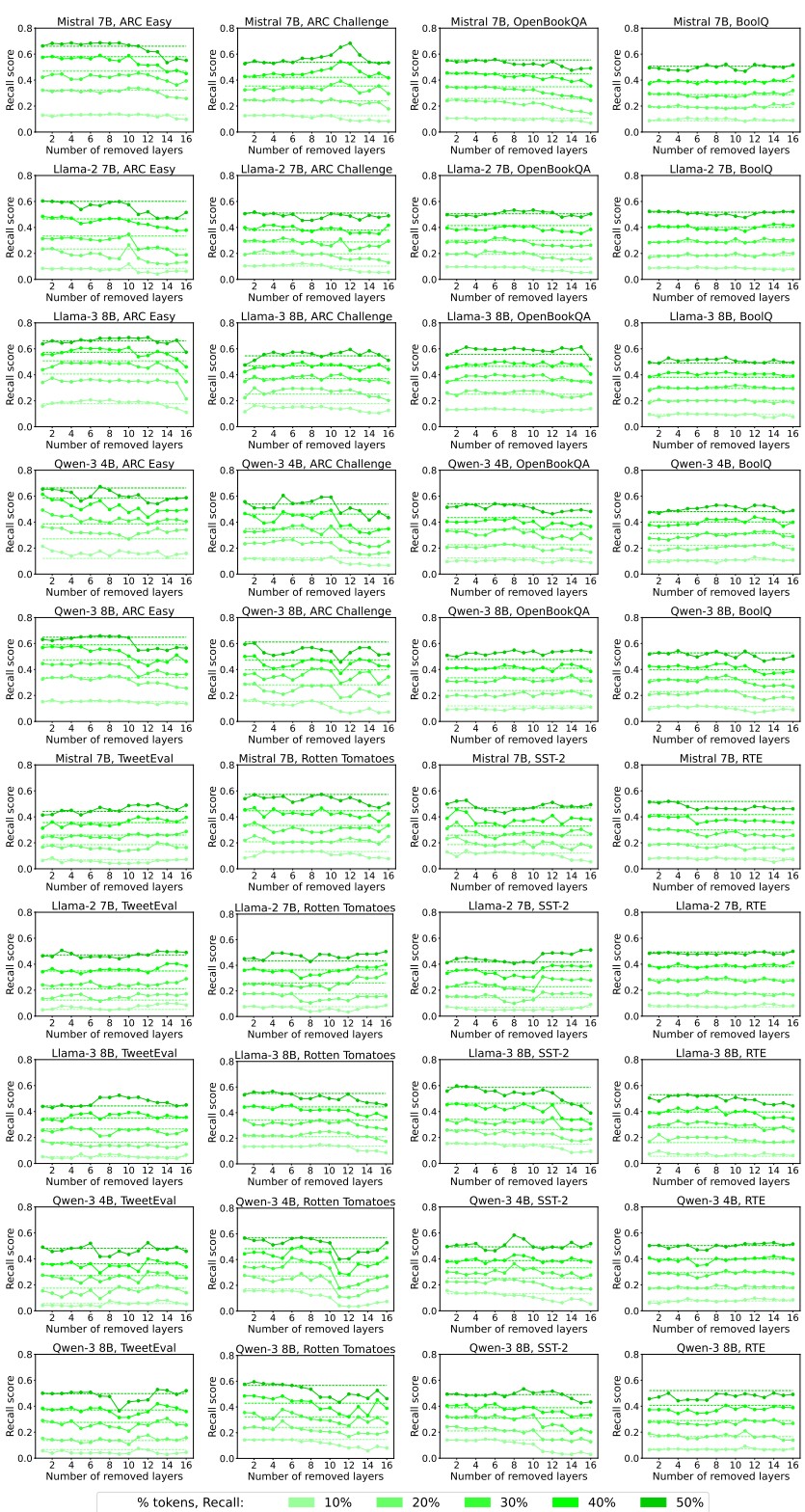

Figure 29: Recall (y-axis) between Kernel SHAP attributions and human annotations as a function of pruned attention layers (x-axis). Colours denote the proportion of top features considered ($K$), and dotted lines indicate unpruned models.

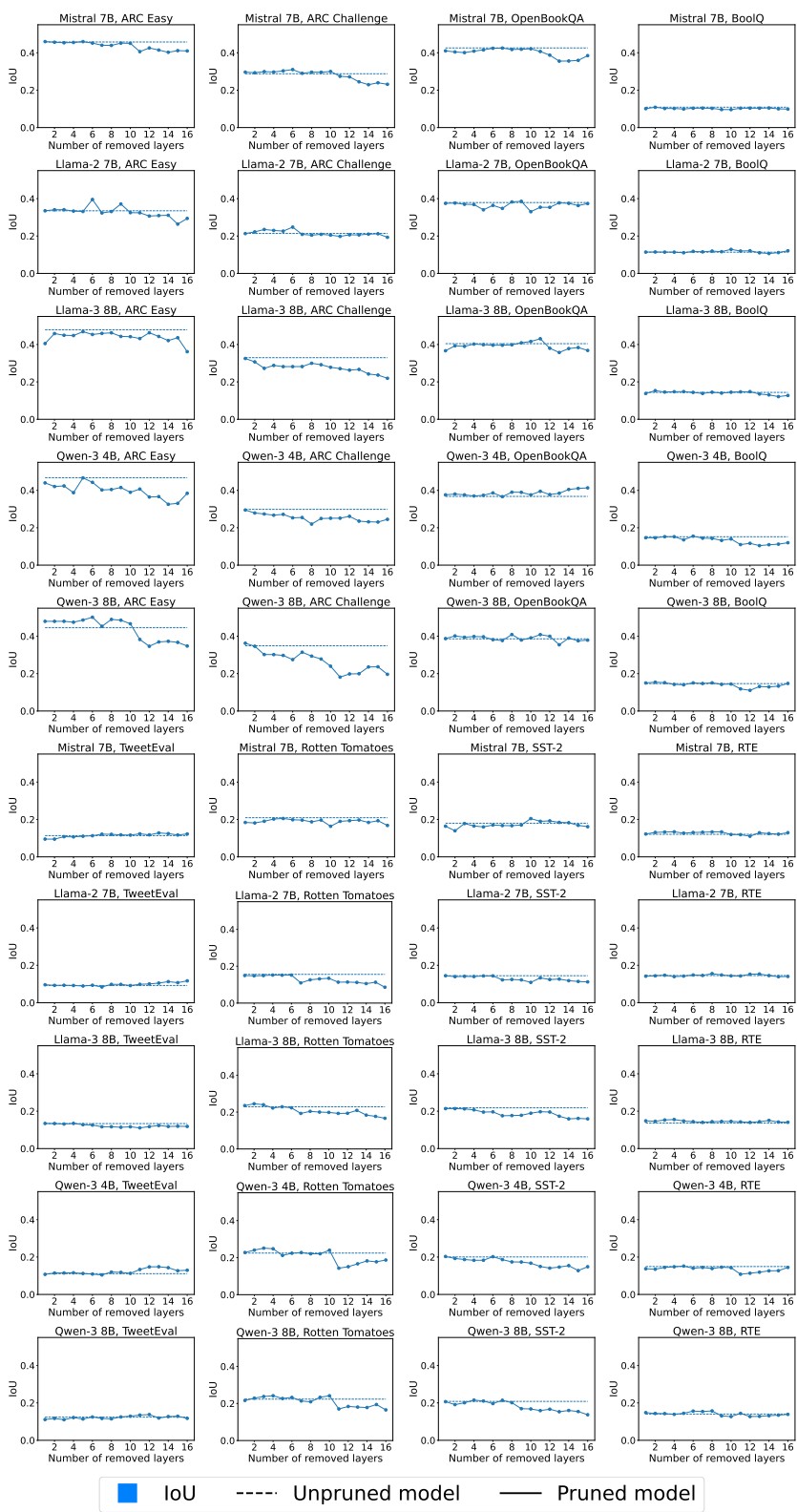

Figure 30: Intersection-over-Union (y-axis) between LIME attributions and human annotations as a function of pruned attention layers (x-axis). The dotted line denotes the unpruned model.

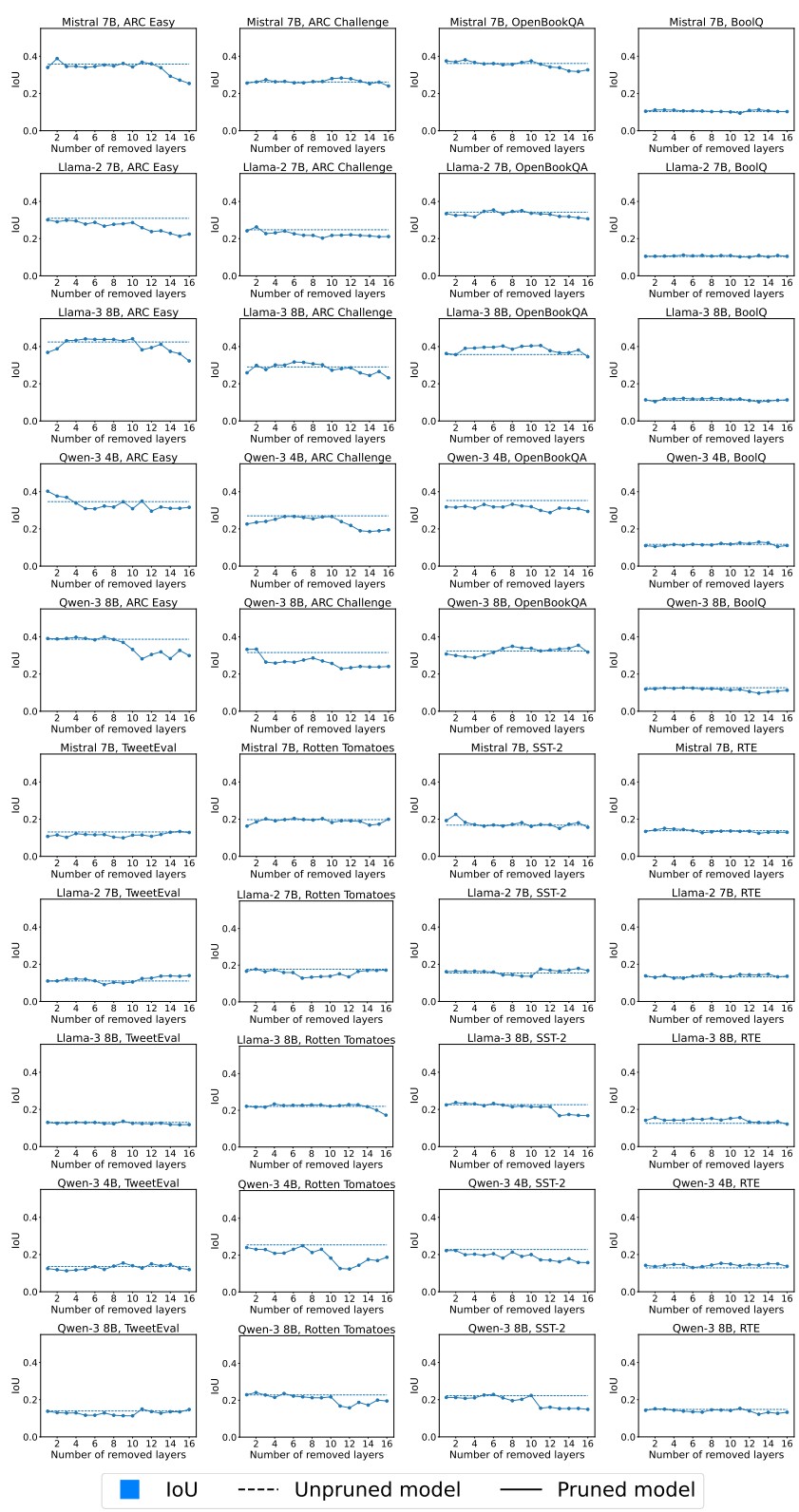

Figure 31: Intersection-over-Union (y-axis) between Kernel SHAP attributions and human annotations as a function of pruned attention layers (x-axis). The dotted line denotes the unpruned model.

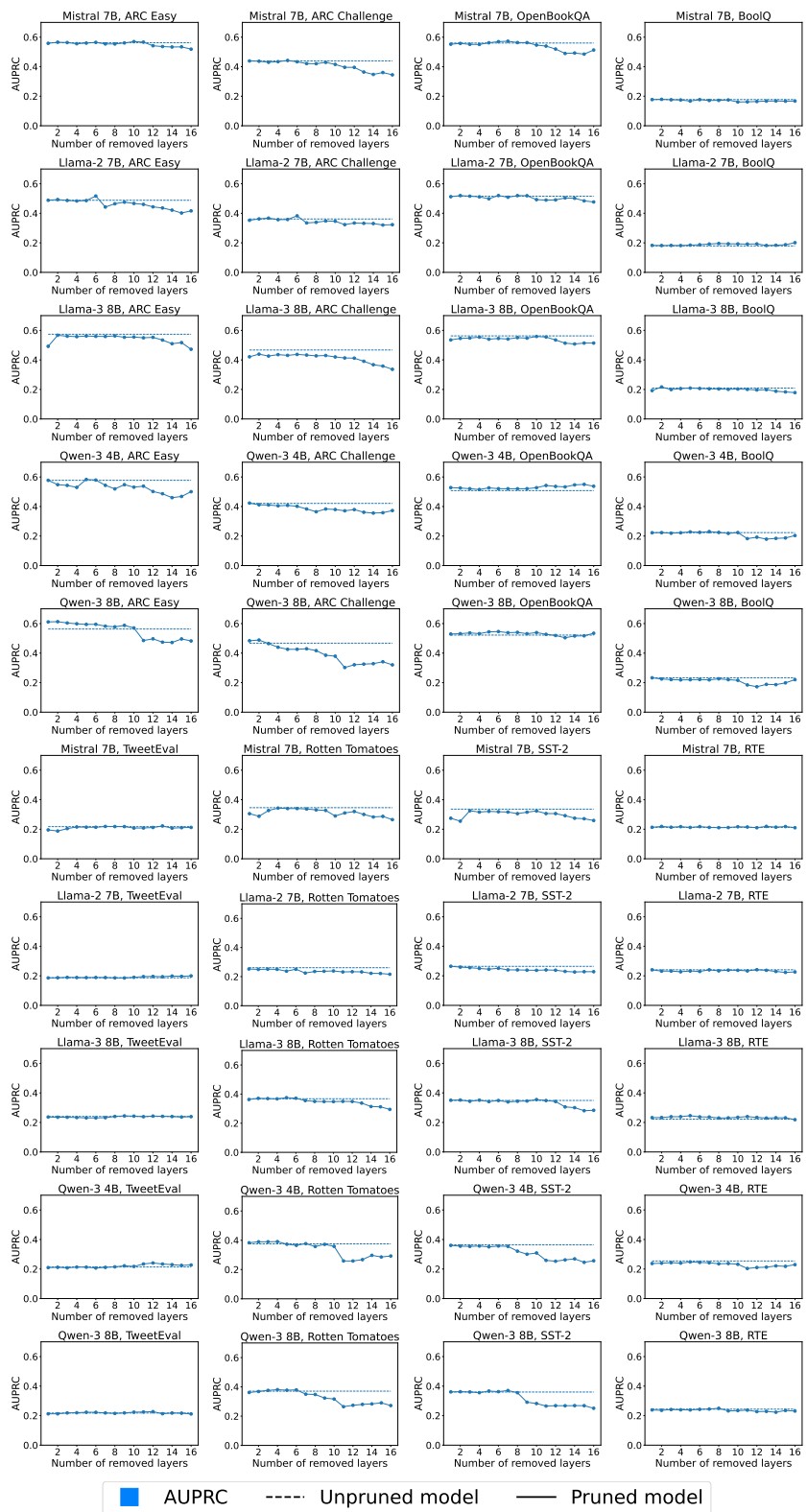

Figure 32: Area Under the Precision-Recall Curve (y-axis) between LIME attributions and human annotations as a function of pruned attention layers (x-axis). The dotted line denotes the unpruned model.

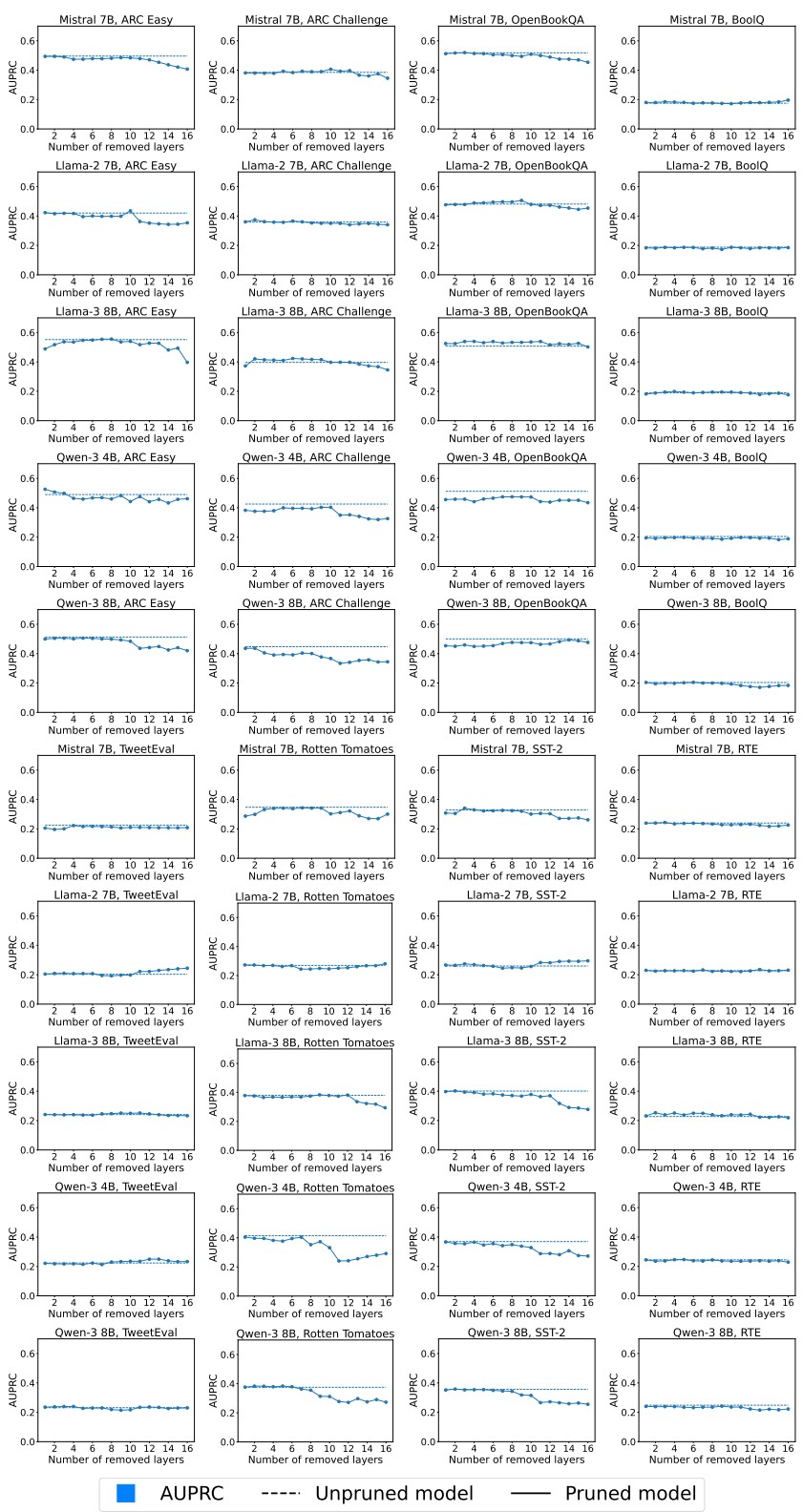

Figure 33: Area Under the Precision-Recall Curve (y-axis) between Kernel SHAP attributions and human annotations as a function of pruned attention layers (x-axis). The dotted line denotes the unpruned model.

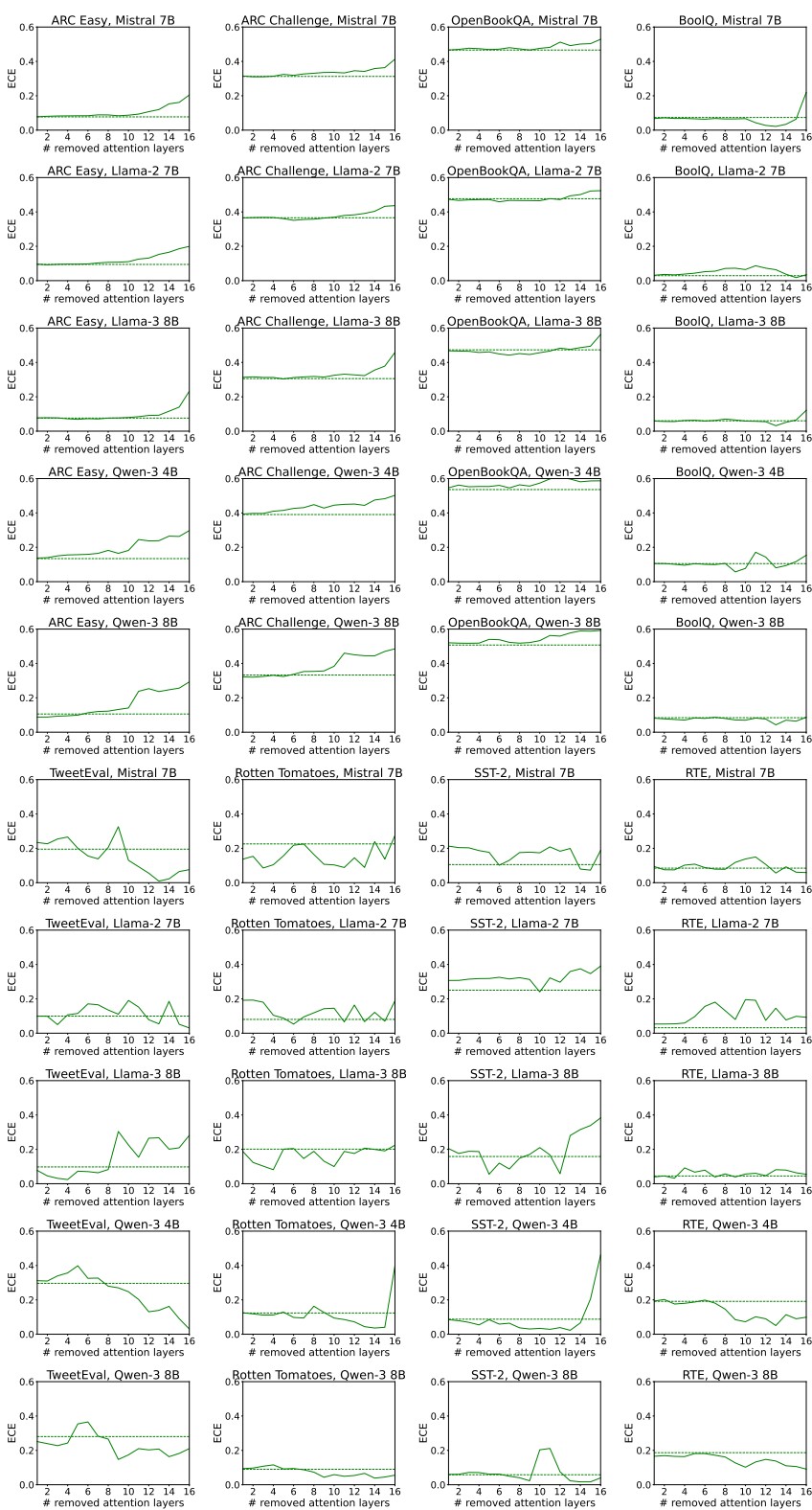

Figure 34: Expected Calibration Error (y axis) for Mistral (first row), Llama-2 (second row), Llama-3.1 (third row), Qwen3-4B (fourth row), and Qwen3-8B (fifth row) at different amounts of removed attention layers (x axis).

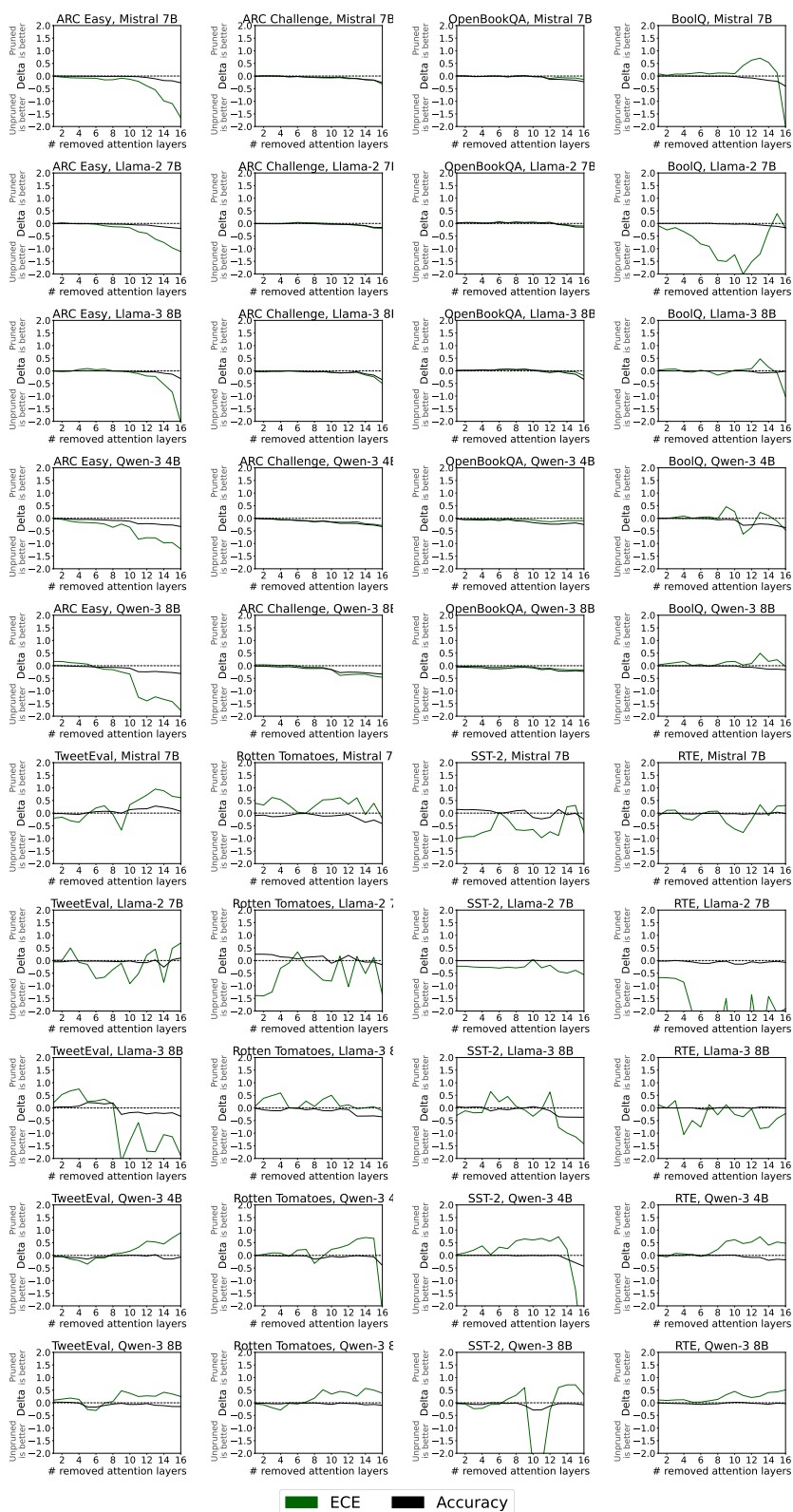

Figure 35: Relative reduction in ECE and accuracy (y axis) for Mistral (first row), Llama-2 (second row), Llama-3.1 (third row), Qwen3-4B (fourth row), and Qwen3-8B (fifth row) at different amounts of removed attention layers (x axis). Accuracy reductions are negated, so negative effects are on the plots' lower side.

Table 8: Results for Llama-2 on ARC Easy. Comp./Suff. denote comprehensiveness/sufficiency; KS denotes Kernel SHAP; orig denotes the unpruned model. Significant changes ($p < 0.05$) are marked with *. Wilcoxon's test is used for Comp./Suff., and McNemar's test for accuracy.

| Pruned Layers | Accuracy | Comp. LIME | Suff. LIME | Comp. KS | Suff. KS | ECE |
|---|---|---|---|---|---|---|
| 0 (orig) | 0.763 | 0.233 | 0.190 | 0.215 | 0.211 | 0.094 |
| 1 | 0.761 | 0.234 | 0.189 | 0.213 | 0.214 | 0.096 |
| 2 | 0.763 | 0.236* | 0.187 | 0.215 | 0.218 | 0.092 |
| 3 | 0.759 | 0.230 | 0.177* | 0.212 | 0.212 | 0.095 |
| 4 | 0.760 | 0.235 | 0.180 | 0.208 | 0.215 | 0.096 |
| 5 | 0.758 | 0.235 | 0.179 | 0.214 | 0.219 | 0.096 |
| 6 | 0.756 | 0.236 | 0.172* | 0.213 | 0.217 | 0.097 |
| 7 | 0.749* | 0.244* | 0.207* | 0.234* | 0.239* | 0.103 |
| 8 | 0.742* | 0.248* | 0.204* | 0.233* | 0.234* | 0.107 |
| 9 | 0.741* | 0.237 | 0.197* | 0.223 | 0.233* | 0.107 |
| 10 | 0.731* | 0.234 | 0.209* | 0.226 | 0.236* | 0.111 |
| 11 | 0.716* | 0.214* | 0.229* | 0.211 | 0.249* | 0.125 |
| 12 | 0.715* | 0.215* | 0.229* | 0.209 | 0.254* | 0.131 |
| 13 | 0.684* | 0.211* | 0.227* | 0.203 | 0.250* | 0.153 |
| 14 | 0.658* | 0.177* | 0.192 | 0.178* | 0.217 | 0.165 |
| 15 | 0.635* | 0.196* | 0.206* | 0.180* | 0.236* | 0.186 |
| 16 | 0.614* | 0.200* | 0.208* | 0.182* | 0.232* | 0.199 |

Table 9: Results for Llama-2 on ARC Challenge. Comp./Suff. denote comprehensiveness/sufficiency; KS denotes Kernel SHAP; orig denotes the unpruned model. Significant changes ($p < 0.05$) are marked with *. Wilcoxon's test is used for Comp./Suff., and McNemar's test for accuracy.

| Pruned Layers | Accuracy | Comp. LIME | Suff. LIME | Comp. KS | Suff. KS | ECE |
|---|---|---|---|---|---|---|
| 0 (orig) | 0.434 | 0.255 | 0.140 | 0.237 | 0.171 | 0.366 |
| 1 | 0.430 | 0.253 | 0.140 | 0.233 | 0.171 | 0.365 |
| 2 | 0.429 | 0.266* | 0.152 | 0.248* | 0.183* | 0.367 |
| 3 | 0.428 | 0.260* | 0.146 | 0.234 | 0.186* | 0.368 |
| 4 | 0.428 | 0.258 | 0.151 | 0.231 | 0.186 | 0.367 |
| 5 | 0.432 | 0.252 | 0.143 | 0.223 | 0.182 | 0.361 |
| 6 | 0.441 | 0.255 | 0.149 | 0.228 | 0.187 | 0.351 |
| 7 | 0.434 | 0.249 | 0.161* | 0.222 | 0.189* | 0.356 |
| 8 | 0.430 | 0.244 | 0.162* | 0.223* | 0.187* | 0.358 |
| 9 | 0.423 | 0.232 | 0.154* | 0.214* | 0.180 | 0.364 |
| 10 | 0.419 | 0.217* | 0.153* | 0.207 | 0.175 | 0.369 |
| 11 | 0.421 | 0.196* | 0.160* | 0.177* | 0.185* | 0.379 |
| 12 | 0.417 | 0.216* | 0.167* | 0.189* | 0.197* | 0.383 |
| 13 | 0.410* | 0.197* | 0.167* | 0.180* | 0.190 | 0.391 |
| 14 | 0.397* | 0.182* | 0.160 | 0.169* | 0.178 | 0.404 |
| 15 | 0.367* | 0.184* | 0.172* | 0.173* | 0.182 | 0.433 |
| 16 | 0.365* | 0.194* | 0.160* | 0.174* | 0.175 | 0.436 |

Table 10: Results for Llama-2 on OpenBookQA. Comp./Suff. denote comprehensiveness/sufficiency; KS denotes Kernel SHAP; orig denotes the unpruned model. Significant changes ($p < 0.05$) are marked with *. Wilcoxon's test is used for Comp./Suff., and McNemar's test for accuracy.

| Pruned Layers | Accuracy | Comp. LIME | Suff. LIME | Comp. KS | Suff. KS | ECE |
|---|---|---|---|---|---|---|
| 0 (orig) | 0.314 | 0.245 | 0.160 | 0.203 | 0.202 | 0.477 |
| 1 | 0.320 | 0.247 | 0.162 | 0.202 | 0.203 | 0.473 |
| 2 | 0.326 | 0.247 | 0.164 | 0.203 | 0.204 | 0.467 |
| 3 | 0.326 | 0.255 | 0.170 | 0.208 | 0.215 | 0.470 |
| 4 | 0.318 | 0.250 | 0.171 | 0.206 | 0.218 | 0.471 |
| 5 | 0.320 | 0.247 | 0.169 | 0.205 | 0.215 | 0.472 |
| 6 | 0.334* | 0.247 | 0.171 | 0.205 | 0.215 | 0.459 |
| 7 | 0.322 | 0.243 | 0.176* | 0.205 | 0.211 | 0.467 |
| 8 | 0.332 | 0.243 | 0.184* | 0.204 | 0.212 | 0.466 |
| 9 | 0.326 | 0.246 | 0.186* | 0.212 | 0.217 | 0.466 |
| 10 | 0.330 | 0.221* | 0.178* | 0.186* | 0.215 | 0.465 |
| 11 | 0.322 | 0.218* | 0.180* | 0.177* | 0.212 | 0.477 |
| 12 | 0.328 | 0.221* | 0.185* | 0.182* | 0.213 | 0.472 |
| 13 | 0.300 | 0.182* | 0.161 | 0.159* | 0.188 | 0.494 |
| 14 | 0.290 | 0.182* | 0.161 | 0.150* | 0.181 | 0.501 |
| 15 | 0.270* | 0.172* | 0.155 | 0.151* | 0.179 | 0.522 |
| 16 | 0.268* | 0.162* | 0.168* | 0.153* | 0.178 | 0.524 |

Table 11: Results for Llama-2 on BoolQ. Comp./Suff. denote comprehensiveness/sufficiency; KS denotes Kernel SHAP; orig denotes the unpruned model. Significant changes ($p < 0.05$) are marked with *. Wilcoxon's test is used for Comp./Suff., and McNemar's test for accuracy.

| Pruned Layers | Accuracy | Comp. LIME | Suff. LIME | Comp. KS | Suff. KS | ECE |
|---|---|---|---|---|---|---|
| 0 (orig) | 0.777 | 0.282 | 0.092 | 0.118 | 0.153 | 0.029 |
| 1 | 0.777 | 0.267* | 0.105* | 0.104* | 0.157 | 0.031 |
| 2 | 0.779 | 0.275 | 0.119* | 0.148* | 0.191* | 0.036 |
| 3 | 0.778 | 0.312* | 0.117* | 0.157* | 0.198* | 0.034 |
| 4 | 0.779 | 0.294* | 0.104* | 0.145* | 0.205* | 0.038 |
| 5 | 0.777 | 0.294* | 0.087 | 0.155* | 0.181* | 0.044 |
| 6 | 0.780 | 0.293* | 0.122* | 0.170* | 0.154 | 0.052 |
| 7 | 0.783 | 0.232* | 0.104* | 0.145* | 0.137 | 0.055 |
| 8 | 0.765 | 0.112* | 0.072 | 0.096 | 0.067* | 0.071 |
| 9 | 0.763 | 0.102* | 0.073 | 0.097 | 0.058* | 0.072 |
| 10 | 0.753* | 0.084* | 0.075 | 0.080 | 0.046* | 0.065 |
| 11 | 0.759* | 0.095* | 0.055* | 0.079 | 0.038* | 0.087 |
| 12 | 0.747* | 0.076* | 0.048* | 0.059* | 0.047* | 0.073 |
| 13 | 0.730* | 0.065* | -0.022* | 0.058* | 0.022* | 0.064 |
| 14 | 0.706* | 0.054* | -0.018* | 0.036* | -0.006* | 0.037 |
| 15 | 0.688* | 0.043* | -0.033* | 0.030* | -0.008* | 0.018 |
| 16 | 0.646* | -0.004* | -0.042* | -0.012* | -0.035* | 0.034 |

Table 12: Results for Llama-2 on TweetEval. Comp./Suff. denote comprehensiveness/sufficiency; KS denotes Kernel SHAP; orig denotes the unpruned model. Significant changes ($p < 0.05$) are marked with *. Wilcoxon's test is used for Comp./Suff., and McNemar's test for accuracy.

| Pruned Layers | Accuracy | Comp. LIME | Suff. LIME | Comp. KS | Suff. KS | ECE |
|---|---|---|---|---|---|---|
| 0 (orig) | 0.482 | 0.072 | 0.133 | 0.086 | 0.128 | 0.100 |
| 1 | 0.458* | 0.038* | 0.048* | 0.046* | 0.041* | 0.099 |
| 2 | 0.458* | 0.016* | 0.055* | 0.031* | 0.041* | 0.098 |
| 3 | 0.472 | -0.017* | -0.000* | -0.002* | -0.005* | 0.050 |
| 4 | 0.476 | 0.017* | 0.123 | 0.042* | 0.125 | 0.107 |
| 5 | 0.470* | 0.021* | 0.091 | 0.038* | 0.110* | 0.115 |
| 6 | 0.471* | 0.032* | 0.086* | 0.043* | 0.115 | 0.171 |
| 7 | 0.469* | 0.122* | 0.094* | 0.109* | 0.116 | 0.165 |
| 8 | 0.464* | 0.110* | 0.095* | 0.103* | 0.112 | 0.135 |
| 9 | 0.459* | 0.056* | 0.054* | 0.057* | 0.070* | 0.111 |
| 10 | 0.474 | 0.131* | 0.137 | 0.109* | 0.168* | 0.191 |
| 11 | 0.444* | 0.160* | 0.060* | 0.097* | 0.080 | 0.152 |
| 12 | 0.442* | 0.052 | -0.035* | 0.027* | -0.029* | 0.078 |
| 13 | 0.494 | 0.188* | 0.067* | 0.097* | 0.081 | 0.054 |
| 14 | 0.356* | 0.081 | -0.057* | 0.022* | -0.059* | 0.185 |
| 15 | 0.498 | 0.139* | 0.072* | 0.063* | 0.067* | 0.052 |
| 16 | 0.530* | 0.006* | -0.045* | 0.043* | -0.008* | 0.031 |

Table 13: Results for Llama-2 on Rotten Tomatoes. Comp./Suff. denote comprehensiveness/sufficiency; KS denotes Kernel SHAP; orig denotes the unpruned model. Significant changes ($p < 0.05$) are marked with *. Wilcoxon's test is used for Comp./Suff., and McNemar's test for accuracy.

| Pruned Layers | Accuracy | Comp. LIME | Suff. LIME | Comp. KS | Suff. KS | ECE |
|---|---|---|---|---|---|---|
| 0 (orig) | 0.623 | 0.092 | 0.128 | 0.105 | 0.136 | 0.080 |
| 1 | 0.780* | 0.046* | 0.038* | 0.044* | 0.030* | 0.192 |
| 2 | 0.779* | 0.043* | 0.030* | 0.041* | 0.019* | 0.193 |
| 3 | 0.768* | 0.047* | 0.037* | 0.044* | 0.027* | 0.180 |
| 4 | 0.714* | 0.044* | 0.098 | 0.052* | 0.098 | 0.104 |
| 5 | 0.701* | 0.046* | 0.099 | 0.054* | 0.114* | 0.088 |
| 6 | 0.662* | 0.049* | 0.060* | 0.057* | 0.071* | 0.053 |
| 7 | 0.705* | 0.059* | 0.057* | 0.061* | 0.064* | 0.095 |
| 8 | 0.717* | 0.052* | 0.041* | 0.054* | 0.048* | 0.119 |
| 9 | 0.733* | 0.034* | 0.025* | 0.035* | 0.034* | 0.143 |
| 10 | 0.555* | 0.083* | 0.171* | 0.101* | 0.181* | 0.145 |
| 11 | 0.641 | 0.100* | 0.127* | 0.105 | 0.139* | 0.066 |
| 12 | 0.751* | -0.067* | -0.110* | -0.053* | -0.083* | 0.164 |
| 13 | 0.635 | -0.153* | -0.213* | -0.110* | -0.174* | 0.067 |
| 14 | 0.583 | -0.111* | -0.261* | -0.093* | -0.208* | 0.122 |
| 15 | 0.587 | -0.052* | -0.193* | -0.056* | -0.159* | 0.070 |
| 16 | 0.520* | 0.072* | -0.132* | 0.026* | -0.099* | 0.188 |

Table 14: Results for Llama-2 on SST-2. Comp./Suff. denote comprehensiveness/sufficiency; KS denotes Kernel SHAP; orig denotes the unpruned model. Significant changes ($p < 0.05$) are marked with *. Wilcoxon's test is used for Comp./Suff., and McNemar's test for accuracy.

| Pruned Layers | Accuracy | Comp. LIME | Suff. LIME | Comp. KS | Suff. KS | ECE |
|---|---|---|---|---|---|---|
| 0 (orig) | 0.494 | 0.108 | 0.052 | 0.084 | 0.002 | 0.250 |
| 1 | 0.491 | 0.200* | 0.097* | 0.168* | 0.060* | 0.307 |
| 2 | 0.491 | 0.220* | 0.101* | 0.186* | 0.063* | 0.307 |
| 3 | 0.491 | 0.227* | 0.093* | 0.192* | 0.055* | 0.315 |
| 4 | 0.491 | 0.233* | 0.074* | 0.202* | 0.036* | 0.318 |
| 5 | 0.491 | 0.271* | 0.062* | 0.233* | 0.033* | 0.318 |
| 6 | 0.491 | 0.176* | 0.046 | 0.139* | -0.001 | 0.325 |
| 7 | 0.491 | 0.139* | -0.027* | 0.106* | -0.040* | 0.316 |
| 8 | 0.491 | 0.123* | -0.081* | 0.086 | -0.076* | 0.323 |
| 9 | 0.491 | 0.178* | -0.090* | 0.144* | -0.089* | 0.313 |
| 10 | 0.498 | 0.080* | -0.141* | 0.040* | -0.112* | 0.240 |
| 11 | 0.491 | 0.023* | -0.106* | 0.022* | -0.063* | 0.322 |
| 12 | 0.491 | -0.012* | -0.110* | 0.001* | -0.069* | 0.297 |
| 13 | 0.491 | 0.004* | -0.057* | 0.039* | -0.027* | 0.359 |
| 14 | 0.491 | 0.110 | -0.006* | 0.111* | 0.029* | 0.375 |
| 15 | 0.491 | 0.112 | -0.011* | 0.088 | -0.002 | 0.347 |
| 16 | 0.491 | 0.246* | 0.115* | 0.191* | 0.132* | 0.391 |

Table 15: Results for Llama-2 on RTE. Comp./Suff. denote comprehensiveness/sufficiency; KS denotes Kernel SHAP; orig denotes the unpruned model. Significant changes ($p < 0.05$) are marked with *. Wilcoxon's test is used for Comp./Suff., and McNemar's test for accuracy.

| Pruned Layers | Accuracy | Comp. LIME | Suff. LIME | Comp. KS | Suff. KS | ECE |
|---|---|---|---|---|---|---|
| 0 (orig) | 0.628 | 0.092 | 0.040 | 0.078 | 0.043 | 0.032 |
| 1 | 0.614 | 0.098* | 0.023* | 0.078 | 0.032* | 0.053 |
| 2 | 0.614 | 0.104* | 0.029* | 0.084* | 0.039 | 0.054 |
| 3 | 0.628 | 0.119* | 0.050* | 0.099* | 0.048 | 0.054 |
| 4 | 0.614 | 0.153* | 0.014* | 0.125* | 0.025* | 0.059 |
| 5 | 0.592 | 0.168* | 0.040* | 0.134* | 0.055* | 0.097 |
| 6 | 0.560* | 0.191* | 0.059* | 0.163* | 0.082* | 0.156 |
| 7 | 0.556* | 0.211* | -0.007* | 0.172* | 0.035 | 0.181 |
| 8 | 0.599 | 0.179* | 0.056* | 0.143* | 0.082* | 0.131 |
| 9 | 0.610 | 0.152* | 0.030 | 0.112* | 0.061* | 0.080 |
| 10 | 0.538* | 0.212* | 0.106* | 0.111* | 0.151* | 0.195 |
| 11 | 0.538* | 0.254* | 0.163* | 0.149* | 0.220* | 0.192 |
| 12 | 0.596 | 0.217* | 0.187* | 0.158* | 0.210* | 0.075 |
| 13 | 0.567* | 0.182* | 0.147* | 0.134* | 0.212* | 0.146 |
| 14 | 0.592 | 0.178* | 0.095* | 0.115* | 0.163* | 0.077 |
| 15 | 0.606 | 0.150* | 0.070* | 0.086 | 0.137* | 0.098 |
| 16 | 0.581 | 0.209* | 0.083* | 0.124* | 0.159* | 0.093 |

Table 16: Results for Mistral on ARC Easy. Comp./Suff. denote comprehensiveness/sufficiency; KS denotes Kernel SHAP; orig denotes the unpruned model. Significant changes ($p < 0.05$) are marked with *. Wilcoxon's test is used for Comp./Suff., and McNemar's test for accuracy.

| Pruned Layers | Accuracy | Comp. LIME | Suff. LIME | Comp. KS | Suff. KS | ECE |
|---|---|---|---|---|---|---|
| 0 (orig) | 0.809 | 0.266 | 0.155 | 0.254 | 0.200 | 0.077 |
| 1 | 0.808 | 0.276* | 0.160* | 0.262* | 0.204 | 0.078 |
| 2 | 0.806 | 0.272* | 0.154 | 0.258* | 0.199 | 0.081 |
| 3 | 0.803 | 0.271* | 0.157* | 0.259* | 0.202 | 0.082 |
| 4 | 0.804 | 0.278* | 0.160* | 0.266* | 0.206 | 0.083 |
| 5 | 0.803 | 0.280* | 0.158 | 0.264 | 0.205 | 0.084 |
| 6 | 0.802 | 0.289* | 0.161 | 0.271* | 0.213* | 0.084 |
| 7 | 0.798* | 0.284* | 0.163 | 0.264 | 0.209* | 0.089 |
| 8 | 0.798* | 0.280* | 0.160 | 0.264* | 0.205* | 0.089 |
| 9 | 0.801 | 0.294* | 0.179* | 0.278* | 0.228* | 0.084 |
| 10 | 0.796* | 0.294* | 0.173* | 0.275* | 0.220* | 0.087 |
| 11 | 0.786* | 0.295* | 0.183* | 0.273* | 0.226* | 0.093 |
| 12 | 0.761* | 0.270* | 0.174* | 0.242 | 0.212* | 0.107 |
| 13 | 0.721* | 0.264 | 0.180* | 0.240 | 0.207* | 0.120 |
| 14 | 0.665* | 0.248 | 0.176* | 0.226* | 0.206 | 0.153 |
| 15 | 0.656* | 0.238 | 0.166* | 0.207* | 0.195 | 0.161 |
| 16 | 0.592* | 0.213* | 0.155 | 0.188* | 0.177 | 0.205 |

Table 17: Results for Mistral on ARC Challenge. Comp./Suff. denote comprehensiveness/sufficiency; KS denotes Kernel SHAP; orig denotes the unpruned model. Significant changes ($p < 0.05$) are marked with *. Wilcoxon's test is used for Comp./Suff., and McNemar's test for accuracy.

| Pruned Layers | Accuracy | Comp. LIME | Suff. LIME | Comp. KS | Suff. KS | ECE |
|---|---|---|---|---|---|---|
| 0 (orig) | 0.504 | 0.278 | 0.141 | 0.257 | 0.175 | 0.313 |
| 1 | 0.500 | 0.273* | 0.136 | 0.254 | 0.169* | 0.315 |
| 2 | 0.504 | 0.269* | 0.136 | 0.255 | 0.171 | 0.310 |
| 3 | 0.503 | 0.266* | 0.135 | 0.246* | 0.167* | 0.311 |
| 4 | 0.501 | 0.271 | 0.144 | 0.251 | 0.177 | 0.314 |
| 5 | 0.492 | 0.260* | 0.135 | 0.243* | 0.166 | 0.324 |
| 6 | 0.496 | 0.263* | 0.141 | 0.244* | 0.171 | 0.319 |
| 7 | 0.486* | 0.256* | 0.139 | 0.244* | 0.171 | 0.327 |
| 8 | 0.484* | 0.257* | 0.147 | 0.243* | 0.177 | 0.331 |
| 9 | 0.480* | 0.254* | 0.148 | 0.240* | 0.183 | 0.336 |
| 10 | 0.477* | 0.244* | 0.142 | 0.229* | 0.175 | 0.336 |
| 11 | 0.483* | 0.246* | 0.142 | 0.224* | 0.177 | 0.333 |
| 12 | 0.464* | 0.231* | 0.137 | 0.210* | 0.169 | 0.346 |
| 13 | 0.451* | 0.204* | 0.127 | 0.186* | 0.159 | 0.342 |
| 14 | 0.433* | 0.188* | 0.124 | 0.174* | 0.151 | 0.359 |
| 15 | 0.423* | 0.187* | 0.114 | 0.164* | 0.155 | 0.364 |
| 16 | 0.373* | 0.148* | 0.091* | 0.126* | 0.120* | 0.413 |

Table 18: Results for Mistral on OpenBookQA. Comp./Suff. denote comprehensiveness/sufficiency; KS denotes Kernel SHAP; orig denotes the unpruned model. Significant changes ($p < 0.05$) are marked with *. Wilcoxon's test is used for Comp./Suff., and McNemar's test for accuracy.

| Pruned Layers | Accuracy | Comp. LIME | Suff. LIME | Comp. KS | Suff. KS | ECE |
|---|---|---|---|---|---|---|
| 0 (orig) | 0.326 | 0.259 | 0.171 | 0.230 | 0.203 | 0.466 |
| 1 | 0.328 | 0.259 | 0.171 | 0.232 | 0.201 | 0.466 |
| 2 | 0.328 | 0.260 | 0.162 | 0.231 | 0.201* | 0.470 |
| 3 | 0.320 | 0.256 | 0.165 | 0.229 | 0.202* | 0.477 |
| 4 | 0.320 | 0.260 | 0.176 | 0.237* | 0.210* | 0.474 |
| 5 | 0.326 | 0.265* | 0.178 | 0.242* | 0.215* | 0.469 |
| 6 | 0.326 | 0.265 | 0.190* | 0.247* | 0.222* | 0.471 |
| 7 | 0.316 | 0.255 | 0.176 | 0.233 | 0.206 | 0.480 |
| 8 | 0.328 | 0.258 | 0.182* | 0.235 | 0.213 | 0.473 |
| 9 | 0.330 | 0.253 | 0.190* | 0.228 | 0.219 | 0.466 |
| 10 | 0.318 | 0.246 | 0.177* | 0.221 | 0.205 | 0.476 |
| 11 | 0.318 | 0.233* | 0.170* | 0.211 | 0.195 | 0.482 |
| 12 | 0.282* | 0.218* | 0.155 | 0.193* | 0.176 | 0.512 |
| 13 | 0.284* | 0.203* | 0.152 | 0.181* | 0.176* | 0.492 |
| 14 | 0.278* | 0.182* | 0.140* | 0.168* | 0.159* | 0.501 |
| 15 | 0.272* | 0.167* | 0.131* | 0.151* | 0.146* | 0.503 |
| 16 | 0.252* | 0.139* | 0.115* | 0.124* | 0.125* | 0.529 |

Table 19: Results for Mistral on BoolQ. Comp./Suff. denote comprehensiveness/sufficiency; KS denotes Kernel SHAP; orig denotes the unpruned model. Significant changes ($p < 0.05$) are marked with *. Wilcoxon's test is used for Comp./Suff., and McNemar's test for accuracy.

| Pruned Layers | Accuracy | Comp. LIME | Suff. LIME | Comp. KS | Suff. KS | ECE |
|---|---|---|---|---|---|---|
| 0 (orig) | 0.836 | 0.344 | 0.145 | 0.315 | 0.175 | 0.074 |
| 1 | 0.836 | 0.347* | 0.150* | 0.325* | 0.178 | 0.067 |
| 2 | 0.839 | 0.342 | 0.153* | 0.323* | 0.181* | 0.072 |
| 3 | 0.835 | 0.333 | 0.154* | 0.313 | 0.185* | 0.068 |
| 4 | 0.835 | 0.334 | 0.154* | 0.317 | 0.182* | 0.068 |
| 5 | 0.833 | 0.326* | 0.158* | 0.320 | 0.180* | 0.066 |
| 6 | 0.829 | 0.318* | 0.159* | 0.314 | 0.178 | 0.063 |
| 7 | 0.831 | 0.317* | 0.157* | 0.304* | 0.177 | 0.068 |
| 8 | 0.830 | 0.316* | 0.157* | 0.303* | 0.180* | 0.065 |
| 9 | 0.831 | 0.305* | 0.154* | 0.286* | 0.180* | 0.065 |
| 10 | 0.824* | 0.286* | 0.153* | 0.270* | 0.178 | 0.067 |
| 11 | 0.785* | 0.219* | 0.139 | 0.236* | 0.162 | 0.043 |
| 12 | 0.772* | 0.195* | 0.140 | 0.251* | 0.158 | 0.027 |
| 13 | 0.731* | 0.149* | 0.105* | 0.181* | 0.112* | 0.022 |
| 14 | 0.691* | 0.123* | 0.104* | 0.174* | 0.102* | 0.034 |
| 15 | 0.661* | 0.104* | 0.102* | 0.178* | 0.088* | 0.064 |
| 16 | 0.501* | -0.003* | 0.053* | 0.127* | 0.004* | 0.220 |

Table 20: Results for Mistral on TweetEval. Comp./Suff. denote comprehensiveness/sufficiency; KS denotes Kernel SHAP; orig denotes the unpruned model. Significant changes ($p < 0.05$) are marked with *. Wilcoxon's test is used for Comp./Suff., and McNemar's test for accuracy.

| Pruned Layers | Accuracy | Comp. LIME | Suff. LIME | Comp. KS | Suff. KS | ECE |
|---|---|---|---|---|---|---|
| 0 (orig) | 0.457 | 0.214 | 0.464 | 0.282 | 0.368 | 0.195 |
| 1 | 0.449* | 0.191* | 0.462* | 0.237* | 0.443* | 0.234 |
| 2 | 0.449* | 0.172* | 0.438* | 0.210* | 0.427* | 0.226 |
| 3 | 0.440* | 0.182* | 0.468* | 0.234* | 0.428* | 0.254 |
| 4 | 0.434* | 0.238* | 0.448* | 0.318* | 0.331* | 0.266 |
| 5 | 0.465* | 0.210 | 0.373* | 0.272* | 0.285* | 0.201 |
| 6 | 0.483* | 0.225* | 0.303* | 0.261* | 0.246* | 0.156 |
| 7 | 0.490* | 0.230* | 0.251* | 0.254* | 0.211* | 0.138 |
| 8 | 0.484* | 0.268* | 0.221* | 0.275 | 0.203* | 0.205 |
| 9 | 0.455 | 0.310* | 0.251* | 0.315* | 0.231* | 0.326 |
| 10 | 0.520* | 0.069* | 0.245* | 0.094* | 0.210* | 0.130 |
| 11 | 0.533* | 0.072* | 0.260* | 0.088* | 0.233* | 0.093 |
| 12 | 0.536* | 0.077* | 0.265* | 0.092* | 0.222* | 0.055 |
| 13 | 0.586* | 0.054* | -0.047* | 0.076* | -0.049* | 0.009 |
| 14 | 0.564* | 0.061* | -0.104* | 0.066* | -0.088* | 0.022 |
| 15 | 0.537* | 0.041* | -0.212* | 0.042* | -0.185* | 0.065 |
| 16 | 0.490 | 0.064* | -0.108* | 0.072* | -0.097* | 0.076 |

Table 21: Results for Mistral on Rotten Tomatoes. Comp./Suff. denote comprehensiveness/sufficiency; KS denotes Kernel SHAP; orig denotes the unpruned model. Significant changes ($p < 0.05$) are marked with *. Wilcoxon's test is used for Comp./Suff., and McNemar's test for accuracy.

| Pruned Layers | Accuracy | Comp. LIME | Suff. LIME | Comp. KS | Suff. KS | ECE |
|---|---|---|---|---|---|---|
| 0 (orig) | 0.858 | 0.097 | 0.111 | 0.097 | 0.119 | 0.226 |
| 1 | 0.777* | 0.100 | 0.109 | 0.098 | 0.089* | 0.137 |
| 2 | 0.791* | 0.104 | 0.115 | 0.101 | 0.099* | 0.154 |
| 3 | 0.735* | 0.106 | 0.096 | 0.087 | 0.088* | 0.085 |
| 4 | 0.747* | 0.079 | 0.033* | 0.079 | 0.037* | 0.104 |
| 5 | 0.784* | 0.075 | 0.048* | 0.076* | 0.048* | 0.156 |
| 6 | 0.839 | 0.091 | 0.080* | 0.092 | 0.081* | 0.218 |
| 7 | 0.847 | 0.099 | 0.088* | 0.097 | 0.089* | 0.225 |
| 8 | 0.803* | 0.098 | 0.076* | 0.092 | 0.072* | 0.164 |
| 9 | 0.756* | 0.093 | 0.061* | 0.087 | 0.059* | 0.107 |
| 10 | 0.761* | 0.205* | 0.200* | 0.206* | 0.184* | 0.104 |
| 11 | 0.785* | 0.228* | 0.194* | 0.212* | 0.172* | 0.088 |
| 12 | 0.815* | 0.192* | 0.155* | 0.175* | 0.144* | 0.145 |
| 13 | 0.688* | 0.033* | 0.060* | 0.043* | 0.060* | 0.089 |
| 14 | 0.545* | 0.016* | 0.055* | 0.050* | 0.049* | 0.238 |
| 15 | 0.623* | 0.055* | 0.054* | 0.076 | 0.054* | 0.137 |
| 16 | 0.500* | 0.026* | 0.069* | 0.076* | 0.059* | 0.275 |

Table 22: Results for Mistral on SST-2. Comp./Suff. denote comprehensiveness/sufficiency; KS denotes Kernel SHAP; orig denotes the unpruned model. Significant changes ($p < 0.05$) are marked with *. Wilcoxon's test is used for Comp./Suff., and McNemar's test for accuracy.

| Pruned Layers | Accuracy | Comp. LIME | Suff. LIME | Comp. KS | Suff. KS | ECE |
|---|---|---|---|---|---|---|
| 0 (orig) | 0.689 | 0.110 | 0.126 | 0.111 | 0.137 | 0.105 |
| 1 | 0.790* | 0.049* | 0.029* | 0.046* | 0.029* | 0.213 |
| 2 | 0.780* | 0.051* | 0.028* | 0.043* | 0.035* | 0.204 |
| 3 | 0.783* | 0.045* | 0.027* | 0.035* | 0.037* | 0.203 |
| 4 | 0.769* | 0.042* | 0.006* | 0.024* | 0.005* | 0.187 |
| 5 | 0.751* | 0.043* | 0.039* | 0.036* | 0.038* | 0.177 |
| 6 | 0.683 | 0.094* | 0.110* | 0.092* | 0.115* | 0.101 |
| 7 | 0.710 | 0.086* | 0.094* | 0.084* | 0.096* | 0.130 |
| 8 | 0.753* | 0.047* | 0.029* | 0.038* | 0.026* | 0.175 |
| 9 | 0.769* | 0.033* | -0.004* | 0.017* | -0.005* | 0.178 |
| 10 | 0.570* | 0.275* | 0.254* | 0.247* | 0.257* | 0.174 |
| 11 | 0.532* | 0.314* | 0.269* | 0.266* | 0.291* | 0.208 |
| 12 | 0.573* | 0.269* | 0.217* | 0.219* | 0.228* | 0.183 |
| 13 | 0.789* | 0.028* | 0.008* | 0.035* | 0.014* | 0.199 |
| 14 | 0.644 | 0.041* | -0.009* | 0.062* | -0.014* | 0.079 |
| 15 | 0.686 | 0.052* | -0.006* | 0.070* | -0.015* | 0.073 |
| 16 | 0.517* | 0.095 | 0.005* | 0.132 | -0.010* | 0.188 |

Table 23: Results for Mistral on RTE. Comp./Suff. denote comprehensiveness/sufficiency; KS denotes Kernel SHAP; orig denotes the unpruned model. Significant changes ($p < 0.05$) are marked with *. Wilcoxon's test is used for Comp./Suff., and McNemar's test for accuracy.

| Pruned Layers | Accuracy | Comp. LIME | Suff. LIME | Comp. KS | Suff. KS | ECE |
|---|---|---|---|---|---|---|
| 0 (orig) | 0.679 | 0.152 | 0.124 | 0.118 | 0.111 | 0.085 |
| 1 | 0.661 | 0.159 | 0.119* | 0.120 | 0.102* | 0.094 |
| 2 | 0.671 | 0.158 | 0.128* | 0.121 | 0.116 | 0.075 |
| 3 | 0.675 | 0.161* | 0.125 | 0.124 | 0.112 | 0.075 |
| 4 | 0.661 | 0.173* | 0.117* | 0.130* | 0.098* | 0.102 |
| 5 | 0.653* | 0.164* | 0.114* | 0.126* | 0.092* | 0.108 |
| 6 | 0.668 | 0.155 | 0.113* | 0.118 | 0.086* | 0.088 |
| 7 | 0.671 | 0.154 | 0.117* | 0.113 | 0.091* | 0.079 |
| 8 | 0.675 | 0.157 | 0.120* | 0.112 | 0.093* | 0.078 |
| 9 | 0.661 | 0.170* | 0.117* | 0.131 | 0.091* | 0.118 |
| 10 | 0.664 | 0.198* | 0.142* | 0.172* | 0.124* | 0.138 |
| 11 | 0.646* | 0.167* | 0.118 | 0.118 | 0.085* | 0.150 |
| 12 | 0.664 | 0.149 | 0.115* | 0.107 | 0.082* | 0.107 |
| 13 | 0.653 | 0.095* | 0.095* | 0.049* | 0.055* | 0.056 |
| 14 | 0.664 | 0.142 | 0.112* | 0.099* | 0.084* | 0.092 |
| 15 | 0.704 | 0.143 | 0.153* | 0.110 | 0.117 | 0.061 |
| 16 | 0.668 | 0.134 | 0.116 | 0.080* | 0.060* | 0.059 |

Table 24: Results for Llama-3 on ARC Easy. Comp./Suff. denote comprehensiveness/sufficiency; KS denotes Kernel SHAP; orig denotes the unpruned model. Significant changes ($p < 0.05$) are marked with *. Wilcoxon's test is used for Comp./Suff., and McNemar's test for accuracy.

| Pruned Layers | Accuracy | Comp. LIME | Suff. LIME | Comp. KS | Suff. KS | ECE |
|---|---|---|---|---|---|---|
| 0 (orig) | 0.815 | 0.200 | 0.126 | 0.172 | 0.157 | 0.076 |
| 1 | 0.814 | 0.165* | 0.153* | 0.162 | 0.162 | 0.077 |
| 2 | 0.813 | 0.189 | 0.136 | 0.170 | 0.162 | 0.078 |
| 3 | 0.814 | 0.203 | 0.130 | 0.181 | 0.157 | 0.077 |
| 4 | 0.816 | 0.203 | 0.126 | 0.178 | 0.156 | 0.072 |
| 5 | 0.819 | 0.205 | 0.124 | 0.174 | 0.158 | 0.069 |
| 6 | 0.817 | 0.199 | 0.117 | 0.174 | 0.147 | 0.073 |
| 7 | 0.813 | 0.199 | 0.121 | 0.176 | 0.147 | 0.071 |
| 8 | 0.806* | 0.197 | 0.121 | 0.176 | 0.150 | 0.076 |
| 9 | 0.806 | 0.210 | 0.137 | 0.185 | 0.160 | 0.077 |
| 10 | 0.798* | 0.204 | 0.129 | 0.180 | 0.153 | 0.080 |
| 11 | 0.790* | 0.219* | 0.147 | 0.190* | 0.177* | 0.084 |
| 12 | 0.781* | 0.207 | 0.132 | 0.190* | 0.165 | 0.092 |
| 13 | 0.780* | 0.196 | 0.120 | 0.178 | 0.152 | 0.093 |
| 14 | 0.738* | 0.179 | 0.128 | 0.163 | 0.156 | 0.116 |
| 15 | 0.707* | 0.157* | 0.101 | 0.143 | 0.129 | 0.140 |
| 16 | 0.564* | 0.123* | 0.070* | 0.106* | 0.094* | 0.232 |

Table 25: Results for Llama-3 on ARC Challenge. Comp./Suff. denote comprehensiveness/sufficiency; KS denotes Kernel SHAP; orig denotes the unpruned model. Significant changes ($p < 0.05$) are marked with *. Wilcoxon's test is used for Comp./Suff., and McNemar's test for accuracy.

| Pruned Layers | Accuracy | Comp. LIME | Suff. LIME | Comp. KS | Suff. KS | ECE |
|---|---|---|---|---|---|---|
| 0 (orig) | 0.513 | 0.222 | 0.092 | 0.202 | 0.119 | 0.306 |
| 1 | 0.504 | 0.209 | 0.109* | 0.191 | 0.133* | 0.313 |
| 2 | 0.503 | 0.227 | 0.102 | 0.202 | 0.130 | 0.316 |
| 3 | 0.510 | 0.220 | 0.094 | 0.202 | 0.121 | 0.313 |
| 4 | 0.509 | 0.229 | 0.102 | 0.206 | 0.129 | 0.313 |
| 5 | 0.509 | 0.237 | 0.115 | 0.217 | 0.143* | 0.304 |
| 6 | 0.508 | 0.230 | 0.101 | 0.207 | 0.131 | 0.312 |
| 7 | 0.499 | 0.237 | 0.109 | 0.213 | 0.145* | 0.316 |
| 8 | 0.497 | 0.239 | 0.113 | 0.208 | 0.148* | 0.319 |
| 9 | 0.498 | 0.234 | 0.112 | 0.202 | 0.145 | 0.315 |
| 10 | 0.485* | 0.229 | 0.115* | 0.196 | 0.142* | 0.325 |
| 11 | 0.474* | 0.222 | 0.109 | 0.193 | 0.133 | 0.332 |
| 12 | 0.475* | 0.209 | 0.100 | 0.182 | 0.123 | 0.328 |
| 13 | 0.495 | 0.209 | 0.112 | 0.194 | 0.145 | 0.323 |
| 14 | 0.451* | 0.172* | 0.090 | 0.157* | 0.108 | 0.356 |
| 15 | 0.426* | 0.187 | 0.110 | 0.167* | 0.127 | 0.378 |
| 16 | 0.331* | 0.147* | 0.085 | 0.117* | 0.101 | 0.458 |

Table 26: Results for Llama-3 on OpenBookQA. Comp./Suff. denote comprehensiveness/sufficiency; KS denotes Kernel SHAP; orig denotes the unpruned model. Significant changes ($p < 0.05$) are marked with *. Wilcoxon's test is used for Comp./Suff., and McNemar's test for accuracy.

| Pruned Layers | Accuracy | Comp. LIME | Suff. LIME | Comp. KS | Suff. KS | ECE |
|---|---|---|---|---|---|---|
| 0 (orig) | 0.332 | 0.237 | 0.139 | 0.214 | 0.156 | 0.473 |
| 1 | 0.338 | 0.176* | 0.193* | 0.156* | 0.203* | 0.467 |
| 2 | 0.340 | 0.222* | 0.164* | 0.199* | 0.189* | 0.466 |
| 3 | 0.340 | 0.245 | 0.153 | 0.223 | 0.184* | 0.465 |
| 4 | 0.346 | 0.244 | 0.150* | 0.222 | 0.178* | 0.458 |
| 5 | 0.340 | 0.247 | 0.154 | 0.224 | 0.178 | 0.461 |
| 6 | 0.350 | 0.248 | 0.156 | 0.224 | 0.180* | 0.449 |
| 7 | 0.354 | 0.234 | 0.151 | 0.214 | 0.175 | 0.443 |
| 8 | 0.348 | 0.235 | 0.163* | 0.215 | 0.183* | 0.452 |
| 9 | 0.354 | 0.234 | 0.160* | 0.215 | 0.184* | 0.446 |
| 10 | 0.338 | 0.217* | 0.155 | 0.200* | 0.170 | 0.457 |
| 11 | 0.324 | 0.202* | 0.141 | 0.188* | 0.154 | 0.466 |
| 12 | 0.310 | 0.195* | 0.136 | 0.177* | 0.159 | 0.483 |
| 13 | 0.324 | 0.195* | 0.140 | 0.177* | 0.165 | 0.476 |
| 14 | 0.302 | 0.174* | 0.134 | 0.163* | 0.158 | 0.486 |
| 15 | 0.288* | 0.149* | 0.105* | 0.140* | 0.121* | 0.494 |
| 16 | 0.220* | 0.111* | 0.068* | 0.105* | 0.086* | 0.562 |

Table 27: Results for Llama-3 on BoolQ. Comp./Suff. denote comprehensiveness/sufficiency; KS denotes Kernel SHAP; orig denotes the unpruned model. Significant changes ($p < 0.05$) are marked with *. Wilcoxon's test is used for Comp./Suff., and McNemar's test for accuracy.

| Pruned Layers | Accuracy | Comp. LIME | Suff. LIME | Comp. KS | Suff. KS | ECE |
|---|---|---|---|---|---|---|
| 0 (orig) | 0.821 | 0.384 | 0.127 | 0.272 | 0.188 | 0.060 |
| 1 | 0.820 | 0.088* | 0.349* | 0.128* | 0.313* | 0.059 |
| 2 | 0.820 | 0.272* | 0.197* | 0.230* | 0.224* | 0.057 |
| 3 | 0.821 | 0.369* | 0.126 | 0.284 | 0.180 | 0.056 |
| 4 | 0.822 | 0.342* | 0.125 | 0.266 | 0.177* | 0.062 |
| 5 | 0.818 | 0.335* | 0.122 | 0.252 | 0.176* | 0.063 |
| 6 | 0.818 | 0.325* | 0.122 | 0.248* | 0.170* | 0.059 |
| 7 | 0.822 | 0.339* | 0.123 | 0.252 | 0.174* | 0.062 |
| 8 | 0.823 | 0.336* | 0.116* | 0.247* | 0.156* | 0.070 |
| 9 | 0.818 | 0.336* | 0.120 | 0.251* | 0.150* | 0.065 |
| 10 | 0.818 | 0.338* | 0.129 | 0.260 | 0.156* | 0.059 |
| 11 | 0.821 | 0.338* | 0.132 | 0.277 | 0.163* | 0.057 |
| 12 | 0.800* | 0.333* | 0.123 | 0.233* | 0.155* | 0.055 |
| 13 | 0.752* | 0.309* | 0.098* | 0.194* | 0.135* | 0.032 |
| 14 | 0.762* | 0.295* | 0.102* | 0.199* | 0.136* | 0.051 |
| 15 | 0.774* | 0.372 | 0.088* | 0.268 | 0.128* | 0.065 |
| 16 | 0.807* | 0.373 | 0.110* | 0.267 | 0.158* | 0.121 |

Table 28: Results for Llama-3 on TweetEval. Comp./Suff. denote comprehensiveness/sufficiency; KS denotes Kernel SHAP; orig denotes the unpruned model. Significant changes ($p < 0.05$) are marked with *. Wilcoxon's test is used for Comp./Suff., and McNemar's test for accuracy.

| Pruned Layers | Accuracy | Comp. LIME | Suff. LIME | Comp. KS | Suff. KS | ECE |
|---|---|---|---|---|---|---|
| 0 (orig) | 0.520 | 0.016 | 0.166 | 0.079 | 0.105 | 0.098 |
| 1 | 0.534* | 0.036* | 0.179 | 0.094* | 0.122* | 0.077 |
| 2 | 0.543* | 0.034* | 0.175 | 0.091 | 0.123* | 0.046 |
| 3 | 0.542* | 0.035* | 0.160 | 0.089* | 0.111* | 0.031 |
| 4 | 0.558* | 0.032* | 0.156 | 0.087 | 0.092 | 0.024 |
| 5 | 0.632* | 0.026 | 0.021* | 0.068 | -0.024* | 0.073 |
| 6 | 0.624* | 0.032* | 0.071* | 0.078 | 0.025* | 0.070 |
| 7 | 0.603* | 0.219* | 0.192 | 0.236* | 0.163* | 0.064 |
| 8 | 0.624* | 0.217* | 0.191 | 0.231* | 0.165* | 0.082 |
| 9 | 0.387* | 0.295* | 0.267* | 0.313* | 0.248* | 0.304 |
| 10 | 0.423* | 0.267* | 0.226* | 0.293* | 0.216* | 0.225 |
| 11 | 0.433* | 0.214* | 0.187 | 0.241* | 0.181* | 0.154 |
| 12 | 0.398* | 0.247* | 0.167 | 0.268* | 0.167* | 0.265 |
| 13 | 0.420* | 0.284* | 0.271* | 0.272* | 0.187* | 0.267 |
| 14 | 0.406* | 0.252* | 0.194 | 0.213* | 0.147* | 0.201 |
| 15 | 0.421* | 0.245* | 0.204* | 0.239* | 0.176* | 0.209 |
| 16 | 0.350* | 0.272* | 0.223* | 0.241* | 0.197* | 0.282 |

Table 29: Results for Llama-3 on Rotten Tomatoes. Comp./Suff. denote comprehensiveness/sufficiency; KS denotes Kernel SHAP; orig denotes the unpruned model. Significant changes ($p < 0.05$) are marked with *. Wilcoxon's test is used for Comp./Suff., and McNemar's test for accuracy.

| Pruned Layers | Accuracy | Comp. LIME | Suff. LIME | Comp. KS | Suff. KS | ECE |
|---|---|---|---|---|---|---|
| 0 (orig) | 0.818 | 0.106 | 0.093 | 0.091 | 0.105 | 0.201 |
| 1 | 0.811 | 0.097 | 0.082 | 0.087 | 0.088 | 0.188 |
| 2 | 0.757* | 0.086 | 0.059* | 0.080 | 0.054* | 0.124 |
| 3 | 0.729* | 0.091 | 0.066 | 0.088 | 0.054* | 0.102 |
| 4 | 0.727* | 0.086 | 0.059* | 0.088 | 0.048* | 0.082 |
| 5 | 0.826 | 0.109 | 0.110* | 0.102* | 0.116* | 0.201 |
| 6 | 0.814 | 0.098* | 0.110* | 0.093 | 0.112 | 0.205 |
| 7 | 0.759* | 0.105 | 0.117* | 0.100 | 0.125* | 0.147 |
| 8 | 0.792* | 0.111 | 0.104 | 0.095 | 0.112 | 0.189 |
| 9 | 0.732* | 0.013* | 0.012* | 0.012* | 0.007* | 0.131 |
| 10 | 0.725* | 0.009* | 0.006* | 0.007* | -0.009* | 0.100 |
| 11 | 0.787* | 0.054* | 0.062* | 0.060* | 0.050* | 0.188 |
| 12 | 0.765* | 0.081* | 0.079 | 0.088 | 0.079* | 0.176 |
| 13 | 0.555* | 0.267* | 0.181* | 0.251* | 0.204* | 0.207 |
| 14 | 0.555* | 0.249* | 0.219* | 0.252* | 0.231* | 0.200 |
| 15 | 0.562* | 0.151* | 0.170* | 0.189* | 0.198* | 0.191 |
| 16 | 0.531* | 0.163* | 0.174* | 0.208* | 0.209* | 0.224 |

Table 30: Results for Llama-3 on SST-2. Comp./Suff. denote comprehensiveness/sufficiency; KS denotes Kernel SHAP; orig denotes the unpruned model. Significant changes ($p < 0.05$) are marked with *. Wilcoxon's test is used for Comp./Suff., and McNemar's test for accuracy.

| Pruned Layers | Accuracy | Comp. LIME | Suff. LIME | Comp. KS | Suff. KS | ECE |
|---|---|---|---|---|---|---|
| 0 (orig) | 0.776 | 0.105 | 0.097 | 0.095 | 0.104 | 0.158 |
| 1 | 0.819* | 0.092* | 0.090 | 0.091 | 0.086 | 0.205 |
| 2 | 0.792 | 0.064* | 0.053* | 0.075* | 0.042* | 0.176 |
| 3 | 0.805 | 0.067* | 0.061* | 0.080 | 0.049* | 0.189 |
| 4 | 0.799 | 0.062* | 0.051* | 0.079* | 0.042* | 0.187 |
| 5 | 0.685* | 0.156* | 0.122* | 0.121* | 0.134* | 0.055 |
| 6 | 0.747* | 0.110 | 0.103 | 0.092 | 0.102* | 0.120 |
| 7 | 0.720* | 0.148* | 0.132* | 0.118* | 0.128* | 0.086 |
| 8 | 0.763 | 0.123* | 0.107 | 0.101 | 0.102 | 0.148 |
| 9 | 0.778 | 0.118* | 0.099 | 0.098 | 0.097 | 0.171 |
| 10 | 0.818* | 0.068* | 0.064* | 0.070* | 0.059* | 0.210 |
| 11 | 0.771 | 0.092 | 0.082* | 0.084 | 0.072* | 0.168 |
| 12 | 0.690* | 0.160* | 0.135* | 0.146* | 0.136* | 0.058 |
| 13 | 0.507* | 0.343* | 0.234* | 0.251* | 0.220* | 0.280 |
| 14 | 0.495* | 0.334* | 0.257* | 0.266* | 0.253* | 0.315 |
| 15 | 0.493* | 0.237* | 0.217* | 0.236* | 0.249* | 0.339 |
| 16 | 0.491* | 0.238* | 0.248* | 0.278* | 0.299* | 0.383 |

Table 31: Results for Llama-3 on RTE. Comp./Suff. denote comprehensiveness/sufficiency; KS denotes Kernel SHAP; orig denotes the unpruned model. Significant changes ($p < 0.05$) are marked with *. Wilcoxon's test is used for Comp./Suff., and McNemar's test for accuracy.

| Pruned Layers | Accuracy | Comp. LIME | Suff. LIME | Comp. KS | Suff. KS | ECE |
|---|---|---|---|---|---|---|
| 0 (orig) | 0.697 | 0.141 | 0.143 | 0.144 | 0.153 | 0.045 |
| 1 | 0.700 | 0.078* | 0.105* | 0.104* | 0.116* | 0.039 |
| 2 | 0.704 | 0.112* | 0.119* | 0.129 | 0.127* | 0.045 |
| 3 | 0.697 | 0.141 | 0.131* | 0.154* | 0.137 | 0.032 |
| 4 | 0.700 | 0.135 | 0.118* | 0.139 | 0.106* | 0.092 |
| 5 | 0.700 | 0.135 | 0.109* | 0.144 | 0.102* | 0.067 |
| 6 | 0.668 | 0.151 | 0.102* | 0.157* | 0.093* | 0.079 |
| 7 | 0.661 | 0.132 | 0.089* | 0.144 | 0.084* | 0.039 |
| 8 | 0.704 | 0.167* | 0.117* | 0.164* | 0.121* | 0.057 |
| 9 | 0.708 | 0.154 | 0.136 | 0.153 | 0.137* | 0.039 |
| 10 | 0.708 | 0.142 | 0.132 | 0.146 | 0.140* | 0.056 |
| 11 | 0.708 | 0.116* | 0.123* | 0.123* | 0.122* | 0.061 |
| 12 | 0.686 | 0.130 | 0.108* | 0.138 | 0.120* | 0.047 |
| 13 | 0.722 | 0.172* | 0.119* | 0.155 | 0.123* | 0.082 |
| 14 | 0.718 | 0.182* | 0.122* | 0.166* | 0.124* | 0.079 |
| 15 | 0.711 | 0.213* | 0.144 | 0.196* | 0.143 | 0.064 |
| 16 | 0.700 | 0.224* | 0.170* | 0.203* | 0.166* | 0.055 |

Table 32: Results for Qwen-3 8B on ARC Easy. Comp./Suff. denote comprehensiveness/sufficiency; KS denotes Kernel SHAP; orig denotes the unpruned model. Significant changes ($p < 0.05$) are marked with *. Wilcoxon's test is used for Comp./Suff., and McNemar's test for accuracy.

| Pruned Layers | Accuracy | Comp. LIME | Suff. LIME | Comp. KS | Suff. KS | ECE |
|---|---|---|---|---|---|---|
| 0 (orig) | 0.835 | 0.232 | 0.155 | 0.184 | 0.214 | 0.106 |
| 1 | 0.838 | 0.238 | 0.146 | 0.180 | 0.205 | 0.088 |
| 2 | 0.834 | 0.233 | 0.153 | 0.182 | 0.207 | 0.088 |
| 3 | 0.822* | 0.221 | 0.149 | 0.177 | 0.203 | 0.093 |
| 4 | 0.811* | 0.234 | 0.155 | 0.192 | 0.199 | 0.095 |
| 5 | 0.806* | 0.245 | 0.161 | 0.196 | 0.204 | 0.099 |
| 6 | 0.790* | 0.248 | 0.176* | 0.206 | 0.204 | 0.113 |
| 7 | 0.785* | 0.241 | 0.165 | 0.202 | 0.202 | 0.121 |
| 8 | 0.778* | 0.253 | 0.177 | 0.214* | 0.216 | 0.122 |
| 9 | 0.769* | 0.237 | 0.172 | 0.198 | 0.203 | 0.132 |
| 10 | 0.753* | 0.235 | 0.184 | 0.202* | 0.211 | 0.141 |
| 11 | 0.636* | 0.171* | 0.140 | 0.139* | 0.170* | 0.238 |
| 12 | 0.632* | 0.211 | 0.181 | 0.184 | 0.207 | 0.253 |
| 13 | 0.645* | 0.209 | 0.175 | 0.178 | 0.203 | 0.236 |
| 14 | 0.624* | 0.176* | 0.142 | 0.148* | 0.165* | 0.247 |
| 15 | 0.609* | 0.175* | 0.135* | 0.144* | 0.164* | 0.256 |
| 16 | 0.576* | 0.197 | 0.164 | 0.164 | 0.180* | 0.293 |

Table 33: Results for Qwen-3 8B on ARC Challenge. Comp./Suff. denote comprehensiveness/sufficiency; KS denotes Kernel SHAP; orig denotes the unpruned model. Significant changes ($p < 0.05$) are marked with *. Wilcoxon's test is used for Comp./Suff., and McNemar's test for accuracy.

| Pruned Layers | Accuracy | Comp. LIME | Suff. LIME | Comp. KS | Suff. KS | ECE |
|---|---|---|---|---|---|---|
| 0 (orig) | 0.558 | 0.273 | 0.122 | 0.189 | 0.200 | 0.333 |
| 1 | 0.549 | 0.273 | 0.110* | 0.183 | 0.191 | 0.322 |
| 2 | 0.549 | 0.277 | 0.105* | 0.185 | 0.187 | 0.320 |
| 3 | 0.538* | 0.270 | 0.093* | 0.186 | 0.171* | 0.324 |
| 4 | 0.530* | 0.268 | 0.098* | 0.192 | 0.166* | 0.331 |
| 5 | 0.536* | 0.244 | 0.101 | 0.172 | 0.164* | 0.324 |
| 6 | 0.520* | 0.250 | 0.097* | 0.183 | 0.155* | 0.337 |
| 7 | 0.498* | 0.222* | 0.087* | 0.158 | 0.144* | 0.353 |
| 8 | 0.497* | 0.227* | 0.088* | 0.168 | 0.133* | 0.354 |
| 9 | 0.500* | 0.233* | 0.107 | 0.182 | 0.138* | 0.356 |
| 10 | 0.474* | 0.254 | 0.119 | 0.194 | 0.160* | 0.384 |
| 11 | 0.405* | 0.191* | 0.123 | 0.155 | 0.152* | 0.460 |
| 12 | 0.416* | 0.210* | 0.142 | 0.179 | 0.183 | 0.450 |
| 13 | 0.414* | 0.189* | 0.128 | 0.159 | 0.163* | 0.445 |
| 14 | 0.403* | 0.184* | 0.135 | 0.155 | 0.162* | 0.445 |
| 15 | 0.387* | 0.187* | 0.131 | 0.148* | 0.166* | 0.470 |
| 16 | 0.373* | 0.183* | 0.145 | 0.156 | 0.160* | 0.485 |

Table 34: Results for Qwen-3 8B on OpenBookQA. Comp./Suff. denote comprehensiveness/sufficiency; KS denotes Kernel SHAP; orig denotes the unpruned model. Significant changes ($p < 0.05$) are marked with *. Wilcoxon's test is used for Comp./Suff., and McNemar's test for accuracy.

| Pruned Layers | Accuracy | Comp. LIME | Suff. LIME | Comp. KS | Suff. KS | ECE |
|---|---|---|---|---|---|---|
| 0 (orig) | 0.312 | 0.214 | 0.136 | 0.188 | 0.181 | 0.507 |
| 1 | 0.296 | 0.213 | 0.119 | 0.163* | 0.172 | 0.520 |
| 2 | 0.294 | 0.213 | 0.119 | 0.162* | 0.170 | 0.518 |
| 3 | 0.290 | 0.199 | 0.101* | 0.149* | 0.151 | 0.518 |
| 4 | 0.288 | 0.200 | 0.119 | 0.157* | 0.159 | 0.518 |
| 5 | 0.274* | 0.197 | 0.112 | 0.151* | 0.149 | 0.540 |
| 6 | 0.274* | 0.200 | 0.106 | 0.154* | 0.147 | 0.538 |
| 7 | 0.280* | 0.186 | 0.085* | 0.144* | 0.127* | 0.523 |
| 8 | 0.290 | 0.171* | 0.086* | 0.139* | 0.129* | 0.518 |
| 9 | 0.292 | 0.175* | 0.108 | 0.144* | 0.136* | 0.522 |
| 10 | 0.286 | 0.160* | 0.100 | 0.137* | 0.132* | 0.533 |
| 11 | 0.264* | 0.116* | 0.092* | 0.116* | 0.102* | 0.563 |
| 12 | 0.272 | 0.134* | 0.093* | 0.124* | 0.109* | 0.559 |
| 13 | 0.246* | 0.138* | 0.084* | 0.119* | 0.100* | 0.578 |
| 14 | 0.244* | 0.149* | 0.097* | 0.130* | 0.117* | 0.590 |
| 15 | 0.250* | 0.172* | 0.107 | 0.157* | 0.121* | 0.588 |
| 16 | 0.242* | 0.151* | 0.086* | 0.131* | 0.100* | 0.592 |

Table 35: Results for Qwen-3 8B on BoolQ. Comp./Suff. denote comprehensiveness/sufficiency; KS denotes Kernel SHAP; orig denotes the unpruned model. Significant changes ($p < 0.05$) are marked with *. Wilcoxon's test is used for Comp./Suff., and McNemar's test for accuracy.

| Pruned Layers | Accuracy | Comp. LIME | Suff. LIME | Comp. KS | Suff. KS | ECE |
|---|---|---|---|---|---|---|
| 0 (orig) | 0.866 | 0.603 | 0.211 | 0.439 | 0.343 | 0.084 |
| 1 | 0.863 | 0.596 | 0.151* | 0.464* | 0.328 | 0.081 |
| 2 | 0.862 | 0.574* | 0.172* | 0.479* | 0.321* | 0.077 |
| 3 | 0.864 | 0.553* | 0.174* | 0.485* | 0.308* | 0.074 |
| 4 | 0.862 | 0.545* | 0.174* | 0.500* | 0.301* | 0.070 |
| 5 | 0.865 | 0.572* | 0.205 | 0.500* | 0.339 | 0.083 |
| 6 | 0.861 | 0.566* | 0.211 | 0.514* | 0.319* | 0.080 |
| 7 | 0.857* | 0.552* | 0.216 | 0.529* | 0.323* | 0.087 |
| 8 | 0.857* | 0.546* | 0.232* | 0.513* | 0.322* | 0.080 |
| 9 | 0.859* | 0.513* | 0.269* | 0.470* | 0.349 | 0.071 |
| 10 | 0.860 | 0.492* | 0.298* | 0.463* | 0.361* | 0.070 |
| 11 | 0.814* | 0.360* | 0.393* | 0.398* | 0.397* | 0.082 |
| 12 | 0.811* | 0.410* | 0.387* | 0.403* | 0.389* | 0.077 |
| 13 | 0.781* | 0.372* | 0.327* | 0.343* | 0.332 | 0.043 |
| 14 | 0.746* | 0.381* | 0.318* | 0.363* | 0.326 | 0.070 |
| 15 | 0.748* | 0.379* | 0.319* | 0.348* | 0.323 | 0.064 |
| 16 | 0.729* | 0.396* | 0.323* | 0.350* | 0.341 | 0.087 |

Table 36: Results for Qwen-3 8B on TweetEval. Comp./Suff. denote comprehensiveness/sufficiency; KS denotes Kernel SHAP; orig denotes the unpruned model. Significant changes ($p < 0.05$) are marked with *. Wilcoxon's test is used for Comp./Suff., and McNemar's test for accuracy.

| Pruned Layers | Accuracy | Comp. LIME | Suff. LIME | Comp. KS | Suff. KS | ECE |
|---|---|---|---|---|---|---|
| 0 (orig) | 0.656 | 0.228 | 0.230 | 0.218 | 0.276 | 0.280 |
| 1 | 0.675* | 0.227* | 0.204 | 0.217* | 0.218* | 0.251 |
| 2 | 0.673* | 0.251* | 0.274* | 0.252* | 0.243 | 0.239 |
| 3 | 0.667* | 0.236* | 0.272* | 0.237* | 0.244 | 0.228 |
| 4 | 0.650 | 0.300* | 0.408* | 0.317* | 0.316 | 0.243 |
| 5 | 0.548* | 0.350* | 0.609* | 0.376* | 0.546* | 0.354 |
| 6 | 0.545* | 0.355* | 0.619* | 0.393* | 0.583* | 0.366 |
| 7 | 0.588* | 0.297* | 0.447* | 0.318* | 0.406* | 0.282 |
| 8 | 0.624* | 0.173* | 0.290* | 0.200* | 0.237* | 0.266 |
| 9 | 0.642 | 0.104* | 0.100* | 0.103* | 0.112* | 0.147 |
| 10 | 0.613* | 0.186* | 0.231* | 0.195 | 0.219* | 0.174 |
| 11 | 0.615* | 0.141* | 0.135* | 0.139* | 0.140* | 0.210 |
| 12 | 0.636* | 0.100* | 0.072* | 0.099* | 0.081* | 0.203 |
| 13 | 0.592* | -0.145* | -0.101* | -0.132* | -0.110* | 0.208 |
| 14 | 0.584* | -0.028* | 0.024* | 0.004* | 0.030* | 0.163 |
| 15 | 0.560* | -0.002* | 0.033* | 0.001* | 0.036* | 0.182 |
| 16 | 0.559* | -0.013* | -0.024* | 0.027* | 0.017* | 0.210 |

Table 37: Results for Qwen-3 8B on Rotten Tomatoes. Comp./Suff. denote comprehensiveness/sufficiency; KS denotes Kernel SHAP; orig denotes the unpruned model. Significant changes ($p < 0.05$) are marked with *. Wilcoxon's test is used for Comp./Suff., and McNemar's test for accuracy.

| Pruned Layers | Accuracy | Comp. LIME | Suff. LIME | Comp. KS | Suff. KS | ECE |
|---|---|---|---|---|---|---|
| 0 (orig) | 0.883 | 0.347 | 0.367 | 0.354 | 0.383 | 0.090 |
| 1 | 0.873* | 0.350 | 0.376* | 0.347 | 0.380 | 0.094 |
| 2 | 0.868* | 0.334* | 0.368 | 0.337* | 0.372 | 0.097 |
| 3 | 0.850* | 0.349 | 0.353 | 0.358* | 0.352* | 0.108 |
| 4 | 0.845* | 0.335* | 0.376 | 0.359 | 0.348* | 0.116 |
| 5 | 0.877 | 0.342 | 0.360 | 0.397* | 0.354* | 0.092 |
| 6 | 0.877 | 0.351 | 0.360 | 0.400* | 0.361* | 0.094 |
| 7 | 0.871 | 0.377* | 0.344* | 0.380* | 0.325* | 0.087 |
| 8 | 0.879 | 0.384* | 0.339* | 0.394* | 0.308* | 0.073 |
| 9 | 0.853* | 0.341 | 0.258* | 0.338 | 0.254* | 0.043 |
| 10 | 0.826* | 0.345 | 0.239* | 0.336 | 0.231* | 0.059 |
| 11 | 0.876 | 0.357 | 0.329* | 0.371 | 0.332* | 0.049 |
| 12 | 0.861 | 0.323 | 0.290* | 0.323* | 0.306* | 0.054 |
| 13 | 0.855* | 0.310* | 0.299* | 0.313* | 0.309* | 0.066 |
| 14 | 0.811* | 0.210* | 0.211* | 0.235* | 0.224* | 0.039 |
| 15 | 0.825* | 0.251* | 0.232* | 0.260* | 0.255* | 0.045 |
| 16 | 0.796* | 0.262* | 0.218* | 0.250* | 0.243* | 0.056 |

Table 38: Results for Qwen-3 8B on SST-2. Comp./Suff. denote comprehensiveness/sufficiency; KS denotes Kernel SHAP; orig denotes the unpruned model. Significant changes ($p < 0.05$) are marked with *. Wilcoxon's test is used for Comp./Suff., and McNemar's test for accuracy.

| Pruned Layers | Accuracy | Comp. LIME | Suff. LIME | Comp. KS | Suff. KS | ECE |
|---|---|---|---|---|---|---|
| 0 (orig) | 0.919 | 0.354 | 0.367 | 0.350 | 0.367 | 0.058 |
| 1 | 0.908* | 0.355 | 0.367 | 0.340* | 0.359 | 0.060 |
| 2 | 0.893* | 0.340* | 0.353* | 0.323* | 0.345* | 0.061 |
| 3 | 0.860* | 0.297* | 0.324* | 0.298* | 0.316* | 0.072 |
| 4 | 0.856* | 0.298* | 0.308* | 0.298* | 0.303* | 0.071 |
| 5 | 0.919 | 0.367* | 0.369 | 0.372* | 0.361 | 0.062 |
| 6 | 0.916 | 0.367* | 0.363 | 0.372* | 0.361 | 0.062 |
| 7 | 0.894* | 0.338 | 0.333* | 0.344 | 0.315* | 0.050 |
| 8 | 0.913 | 0.359 | 0.328* | 0.359 | 0.311* | 0.041 |
| 9 | 0.823* | 0.315* | 0.222* | 0.283* | 0.237* | 0.023 |
| 10 | 0.656* | 0.341* | 0.179* | 0.290* | 0.210* | 0.203 |
| 11 | 0.664* | 0.259* | 0.266* | 0.266* | 0.275* | 0.211 |
| 12 | 0.800* | 0.310* | 0.274* | 0.283* | 0.285* | 0.075 |
| 13 | 0.885* | 0.341 | 0.284* | 0.328* | 0.301* | 0.023 |
| 14 | 0.890* | 0.319* | 0.262* | 0.299* | 0.283* | 0.017 |
| 15 | 0.882* | 0.332* | 0.258* | 0.313* | 0.273* | 0.017 |
| 16 | 0.843* | 0.280* | 0.216* | 0.276* | 0.232* | 0.040 |

Table 39: Results for Qwen-3 8B on RTE. Comp./Suff. denote comprehensiveness/sufficiency; KS denotes Kernel SHAP; orig denotes the unpruned model. Significant changes ($p < 0.05$) are marked with *. Wilcoxon's test is used for Comp./Suff., and McNemar's test for accuracy.

| Pruned Layers | Accuracy | Comp. LIME | Suff. LIME | Comp. KS | Suff. KS | ECE |
|---|---|---|---|---|---|---|
| 0 (orig) | 0.783 | 0.328 | 0.321 | 0.305 | 0.327 | 0.187 |
| 1 | 0.780 | 0.306* | 0.316 | 0.287* | 0.328 | 0.166 |
| 2 | 0.769 | 0.315 | 0.311* | 0.292 | 0.324 | 0.169 |
| 3 | 0.762 | 0.280* | 0.289* | 0.289* | 0.308* | 0.165 |
| 4 | 0.758* | 0.276* | 0.286* | 0.307 | 0.312* | 0.164 |
| 5 | 0.747* | 0.291* | 0.289* | 0.316 | 0.308* | 0.182 |
| 6 | 0.733* | 0.295* | 0.287* | 0.305 | 0.300* | 0.181 |
| 7 | 0.747* | 0.251* | 0.268* | 0.286* | 0.281* | 0.172 |
| 8 | 0.751* | 0.238* | 0.260* | 0.271* | 0.263* | 0.161 |
| 9 | 0.783 | 0.246* | 0.240* | 0.272* | 0.256* | 0.127 |
| 10 | 0.798 | 0.279* | 0.257* | 0.284* | 0.260* | 0.102 |
| 11 | 0.794 | 0.294* | 0.324 | 0.319 | 0.348* | 0.132 |
| 12 | 0.769 | 0.291* | 0.318 | 0.317 | 0.344* | 0.148 |
| 13 | 0.758 | 0.241* | 0.296* | 0.273* | 0.313 | 0.138 |
| 14 | 0.740* | 0.217* | 0.255* | 0.247* | 0.274* | 0.110 |
| 15 | 0.773 | 0.242* | 0.259* | 0.265* | 0.285* | 0.106 |
| 16 | 0.758 | 0.251* | 0.272* | 0.282 | 0.276* | 0.091 |

Table 40: Results for Qwen-3 4B on ARC Easy. Comp./Suff. denote comprehensiveness/sufficiency; KS denotes Kernel SHAP; orig denotes the unpruned model. Significant changes ($p < 0.05$) are marked with *. Wilcoxon's test is used for Comp./Suff., and McNemar's test for accuracy.

| Pruned Layers | Accuracy | Comp. LIME | Suff. LIME | Comp. KS | Suff. KS | ECE |
|---|---|---|---|---|---|---|
| 0 (orig) | 0.805 | 0.236 | 0.146 | 0.204 | 0.193 | 0.134 |
| 1 | 0.800 | 0.233 | 0.137* | 0.194* | 0.192 | 0.138 |
| 2 | 0.795* | 0.222* | 0.145 | 0.193 | 0.194 | 0.140 |
| 3 | 0.777* | 0.196* | 0.128 | 0.169* | 0.161 | 0.150 |
| 4 | 0.765* | 0.214 | 0.130 | 0.189 | 0.177 | 0.156 |
| 5 | 0.770* | 0.214 | 0.146 | 0.191 | 0.183 | 0.157 |
| 6 | 0.760* | 0.218 | 0.135 | 0.197 | 0.175 | 0.159 |
| 7 | 0.752* | 0.226 | 0.152* | 0.199 | 0.183 | 0.165 |
| 8 | 0.735* | 0.208* | 0.136 | 0.186 | 0.170 | 0.182 |
| 9 | 0.733* | 0.200* | 0.150 | 0.174 | 0.173 | 0.165 |
| 10 | 0.712* | 0.194* | 0.131 | 0.167 | 0.151* | 0.182 |
| 11 | 0.628* | 0.156* | 0.125 | 0.124* | 0.133* | 0.245 |
| 12 | 0.635* | 0.161* | 0.126 | 0.127* | 0.146* | 0.237 |
| 13 | 0.624* | 0.142* | 0.117 | 0.118* | 0.135* | 0.238 |
| 14 | 0.595* | 0.135* | 0.127 | 0.110* | 0.140* | 0.265 |
| 15 | 0.591* | 0.121* | 0.107* | 0.104* | 0.120* | 0.263 |
| 16 | 0.542* | 0.126* | 0.092* | 0.106* | 0.110* | 0.297 |

Table 41: Results for Qwen-3 4B on ARC Challenge. Comp./Suff. denote comprehensiveness/sufficiency; KS denotes Kernel SHAP; orig denotes the unpruned model. Significant changes ($p < 0.05$) are marked with *. Wilcoxon's test is used for Comp./Suff., and McNemar's test for accuracy.

| Pruned Layers | Accuracy | Comp. LIME | Suff. LIME | Comp. KS | Suff. KS | ECE |
|---|---|---|---|---|---|---|
| 0 (orig) | 0.508 | 0.273 | 0.126 | 0.232 | 0.180 | 0.391 |
| 1 | 0.507 | 0.268 | 0.137* | 0.231 | 0.176 | 0.394 |
| 2 | 0.497 | 0.268 | 0.149* | 0.229 | 0.191 | 0.397 |
| 3 | 0.491 | 0.275 | 0.159* | 0.241 | 0.194 | 0.397 |
| 4 | 0.477* | 0.266 | 0.161* | 0.231 | 0.190 | 0.410 |
| 5 | 0.477* | 0.264 | 0.160* | 0.239 | 0.196 | 0.415 |
| 6 | 0.467* | 0.263 | 0.157* | 0.234 | 0.197* | 0.426 |
| 7 | 0.459* | 0.259 | 0.140 | 0.230 | 0.177 | 0.431 |
| 8 | 0.445* | 0.239 | 0.131 | 0.205 | 0.169 | 0.448 |
| 9 | 0.448* | 0.244 | 0.147 | 0.209 | 0.179 | 0.428 |
| 10 | 0.430* | 0.241 | 0.141 | 0.211 | 0.176 | 0.445 |
| 11 | 0.406* | 0.199* | 0.095 | 0.168* | 0.125* | 0.450 |
| 12 | 0.408* | 0.214* | 0.115 | 0.180* | 0.148 | 0.452 |
| 13 | 0.404* | 0.190* | 0.108 | 0.164* | 0.140* | 0.444 |
| 14 | 0.381* | 0.180* | 0.087* | 0.153* | 0.119* | 0.475 |
| 15 | 0.370* | 0.190* | 0.098 | 0.151* | 0.129* | 0.483 |
| 16 | 0.340* | 0.149* | 0.063* | 0.129* | 0.083* | 0.502 |

Table 42: Results for Qwen-3 4B on OpenBookQA. Comp./Suff. denote comprehensiveness/sufficiency; KS denotes Kernel SHAP; orig denotes the unpruned model. Significant changes ($p < 0.05$) are marked with *. Wilcoxon's test is used for Comp./Suff., and McNemar's test for accuracy.

| Pruned Layers | Accuracy | Comp. LIME | Suff. LIME | Comp. KS | Suff. KS | ECE |
|---|---|---|---|---|---|---|
| 0 (orig) | 0.292 | 0.206 | 0.126 | 0.169 | 0.149 | 0.536 |
| 1 | 0.284 | 0.200 | 0.124 | 0.159 | 0.144 | 0.546 |
| 2 | 0.276 | 0.199 | 0.126 | 0.161 | 0.148 | 0.562 |
| 3 | 0.274 | 0.185* | 0.119 | 0.157 | 0.146 | 0.552 |
| 4 | 0.270 | 0.191* | 0.116 | 0.163 | 0.146 | 0.554 |
| 5 | 0.276 | 0.194 | 0.129 | 0.165 | 0.156 | 0.554 |
| 6 | 0.266 | 0.180* | 0.130 | 0.153 | 0.152 | 0.561 |
| 7 | 0.278 | 0.175* | 0.121 | 0.146* | 0.141 | 0.546 |
| 8 | 0.262 | 0.163* | 0.120 | 0.134* | 0.137 | 0.564 |
| 9 | 0.258* | 0.149* | 0.101* | 0.123* | 0.125 | 0.556 |
| 10 | 0.242* | 0.171* | 0.122 | 0.140 | 0.150 | 0.574 |
| 11 | 0.232* | 0.144* | 0.084* | 0.119* | 0.109* | 0.598 |
| 12 | 0.224* | 0.149* | 0.077* | 0.126* | 0.111 | 0.610 |
| 13 | 0.224* | 0.133* | 0.066* | 0.125* | 0.093* | 0.597 |
| 14 | 0.232* | 0.127* | 0.079* | 0.113* | 0.100* | 0.581 |
| 15 | 0.238* | 0.113* | 0.062* | 0.102* | 0.076* | 0.586 |
| 16 | 0.220* | 0.110* | 0.050* | 0.098* | 0.063* | 0.587 |

Table 43: Results for Qwen-3 4B on BoolQ. Comp./Suff. denote comprehensiveness/sufficiency; KS denotes Kernel SHAP; orig denotes the unpruned model. Significant changes ($p < 0.05$) are marked with *. Wilcoxon's test is used for Comp./Suff., and McNemar's test for accuracy.

| Pruned Layers | Accuracy | Comp. LIME | Suff. LIME | Comp. KS | Suff. KS | ECE |
|---|---|---|---|---|---|---|
| 0 (orig) | 0.851 | 0.476 | 0.248 | 0.379 | 0.381 | 0.105 |
| 1 | 0.851 | 0.477 | 0.264* | 0.398* | 0.380 | 0.106 |
| 2 | 0.850 | 0.480 | 0.285* | 0.397 | 0.402* | 0.106 |
| 3 | 0.850 | 0.465* | 0.309* | 0.399* | 0.398* | 0.101 |
| 4 | 0.845 | 0.420* | 0.355* | 0.381 | 0.389* | 0.096 |
| 5 | 0.849 | 0.455* | 0.373* | 0.394* | 0.403* | 0.105 |
| 6 | 0.843* | 0.441* | 0.332* | 0.367 | 0.389 | 0.101 |
| 7 | 0.838* | 0.441* | 0.267 | 0.377 | 0.353* | 0.099 |
| 8 | 0.799* | 0.420* | 0.251 | 0.346* | 0.302* | 0.106 |
| 9 | 0.813* | 0.353* | 0.169* | 0.329* | 0.304* | 0.057 |
| 10 | 0.800* | 0.358* | 0.173* | 0.342* | 0.309* | 0.077 |
| 11 | 0.616* | 0.201* | 0.167* | 0.221* | 0.177* | 0.171 |
| 12 | 0.628* | 0.220* | 0.147* | 0.208* | 0.184* | 0.143 |
| 13 | 0.661* | 0.246* | 0.179* | 0.242* | 0.206* | 0.081 |
| 14 | 0.647* | 0.181* | 0.141* | 0.208* | 0.168* | 0.094 |
| 15 | 0.601* | 0.163* | 0.128* | 0.173* | 0.145* | 0.117 |
| 16 | 0.537* | 0.164* | 0.140* | 0.186* | 0.164* | 0.154 |

Table 44: Results for Qwen-3 4B on TweetEval. Comp./Suff. denote comprehensiveness/sufficiency; KS denotes Kernel SHAP; orig denotes the unpruned model. Significant changes ($p < 0.05$) are marked with *. Wilcoxon's test is used for Comp./Suff., and McNemar's test for accuracy.

| Pruned Layers | Accuracy | Comp. LIME | Suff. LIME | Comp. KS | Suff. KS | ECE |
|---|---|---|---|---|---|---|
| 0 (orig) | 0.650 | 0.180 | 0.155 | 0.197 | 0.172 | 0.296 |
| 1 | 0.630* | 0.269* | 0.262* | 0.279* | 0.255* | 0.312 |
| 2 | 0.624* | 0.253* | 0.286* | 0.264* | 0.255* | 0.310 |
| 3 | 0.597* | 0.209* | 0.285* | 0.235* | 0.226* | 0.339 |
| 4 | 0.587* | 0.225* | 0.323* | 0.258* | 0.261* | 0.357 |
| 5 | 0.554* | 0.285* | 0.300* | 0.303* | 0.271* | 0.399 |
| 6 | 0.616* | 0.134* | 0.084* | 0.140* | 0.073* | 0.325 |
| 7 | 0.619* | 0.099* | 0.081* | 0.104* | 0.084* | 0.327 |
| 8 | 0.644 | 0.083* | 0.101* | 0.078* | 0.117* | 0.280 |
| 9 | 0.644 | 0.078* | 0.085* | 0.075* | 0.080* | 0.270 |
| 10 | 0.653 | 0.088* | 0.075* | 0.079* | 0.060* | 0.247 |
| 11 | 0.651 | 0.146 | 0.207* | 0.151 | 0.143 | 0.202 |
| 12 | 0.634 | 0.041* | -0.008* | 0.038* | -0.038* | 0.131 |
| 13 | 0.670 | 0.071* | 0.052* | 0.042* | 0.041* | 0.139 |
| 14 | 0.558* | 0.075* | -0.006* | 0.063* | -0.014* | 0.162 |
| 15 | 0.556* | 0.056* | -0.002* | 0.050* | -0.013* | 0.091 |
| 16 | 0.608* | 0.071* | 0.058* | 0.035* | 0.042* | 0.031 |

Table 45: Results for Qwen-3 4B on Rotten Tomatoes. Comp./Suff. denote comprehensiveness/sufficiency; KS denotes Kernel SHAP; orig denotes the unpruned model. Significant changes ($p < 0.05$) are marked with *. Wilcoxon's test is used for Comp./Suff., and McNemar's test for accuracy.

| Pruned Layers | Accuracy | Comp. LIME | Suff. LIME | Comp. KS | Suff. KS | ECE |
|---|---|---|---|---|---|---|
| 0 (orig) | 0.861 | 0.437 | 0.332 | 0.416 | 0.345 | 0.124 |
| 1 | 0.860 | 0.429* | 0.339* | 0.419 | 0.352* | 0.125 |
| 2 | 0.856 | 0.423* | 0.349* | 0.427 | 0.357* | 0.119 |
| 3 | 0.851* | 0.401* | 0.333 | 0.410 | 0.330 | 0.112 |
| 4 | 0.842* | 0.377* | 0.317 | 0.389* | 0.310* | 0.112 |
| 5 | 0.849 | 0.409* | 0.331 | 0.424 | 0.330 | 0.129 |
| 6 | 0.852* | 0.419* | 0.336 | 0.411 | 0.333 | 0.098 |
| 7 | 0.841* | 0.359* | 0.330 | 0.363* | 0.327 | 0.094 |
| 8 | 0.734* | 0.198* | 0.289* | 0.214* | 0.299* | 0.163 |
| 9 | 0.771* | 0.242* | 0.249* | 0.249* | 0.254* | 0.127 |
| 10 | 0.828* | 0.283* | 0.284* | 0.297* | 0.276* | 0.094 |
| 11 | 0.797* | 0.272* | 0.268* | 0.264* | 0.275* | 0.085 |
| 12 | 0.828* | 0.272* | 0.272* | 0.270* | 0.278* | 0.072 |
| 13 | 0.852 | 0.294* | 0.286* | 0.292* | 0.297* | 0.044 |
| 14 | 0.834* | 0.240* | 0.256* | 0.240* | 0.278* | 0.036 |
| 15 | 0.811* | 0.244* | 0.230* | 0.231* | 0.242* | 0.040 |
| 16 | 0.521* | 0.272* | 0.254* | 0.305* | 0.292* | 0.398 |

Table 46: Results for Qwen-3 4B on SST-2. Comp./Suff. denote comprehensiveness/sufficiency; KS denotes Kernel SHAP; orig denotes the unpruned model. Significant changes ($p < 0.05$) are marked with *. Wilcoxon's test is used for Comp./Suff., and McNemar's test for accuracy.

| Pruned Layers | Accuracy | Comp. LIME | Suff. LIME | Comp. KS | Suff. KS | ECE |
|---|---|---|---|---|---|---|
| 0 (orig) | 0.899 | 0.391 | 0.350 | 0.399 | 0.354 | 0.088 |
| 1 | 0.899 | 0.392 | 0.354 | 0.400 | 0.358 | 0.085 |
| 2 | 0.900 | 0.400* | 0.353 | 0.412* | 0.353 | 0.079 |
| 3 | 0.903 | 0.388 | 0.352 | 0.407* | 0.336* | 0.069 |
| 4 | 0.901 | 0.376* | 0.343 | 0.387* | 0.323* | 0.054 |
| 5 | 0.899 | 0.407* | 0.352* | 0.419* | 0.336* | 0.085 |
| 6 | 0.903 | 0.402 | 0.294* | 0.400 | 0.329* | 0.059 |
| 7 | 0.905 | 0.413* | 0.295* | 0.414 | 0.334 | 0.064 |
| 8 | 0.884 | 0.329* | 0.298* | 0.323* | 0.304* | 0.037 |
| 9 | 0.892 | 0.332* | 0.230* | 0.325* | 0.247* | 0.030 |
| 10 | 0.900 | 0.351* | 0.246* | 0.346* | 0.265* | 0.034 |
| 11 | 0.890 | 0.296* | 0.299* | 0.299* | 0.310* | 0.029 |
| 12 | 0.900 | 0.335* | 0.328 | 0.342* | 0.341 | 0.039 |
| 13 | 0.899 | 0.307* | 0.310* | 0.323* | 0.319* | 0.023 |
| 14 | 0.763* | 0.204* | 0.247* | 0.217* | 0.256* | 0.066 |
| 15 | 0.636* | 0.213* | 0.223* | 0.198* | 0.234* | 0.205 |
| 16 | 0.509* | 0.309* | 0.286* | 0.320* | 0.314* | 0.461 |

Table 47: Results for Qwen-3 4B on RTE. Comp./Suff. denote comprehensiveness/sufficiency; KS denotes Kernel SHAP; orig denotes the unpruned model. Significant changes ($p < 0.05$) are marked with *. Wilcoxon's test is used for Comp./Suff., and McNemar's test for accuracy.

| Pruned Layers | Accuracy | Comp. LIME | Suff. LIME | Comp. KS | Suff. KS | ECE |
|---|---|---|---|---|---|---|
| 0 (orig) | 0.758 | 0.317 | 0.308 | 0.298 | 0.309 | 0.191 |
| 1 | 0.758 | 0.313 | 0.309 | 0.299 | 0.300 | 0.193 |
| 2 | 0.758 | 0.300* | 0.296 | 0.291 | 0.290* | 0.202 |
| 3 | 0.758 | 0.280* | 0.303 | 0.291 | 0.304 | 0.176 |
| 4 | 0.758 | 0.283* | 0.319* | 0.288 | 0.329* | 0.180 |
| 5 | 0.783 | 0.303* | 0.332* | 0.338* | 0.331* | 0.187 |
| 6 | 0.747 | 0.301 | 0.346* | 0.325* | 0.352* | 0.199 |
| 7 | 0.762 | 0.308 | 0.340* | 0.325* | 0.357* | 0.181 |
| 8 | 0.758 | 0.302 | 0.384* | 0.292 | 0.397* | 0.146 |
| 9 | 0.769 | 0.283* | 0.295 | 0.276 | 0.330* | 0.085 |
| 10 | 0.769 | 0.276* | 0.286 | 0.259* | 0.305 | 0.073 |
| 11 | 0.715 | 0.255* | 0.255* | 0.262* | 0.286* | 0.102 |
| 12 | 0.697 | 0.247* | 0.241* | 0.241* | 0.254* | 0.090 |
| 13 | 0.697 | 0.234* | 0.240* | 0.194* | 0.233* | 0.051 |
| 14 | 0.610* | 0.233* | 0.252* | 0.192* | 0.208* | 0.114 |
| 15 | 0.643* | 0.214* | 0.236* | 0.172* | 0.186* | 0.090 |
| 16 | 0.625* | 0.203* | 0.240* | 0.114* | 0.149* | 0.099 |

