# OpenReview forum: "Don't Go Breaking My LLM: The Impact of Pruning Attention Layers on Explanation Faithfulness and Confidence Calibration"
_TMLR — Accepted by TMLR_

### Review · Reviewer_nawj · 2026-03-22

**Summary Of Contributions:**

This paper studies how attention layer pruning in LLMs affects the explanation faithfulness and confidence calibration via extensive experiments. The experiment is conducted across 5 LLMs and 8 datasets. The results suggest that while the accuracy remains relatively stable under attention pruning, the interpretability and calibration degrade independently of accuracy.

**Additional Comments:**

Please see the above sections.

**Audience:**

Yes

**Audience Explanation:**

- This paper studies how attention pruning would effect the explanation faithfulness and confidence calibration, while previous works usually focuses on accuracy. It fills a gap in the literature.
- The experiments are thorough. The experiments are conducted across 5 models and multiple datasets.
- The findings are insightful. While accuracy is relatively stable under attention pruning, the confidence and faithfulness degrade more apparently.

**Claims And Evidence:**

No

**Claims Explanation:**

- The paper is purely empirical, and theoretical analysis or intuitions are not provided.
- Only the attention modules are tested. Previous works (e.g., He et al., 2024) consider the whole block, the MLP layer and the attention modules, which is more comprehensive.

**Requested Changes:**

- It would be great if the authors can provide some theoretical analysis or intuitions about the observations.
- Furthermore, as the models trained via reinforcement learning exhibit better reasoning capabilities, and the parameters are changed mostly in the MLP layer [1]. Therefore, the incorporation of the study on the MLP would enhance the comprehensiveness of the manuscript.


[1] Jie Shao, Jianxin Wu. Who Reasons in the Large Language Models?

---

> ### Author Response · Authors · 2026-04-10
>
> We thank Reviewer nawj for the feedback and provide our response. As per TMLR policy, we will update the paper only once all reviews are available. In the meantime, we aim to initiate discussion on the requested points.
>
> ## Intuitions behind the observations
>
> We summarise the key intuitions, which we will add to the final paper's version.
>
> ### Pruning attention & explanation faithfulness
>
> As noted in the paper (Sec. 4.2, end of 2nd paragraph, p. 8): pruning attention layers can lead to explanations that are less faithful to the model. This behaviour could be partly attributed to a decrease in the model’s average confidence as more attention layers are pruned. Since both comprehensiveness and sufficiency rely on the model’s confidence scores, a drop in overall confidence tends to lower their values as well.
>
> We will extend this with the following text:
> In addition, layers are pruned based on the magnitude of the transformations they apply to their inputs [1]. While individually small, these layer-wise transformations may still encode information important for explanations, so removing them may contribute to degrading explanation quality.
>
> ### Short-label datasets yield unstable faithfulness trends under pruning
>
> As noted in the paper (Sec. 4.2, 3rd paragraph, p. 8): On datasets with short labels, comprehensiveness and sufficiency scores fluctuate unstably as attention layers are pruned, while they exhibit more stable trends on datasets with longer labels.
>
> We will extend this with the following text:
> This may be explained by the fact that faithfulness measures compute average attribution scores over label tokens. On short labels (one or two tokens), the measure is sensitive to token-level variation and noise, possibly leading to increased fluctuations. In contrast, longer labels aggregate over more tokens, reducing noise and therefore fluctuation trends.
>
> ### Pruning attention & changes in explanation faithfulness vs accuracy
>
> As noted in the paper (Sec. 4.2, 3rd paragraph, p. 8): pruning attention layers can lead to unstable fluctuations in explanation faithfulness that do not consistently mirror changes in model accuracy
>
> We will extend this with the following text:
> We hypothesise that layer pruning can reduce reliance on both informative and noisy features; however, this effect is not uniform across layers. Depending on which layers are removed, pruning may differentially affect informative and noisy signals, leading to non-monotonic behaviour and the observed fluctuations that do not necessarily align with accuracy drops.
>
> ### Pruning attention & confidence calibration
>
> As noted in the paper (Sec. 4.3, 2nd paragraph, p. 9): pruning attention layers can worsen the confidence calibration of the model.
>
> We will extend this with the following text:
> We hypothesise this is because layer pruning perturbs internal representations and logits, therefore models' confidence too. Calibration depends on alignment between confidence and accuracy [2], so even small logits shifts can degrade calibration by affecting confidence but not enough for a prediction flip.
> In addition, pruning attention layers may make the model over or under confident on individual samples, which further contributes to deterioration in confidence calibration.
>
> ### Short-label datasets yield unstable confidence calibration trends under pruning
>
> As noted in the paper (Sec. 4.3, 3rd paragraph, p. 9): calibration error scores fluctuate unstably on datasets with short labels as attention layers are pruned, while having more stable trends on datasets with longer average label lengths.
>
> We will extend this with the following text:
> We believe this trend is partially explained by the fact that for single-token or short-token predictions, model confidence depends heavily on a very small number of tokens, since log-likelihoods are computed over only a limited subset of the sequence. As a result, pruning-induced noise or information loss has a disproportionately large effect, leading to greater fluctuations in predicted confidence for short texts and, consequently, more unstable ECE estimates.
>
> ### Pruning attention & changes in confidence calibration vs accuracy
>
> As noted in the paper (Sec. 4.3, 3rd paragraph, p. 9): pruning attention layers can lead to unstable fluctuations of model calibration that do not consistently mirror changes in model accuracy.
>
> We will extend this with the following text:
> As previously mentioned, calibration depends on the alignment between confidence and accuracy, and even small logit perturbations can change confidence without affecting predictions, but changing the calibration scores. Additionally, pruning can remove redundant attention heads that are not critical for accuracy [3], yet, the same heads may be important for confidence estimation. As a result, we believe that the model can maintain similar accuracy while becoming overconfident or underconfident, leading to unstable and non-monotonic changes in calibration error.

---

> > ### Author Response · Authors · 2026-04-10
> >
> > ## Adding a study on pruning MLP would enhance the comprehensiveness of the manuscript.
> >
> > We will clarify this point in the revised paper (Sec. 3.2, end of 1st paragraph, p. 4) by adding the following paragraph:
> >
> >
> > We do not consider MLP pruning in our experiments, as, unlike attention layer pruning, which largely preserves performance, removing even a few (e.g., four) MLP layers leads to substantial accuracy drops. For example, in [4], Mistral’s MMLU accuracy drops from over 0.6 to less than 0.3 after removing four MLP layers and to less than 0.25 (near random) after removing 12 layers, while remaining above 0.6 when removing up to 12 attention layers. Similarly, in [1], Mistral’s average accuracy across multiple tasks decreases from 0.703 to 0.534 when removing 8 MLP layers, whereas removing 8 attention layers leads to 0.697 accuracy (negligible drop). For LLaMA, removing 8 MLP layers reduces accuracy from 0.682 to 0.619, compared to 0.681 for attention removal.
> >
> > In addition, prior work shows that MLP layers are critical for LLM reasoning [5]. These results indicate that removing MLP layers eliminates a critical component of the network and rapidly degrades model performance, often to near-random levels. In such settings, model outputs become unreliable, making it not meaningful to evaluate explanation quality on severely degraded predictions.
> >
> > For this reason, we focus on attention layer pruning for analysing trends in explanation faithfulness and confidence calibration under progressive layer removal.
> >
> >
> >
> > We hope that this clarifies the scope of the paper and the rationale for our attention layer pruning focus, and we would appreciate the reviewer’s feedback on the discussed points.
> >
> >
> > ## References
> >
> >
> > [1] He, Shwai, et al. "What matters in transformers? not all attention is needed." arXiv preprint arXiv:2406.15786 (2024).
> >
> > [2] Guo, Chuan, et al. "On calibration of modern neural networks." International conference on machine learning. PMLR, 2017.
> >
> > [3] Michel, Paul, Omer Levy, and Graham Neubig. "Are sixteen heads really better than one?." Advances in neural information processing systems 32 (2019).
> >
> > [4] Siddiqui, Shoaib Ahmed, et al. "A deeper look at depth pruning of LLMs." ICML 2024 Workshop on Theoretical Foundations of Foundation Models.
> >
> > [5] Shao, Jie, and Jianxin Wu. "Who Reasons in the Large Language Models?." The Thirty-ninth Annual Conference on Neural Information Processing Systems.

---

### Review · Reviewer_Waeu · 2026-04-10

**Summary Of Contributions:**

This paper provides the first systematic investigation into how pruning attention layers affects properties beyond raw accuracy, specifically explanation faithfulness and confidence calibration. While prior work suggests up to 33% of attention layers can be removed with minimal accuracy loss, this study reveals that such pruning often degrades the model’s reliability and interpretability.

Key Contributions:
- Faithfulness Analysis: The first evaluation of how attention pruning impacts the faithfulness of LIME and Kernel SHAP attributions, using comprehensiveness and sufficiency metrics.
- Calibration Study: The first analysis of how pruning affects the alignment between model confidence and accuracy via Estimated Calibration Error (ECE).
- Cross-Metric Comparison: A relational study showing that accuracy is often a poor proxy for faithfulness and calibration, as the latter two can fluctuate significantly even when accuracy remains stable.
- Extensive Benchmarking: Experiments across 5 LLMs (Mistral, Llama-2/3, Qwen-3 4B/8B) and 8 datasets.

Strengths:
- Novelty: Addresses an overlooked but critical gap in model compression research.
- Methodological Rigor: Uses established metrics (AOPC, ECE) and conducts a massive experimental campaign (6000+ GPU hours).
- Practical Insights: Provides actionable recommendations, such as including explainability and calibration metrics in pruning evaluations.

Weaknesses:
- Methodological Breadth: Focuses only on one pruning strategy (importance-based layer removal).
- Sample Size: Due to computational costs, faithfulness is evaluated on only 200 samples per dataset.

**Audience:**

Yes

**Audience Explanation:**

The shift toward deploying LLMs on edge devices makes pruning a critical area of study. Researchers and practitioners in Model Compression, AI Safety, and Explainable AI (XAI) will find the revelation that "accuracy-preserving" pruning can secretly break a model's interpretability and trustworthiness to be a vital warning for real-world deployments.

**Broader Impact Concerns:**

The paper already includes a Broader Impact and Ethics Statement. It correctly identifies that while pruning democratizes AI access, it introduces risks if the resulting models provide unfaithful explanations or overconfident, inaccurate predictions in sensitive domains. No additional concerns are raised; the current statement is sufficient.

**Claims And Evidence:**

Yes

**Claims Explanation:**

- Clear Trends: The authors provide comprehensive tables and figures showing that while accuracy drops are often modest, faithfulness and calibration metrics exhibit more severe and unstable degradation.
- Statistical Significance: The authors use appropriate statistical tests (McNemar’s for accuracy, Wilcoxon for faithfulness) and clearly mark significant changes in their results.
- Mechanistic Hypotheses: They offer plausible explanations for observed phenomena, such as why models with shorter labels exhibit more fluctuations (sensitivity to pruning-induced noise).

**Requested Changes:**

Critical to Acceptance:

- Clarification on Pruning Scope: Explicitly state in the main body (not just the limitations) that these results specifically apply to layer-wise pruning and may differ for unstructured or weight-based pruning.
- Aggregation Details: Provide more detail on how "accuracy fluctuations" were defined and counted for Figure 2 to ensure reproducibility.

Suggested Improvements:
- Additional Attribution Methods: Including a gradient-based or attention-based attribution method would strengthen the claim that these findings are not specific to surrogate-based explainers like LIME/SHAP.
- Plausibility vs. Faithfulness: As noted by the authors, a small study on whether the perceived quality of explanations (plausibility) changes alongside faithfulness would add depth.

---

> ### Author Response · Authors · 2026-04-22
>
> We thank Reviewer Waeu for their feedback and provide our response.
>
> ## Critical to acceptance:
>
> ### State in the main body that results are for layer-wise pruning and may differ for unstructured or weight-based pruning.
>
> We updated the paper (Sec. 3, 1st paragraph, p.3) to clarify this:
>
> We study the effects of layer pruning, where entire layers are removed from the models. All analyses and results in this work are defined within this setting; different pruning settings (e.g., unstructured or weight-based pruning) may exhibit different behaviours.
>
> ### More details on how accuracy fluctuations are defined (Fig. 2).
>
> We updated the paper (Sec. 4.1, last paragraph, p. 8) by referencing the appendix, where we now provide a detailed definition of how fluctuations are computed (Sec. App. A.4, new section, p.20-21). In particular:
>
> We compute accuracy, ECE, sufficiency, and comprehensiveness fluctuations. For faithfulness measures, scores are averaged over three random seeds used for attributions. We evaluate fluctuations per model by tracking score trends across pruning levels, from the unpruned model up to removing 16 attention layers. We first determine the initial trend (increasing or decreasing) as we prune the first layer; a fluctuation is counted when the score changes by at least 5\% in the opposite direction of the current trend, at which point the trend is updated, and tracking continues. We report the average number of fluctuations across models (y-axis). To account for tokenisation differences, the x-axis shows the average number of tokens per label across models on the dataset.
>
> In summary, this analysis shows how stable accuracy and faithfulness are under layer pruning, by quantifying how often their trends change. We relate this to the number of tokens used to represent labels.
>
> ## Suggested improvements:
>
> ### Additional attribution methods
>
> We agree that extending the analysis with additional attribution methods would be valuable for assessing the generalisability of our findings beyond surrogate-based explainers. We clarified this limitation more explicitly (Sec. 5, paragraph 4, p.13) and emphasised its implications by stating:
>
> Additionally, evaluating pruned models using a broader set of feature attribution methods would further strengthen the generalisability of the conclusions to explainers beyond surrogate-based methods.
>
> In this work, we focus on widely used attribution methods (LIME and SHAP), which perform well on faithfulness tasks and tend to yield more comprehensive explanations than attention-based methods [1]. Extending the empirical analysis to other attribution families (e.g., gradient and attention based methods) is a valuable direction for future work (Sec. 5, paragraph 3, p.13).
>
> ### Explore plausibility. Even a small study on whether plausibility changes alongside faithfulness would add depth
>
> We agree that evaluating plausibility adds an important perspective to the paper, and we include a plausibility study in the revision (methodology detailed in Sec. 3.5, p. 6; results in Sec. 4.4, p. 11–12), along with corresponding plots (Fig. 7, p. 12; and Figs. 24-33, p. 39–48).
>
> In the study, we collect human annotations of relevant evidence spans for randomly sampled instances across datasets, following [2]. We report annotation instructions (Sec. A.5, p. 21) and inter-annotator agreement (Table 4, p. 6) for transparency.
> We use the annotations to evaluate attributions' plausibility using standard measures (AUPRC, Precision@K, Recall@K, F1@K, and IoU), following prior work [3].
>
> Our results show that plausibility is broadly stable under pruning across datasets and measures, with mild degradation as pruning increases. In contrast to faithfulness, which varies substantially under pruning (Sec. 4.3).
> We hypothesise this is because pruning mostly preserves feature ranking, and plausibility measures compare the top features directly against human annotations.
>
> Overall, our findings indicate that pruned models' explanations can remain plausible for humans, even when they become less faithful to the model's decision process, highlighting a decoupling between the two aspects. We included this analysis to better characterise trade-offs between accuracy, faithfulness and plausibility, and updated the conclusion accordingly (Sec. 5, 3rd paragraph, p. 13).
>
> ## References
>
> [1] DeYoung, Jay, et al. "ERASER: A benchmark to evaluate rationalized NLP models." Proceedings of the 58th annual meeting of the association for computational linguistics. 2020.
>
> [2] Hayati, Shirley Anugrah, Dongyeop Kang, and Lyle Ungar. "Does bert learn as humans perceive? understanding linguistic styles through lexica." Proceedings of the 2021 Conference on Empirical Methods in Natural Language Processing. 2021.
>
> [3] Edin, Joakim, et al. "An unsupervised approach to achieve supervised-level explainability in healthcare records." Proceedings of the 2024 conference on empirical methods in natural language processing. 2024.

---

### Review · Reviewer_yZ3m · 2026-04-11

**Summary Of Contributions:**

This paper investigates the impact of attention layer pruning in LLMs on accuracy, explanation faithfulness, and confidence calibration. The results show that while accuracy is relatively robust to layer removal, interpretability and calibration metrics degrade significantly as pruning increases. Notably, these degradations do not correlate with accuracy drops. Based on these findings, the authors argue that pruned LLMs should be evaluated using comprehensive metrics beyond mere accuracy.

**Audience:**

Yes

**Audience Explanation:**

Moving beyond accuracy to study the "hidden costs" of pruning on model reliability is a valuable contribution. The call for more comprehensive evaluation metrics is well-justified and highly relevant to the community.

**Claims And Evidence:**

Yes

**Claims Explanation:**

The experiments are thorough and provide solid empirical grounding for the paper's conclusions.

**Requested Changes:**

**Weaknesses**

* Limited Mechanistic Insight: The paper primarily documents that explanation faithfulness and calibration degrade, which, while practically useful, is somewhat expected when model capacity is reduced. The contribution remains largely observational.
* Lack of Actionable Mitigation: The work stops short of exploring the underlying mechanisms driving these degradations. It would be significantly strengthened by investigating whether specific pruning strategies could mitigate these effects, or if certain layers are more critical for interpretability than for raw accuracy.

**Requested Changes**

* Deepen the Analysis: As noted above, I encourage the authors to move beyond reporting observations. Please consider adding discussions or preliminary experiments regarding the "why" behind these results, or proposing how pruning methodologies might be adapted to preserve calibration and faithfulness.

Minor changes:
* The font sizes for axis ticks and labels in several figures are currently too small to be easily readable. Please increase the font size in the final version to ensure accessibility.

---

> ### Author Response · Authors · 2026-04-17
>
> We thank Reviewer yZ3m for their feedback and provide our response below.
>
> ### Discussion on observed trends, including potential causes:
>
>
> #### Pruning attention & explanation faithfulness
>
> As noted in the paper (Sec. 4.2, end of 2nd paragraph, p. 8): pruning attention layers can lead to explanations that are less faithful to the model. This behaviour could be partly attributed to a decrease in the model’s average confidence as more attention layers are pruned. Since both comprehensiveness and sufficiency rely on the model’s confidence scores, a drop in overall confidence tends to lower their values as well.
>
> We extended this with the following text:
> In addition, layers are pruned based on the magnitude of the transformations they apply to their inputs [1]. While individually small, these layer-wise transformations may still encode information important for explanations, so removing them may contribute to degrading explanation quality.
>
> #### Short-label datasets yield unstable faithfulness trends under pruning
>
> As noted in the paper (Sec. 4.2, 3rd paragraph, p. 8): On datasets with short labels, comprehensiveness and sufficiency scores fluctuate unstably as attention layers are pruned, while they exhibit more stable trends on datasets with longer labels.
>
> We extended this with the following text:
> This may be explained by the fact that faithfulness measures compute average attribution scores over label tokens. On short labels (one or two tokens), the measure is sensitive to token-level variation and noise, possibly leading to increased fluctuations. In contrast, longer labels aggregate over more tokens, reducing noise and therefore fluctuation trends.
>
> #### Pruning attention & changes in explanation faithfulness vs accuracy
>
> As noted in the paper (Sec. 4.2, 3rd paragraph, p. 9): pruning attention layers can lead to unstable fluctuations in explanation faithfulness that do not consistently mirror changes in model accuracy
>
> We extended this with the following text:
> We hypothesise that layer pruning can reduce reliance on both informative and noisy features; however, this effect is not uniform across layers. Depending on which layers are removed, pruning may differentially affect informative and noisy signals, leading to non-monotonic behaviour and the observed fluctuations that do not necessarily align with accuracy drops.
>
> #### Pruning attention & confidence calibration
>
> As noted in the paper (Sec. 4.3, 2nd paragraph, p. 10): pruning attention layers can worsen the confidence calibration of the model.
>
> We extended this with the following text:
> We hypothesise this is because layer pruning perturbs internal representations and logits, therefore models' confidence too. Calibration depends on alignment between confidence and accuracy [2], so even small logits shifts can degrade calibration by affecting confidence but not enough for a prediction flip.
> In addition, pruning attention layers may make the model over or under confident on individual samples, which further contributes to deterioration in confidence calibration.
>
> #### Short-label datasets yield unstable confidence calibration trends under pruning
>
> As noted in the paper (Sec. 4.3, 3rd paragraph, p. 10): calibration error scores fluctuate unstably on datasets with short labels as attention layers are pruned, while having more stable trends on datasets with longer average label lengths.
>
> We extended this with the following text:
> We believe this trend is partially explained by the fact that for single-token or short-token predictions, model confidence depends heavily on a very small number of tokens, since log-likelihoods are computed over only a limited subset of the sequence. As a result, pruning-induced noise or information loss has a disproportionately large effect, leading to greater fluctuations in predicted confidence for short texts and, consequently, more unstable ECE estimates.
>
> #### Pruning attention & changes in confidence calibration vs accuracy
>
> As noted in the paper (Sec. 4.3, 3rd paragraph, p. 11): pruning attention layers can lead to unstable fluctuations of model calibration that do not consistently mirror changes in model accuracy.
>
> We extended this with the following text:
> As previously mentioned, calibration depends on the alignment between confidence and accuracy, and even small logit perturbations can change confidence without affecting predictions, but changing the calibration scores. Additionally, pruning can remove redundant attention heads that are not critical for accuracy [3], yet, the same heads may be important for confidence estimation. As a result, we believe that the model can maintain similar accuracy while becoming overconfident or underconfident, leading to unstable and non-monotonic changes in calibration error.

---

> > ### Author Response · Authors · 2026-04-17
> >
> > ### Adaptation of pruning methodologies
> >
> > We have also expanded the results to include recommendations on how the effect of layer pruning on faithfulness and confidence calibration may be mitigated (New Sec. 4.4, p. 11). We include the following discussion:
> >
> > We present recommendations for designing layer-wise pruning strategies that explicitly account for explanation faithfulness and confidence calibration, motivated by the degradation patterns observed in our results.
> >
> > One direction is to design pruning methods that explicitly account for faithfulness and confidence calibration, as preserving these properties ensures that pruned models remain not only accurate but also reliable in their explanations and predictive confidence.
> > This can be achieved by iteratively removing layers and reverting changes when they compromise these properties, or by incorporating faithfulness and calibration directly into the scoring function used to determine the order of layer pruning. Alternatively, post-hoc correction methods can be applied after pruning using adapters (e.g., LoRA) to recover performance while explicitly optimising faithfulness and confidence calibration.
> >
> >
> > ### Increase the plots' font size.
> > We have increased the font size in all plots in the revised version of the paper.
> >
> >
> > ## References
> >
> >
> > [1] He, Shwai, et al. "What matters in transformers? not all attention is needed." arXiv preprint arXiv:2406.15786 (2024).
> >
> > [2] Guo, Chuan, et al. "On calibration of modern neural networks." International conference on machine learning. PMLR, 2017.
> >
> > [3] Michel, Paul, Omer Levy, and Graham Neubig. "Are sixteen heads really better than one?." Advances in neural information processing systems 32 (2019).

---

### Decision · Action_Editor_jfr7 · 2026-06-04

**Recommendation:** Accept as is

**Audience:**

Yes

**Audience Explanation:**

Pruning as well as XAI for LLMs is of broad interest to TMLR's audience.

**Claims And Evidence:**

Yes

**Claims Explanation:**

The experiments in the paper are quite exhaustive, as noted by all the reviewers.
And much more explanations have also gotten added in the discussion phase.